# Pi-Pi contacts are an overlooked protein feature relevant to phase separation

Robert McCoy Vernon[1], Paul Andrew Chong[1], Brian Tsang[1,2], Tae Hun Kim[1], Alaji Bah[1†], Patrick Farber[1‡], Hong Lin[1], Julie Deborah Forman-Kay[1,2]*

[1]Program in Molecular Medicine, Hospital for Sick Children, Toronto, Canada; [2]Department of Biochemistry, University of Toronto, Toronto, Canada

**Abstract** Protein phase separation is implicated in formation of membraneless organelles, signaling puncta and the nuclear pore. Multivalent interactions of modular binding domains and their target motifs can drive phase separation. However, forces promoting the more common phase separation of intrinsically disordered regions are less understood, with suggested roles for multivalent cation-pi, pi-pi, and charge interactions and the hydrophobic effect. Known phase-separating proteins are enriched in pi-orbital containing residues and thus we analyzed pi-interactions in folded proteins. We found that pi-pi interactions involving non-aromatic groups are widespread, underestimated by force-fields used in structure calculations and correlated with solvation and lack of regular secondary structure, properties associated with disordered regions. We present a phase separation predictive algorithm based on pi interaction frequency, highlighting proteins involved in biomaterials and RNA processing.
DOI: https://doi.org/10.7554/eLife.31486.001

*For correspondence:
forman@sickkids.ca

Present address: †Department of Biochemistry and Molecular Biology, SUNY Upstate Medical University, New York, United States; ‡Zymeworks, Vancouver, Canada

Competing interests: The authors declare that no competing interests exist.

## Introduction

Protein phase separation has important implications for cellular organization and signaling (*Mitrea and Kriwacki, 2016*; *Brangwynne et al., 2009*; *Su et al., 2016*), RNA processing (*Sfakianos et al., 2016*), biological materials (*Yeo et al., 2011*) and pathological aggregation (*Taylor et al., 2016*). For some systems, multivalent interactions between modular binding domains and cognate peptide motifs underlie phase-separation (*Li et al., 2012*; *Banjade and Rosen, 2014*). However, many phase-separating proteins contain large intrinsically disordered protein regions (IDRs) with low complexity sequences that do not form stable folded structure (reviewed in [*Mitrea and Kriwacki, 2016*; *Chong and Forman-Kay, 2016*]), including the Nephrin intracellular domain (NICD) (*Pak et al., 2016*), polyglutamine tracts (*Crick et al., 2013*), tropoelastin (*Yeo et al., 2011*), FUS (*Burke et al., 2015*; *Kato et al., 2012*), Ddx4 and the homologous LAF-1 (*Nott et al., 2015*; *Elbaum-Garfinkle et al., 2015*) and FG-repeat nucleoporins (*Frey et al., 2006*). The underlying physical principles and chemical interactions that drive phase separation in these IDRs are not well understood. Multivalent (*Li et al., 2012*; *Pierce et al., 2016*) electrostatic (*Pak et al., 2016*; *Lin et al., 2016*) and cation-pi (*Nott et al., 2015*; *Kim et al., 2016*; *Sherrill, 2013*) interactions and the hydrophobic effect (*Yeo et al., 2011*) have all been proposed to contribute to IDR phase separation, the latter suggested to be dominant for tropoelastin (*Luan et al., 1990*). For Ddx4, electrostatic interactions between charge blocks has been demonstrated (*Nott et al., 2015*; *Lin et al., 2016*). The abundance of Phe-Gly/Gly-Phe and Arg-Gly/Gly-Arg dipeptides in Ddx4 and the fact that Phe to Ala mutations inhibit phase separation also point to pi-pi and/or cation-pi interactions. The Phe-Gly repeats in FG nucleoporins similarly indicate pi-pi interactions, but the lack of aromatics in elastins and designed phase-separating sequences (*Quiroz and Chilkoti, 2015*) seems to suggest that they are not essential. Clearly, a number of physical interactions may be sufficient for driving phase

separation without being universally necessary, and a better understanding of these interactions is needed to define the balance of forces biological systems use for driving protein phase transitions.

Although pi-pi interactions are commonly associated with aromatic rings, where interaction energy is thought to involve induced quadrupolar electrostatic interactions (*Sherrill, 2013*), π (pi) orbitals of bonded sp$^2$-hybridized atoms are also found in peptide backbone amide groups and sidechain amide, carboxyl or guanidinium groups. Sidechains with pi bonds include Tyr, Phe, Trp, His, Gln, Asn, Glu, Asp and Arg. Small residues with relatively exposed backbone peptide bonds include Gly, Ser, Thr and Pro. Notably, low complexity IDRs implicated in phase separation of FUS, EWS, hnRNPA1, TIA-1, TDP-43 and the RNA Pol II C-terminal domain (CTD) (*Mitrea and Kriwacki, 2016*; *Taylor et al., 2016*; *Kato et al., 2012*) are very enriched in these residues that have high potential for formation of pi-pi interactions, relative to average occurrence in the proteome. Even elastins, which lack sidechain pi groups but have Val-Pro-Gly-Xxx-Gly repeats (*Yeo et al., 2011*), are highly enriched in Pro and Gly residues with exposed pi-containing peptide backbones.

Given the high frequency of aromatic residues, arginine and glutamine in many phase-separating sequences, we were motivated to investigate the structural behavior of pi-pi interactions in order to better understand how their observed physical behavior relates to their potential role in phase separation. We first characterized the frequency and correlations of pi interactions in a non-redundant protein set from the RCSB protein data bank (PDB) of folded proteins (*Berman et al., 2000*). We discovered that planar pi-pi contacts involving a non-aromatic group, including those involving the backbone amide group, are the predominant form of pi-pi interaction, and we showed that planar pi-contact rates can be predicted from sequence. Then, we tested the relevance of these planar pi-pi interactions to phase separation by training a phase separation predictor using only these expected pi-contact rates. We then demonstrated that three of the predicted proteins, FMR1, a multifunctional RNA-binding protein and a neuronal granule component (*El Fatimy et al., 2016*), engrailed-2, a DNA

-binding homeobox protein, and the pAP isoform of the Human cytomegalovirus capsid scaffolding protein phase separate in isolation in vitro. Analysis of predictions for the full human proteome suggests strong phase-separation propensities for proteins involved in biomaterials and RNA processing, with likely regulation by splicing and post-translational modifications (PTMs).

## Results

### Prevalence of Pi contacts in the PDB

To determine the frequency of pi-pi interactions and better understand their nature and physical properties, we performed a bioinformatics analysis of folded proteins. We searched the PDB for pi-pi interactions by measuring contact distances between planar surfaces and comparing planar orientations (see Materials and methods), choosing to focus on interactions involving pi-orbital planar surfaces as this category shows the most enrichment over expectations, both in terms of overall frequency (*Appendix 1—figure 1*) and in relation to resolution. Face-to-face planar pi-pi contacts were defined using a simple distance- and orientation-based metric designed to consistently capture this enrichment across diverse sp$^2$-containing groups (*Appendix 1—figure 1A,B,C*).

Our analysis was originally intended to explore the known interactions of aromatic sidechains with each other and with arginine, but in order to provide a control group we defined our contact parameters in a way that allowed us to treat all sp$^2$ groups in the same fashion. In high-resolution ($\leq$1.8 Å) and low R-factor ($\leq$0.18) protein crystal structures (N = 5718), we found that planar pi-pi stacking interactions involving non-aromatic atoms outnumber aromatic-aromatic stacking interactions by approximately 13 to 1 (*Figure 1A* and *Appendix 1—table 1*) suggesting that, while aromatic sidechains may be enriched in stacking interactions relative to their frequency, there is a more general role for pi-contacts that involve non-aromatic atoms. The vast majority of planar pi-orbital contacts in proteins involve one of five non-aromatic sp$^2$-hybridized sidechains or the peptide bond itself. Fully 36% of observed pi-pi stacking interactions do not involve an aromatic partner, with face-to-face planar contacts between different backbone peptide bonds occurring as often as aromatic face-to-face contacts (*Figure 1A*). Across the high-resolution set, we observe that 58% of heavy atoms are sp$^2$, of which 10.5% are involved in pi-stacking. Furthermore, 28% of heavy atoms that are not directly involved in pi-stacking are found within van der Waals (VDW) contact distance (4.9 Å) of

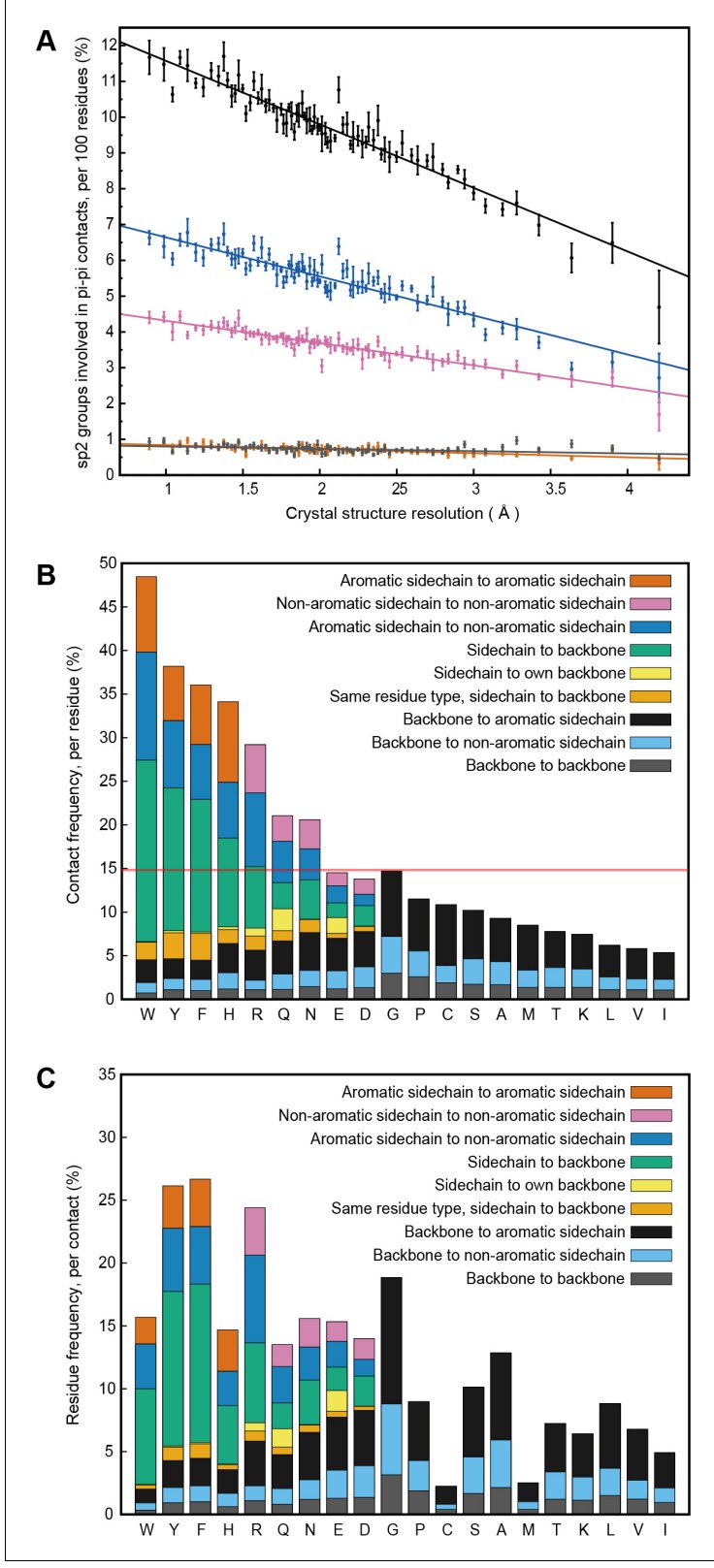

**Figure 1.** PDB statistics for planar pi-pi interactions. (**A**) Average number of sp$^2$ groups involved in planar pi-pi contacts per 100 protein residues binned by crystal structure resolution. Values are shown for contacts defined by the nature of the involved sp$^2$ groups, with all groups in black, aromatic to non-aromatic sp$^2$ in blue, non-aromatic to non-aromatic in pink, backbone to backbone in gray, and aromatic to aromatic in orange. Error bars show

*Figure 1 continued on next page*

*Figure 1 continued*

bootstrap SEM. (**B**) Planar pi-pi contact interaction frequencies for each residue type, with the average across all residue types shown as a red line, and (**C**) frequency of each residue type in contributing to planar pi-pi interactions, with bars showing overall frequency colored proportionally by the nature of the contact partners.

*Figure 1—source data 1* and *2*.

DOI: https://doi.org/10.7554/eLife.31486.002

The following source data and figure supplements are available for figure 1:

**Source data 1.** Pi-Pi contact annotations for the full PDB set.

DOI: https://doi.org/10.7554/eLife.31486.005

**Source data 2.** Residue and amino acid counts for the full PDB set.

DOI: https://doi.org/10.7554/eLife.31486.006

**Figure supplement 1.** Proportion of sidechain to backbone VDW contacts that satisfy planar contact criterion.

DOI: https://doi.org/10.7554/eLife.31486.003

**Figure supplement 2.** Selected sidechain-to-sidechain contact frequencies by resolution.

DOI: https://doi.org/10.7554/eLife.31486.004

atoms that are. Thus, planar pi-pi interactions form a common feature of the protein chemical environment. Comparisons to previous work showing that aromatic-aromatic interactions in proteins are instead biased toward face-to-edge or parallel displaced geometries (*Hunter et al., 1991*; *Martinez and Iverson, 2012*) are complicated by the observations that VDW contacts between aromatic sidechains coincide with face-to-face pi-pi stacking to a third $sp^2$-hybridized group 49% of the time, and that parallel displacement often accommodates an additional non-aromatic pi-contact to the same planar surface.

Analysis of protein structures showed that the frequencies of planar pi interactions strongly correlate with the power of the experimental data to constrain the structure and with the fit to the data. We identified a linear relationship between contact frequencies and the resolution of crystal structures (*Figure 1A*). We identify a similar dependence of contact frequency on the relative number of sidechain-specific distance constraints in NMR structures (*Appendix 1—figure 2A*) and confirm that the dependence on resolution persists for identical sequences solved multiple times at different resolutions (*Appendix 1—figure 2B*). These data suggest that the relative importance of pi-pi interactions are underestimated in the force-fields that are used in the structure calculations and thus appear more frequently in structures that are heavily constrained by experimental observations. In addition, pi-pi contact frequencies for amino acid and other small $sp^2$-containing ligands bound to proteins (including non-aromatic ligands) are higher than the frequencies observed for the same chemical group found within proteins, despite or perhaps because of their relative freedom of movement (*Appendix 1—table 2*).

To examine whether $sp^2$-containing sidechains engage in stacking behavior beyond what could be expected for average contact frequencies and overall packing considerations, we determined sidechain contacts to backbone peptide groups, focusing on the percentage of VDW contacts (with two or more pairs of atoms within 4.9 Å) which satisfy our planar-pi criterion, and then compared the frequencies observed for $sp^2$ sidechain groups to those observed for planar surfaces on the terminal end of $sp^3$ sidechains, using atom groups as listed in the Materials and methods section. This metric addresses the issue of amino acid composition effects by taking advantage of the even distribution of backbone groups and allows for normalization of contact frequency for sidechains of different size. Enrichment of $sp^2$ planar contacts relative to $sp^3$ is clearly observed for all $sp^2$ sidechains except Asn and Gln, which our previous analysis showed are more likely to form contacts with their backbone than with their sidechains (*Figure 1—figure supplement 1*). Further analysis of the relative frequency of planar pi VDW contacts to other VDW contacts as a function of resolution demonstrates that for some contact types the increased pi-contact frequencies with increasing resolution (lower values in Å) are at the expense of decrease in other VDW contacts, suggesting that these contacts represent a specific geometric constraint present in the experimental data, rather than an overall increase in VDW contact frequency at higher resolution (*Figure 1—figure supplement 2*).

Aromatic groups are known to have favorable interactions with other aromatics, with the peptide backbone (*Tóth et al., 2001*), and with charged groups. We observed that the guanidine group of arginine is either the first or second most likely planar pi-stacking partner for any given aromatic

sidechain, a phenomenon previously described as cation-pi (*Sherrill, 2013*; *Martinez and Iverson, 2012*). However, we also observed planar stacking interactions between non-aromatic groups of all kinds, including both anion-to-anion and cation-to-cation, with relative frequencies shown in *Figure 1B,C*. Surprisingly, 3.6% of arginine sidechains are found in direct, parallel pi-stacking contact with another arginine, despite repulsive charges (*Figure 1—figure supplement 2*), suggesting that these guanidinium interactions are better described as pi-pi, rather than cation-pi (example shown in *Figure 2A*).

Modeling and analysis of protein structures typically involves the use of coarse-grained energy functions. To test the degree to which contact frequencies in solved structures derive from experimental constraints, rather than the force fields used, we explored how well planar pi interactions are captured by the simple energy functions used in certain protein modeling protocols. We examined a few different modeling protocols by either running available methods or downloading pre-computed datasets (see Materials and methods). In general, planar pi-pi contacts were lost during simulations (*Appendix 1—figure 3A*) and energy minimization (*Appendix 1—figure 3B*). In one older molecular dynamics simulation of folded proteins, made available for 100 proteins via Dynameomics (*Kehl et al., 2008*), 90% of the planar pi-pi contacts found in the starting structures were lost during simulation, with the majority being lost within the first few simulation steps. Similarly, modeling of the energetic effect of mutations, the ΔΔG of unfolding, using both FOLDX (*Schymkowitz et al., 2005*) and Rosetta (*Kellogg et al., 2011*), shows decreased prediction accuracy at positions involved in pi-contacts (*Appendix 1—figure 3C-F*), based on comparison to a reference set of ΔΔG measurements (*Bava et al., 2004*). These observed issues in modeling pi-contacts may be overcome by more recent and sophisticated energy functions, but our results are consistent with the inherent energetic importance of planar pi interactions, rather than their observation being due to simple force fields used in refining protein structures.

## Enrichment of pi-pi contacts in catalytic, capping and RNA-binding sites

For exploring the contribution of pi contacts to general structural and functional properties of proteins, we examined contact enrichment for $sp^2$ groups found in a diverse range of interactions. We observe increased frequency of pi-pi contacts at positions with known catalytic function (*Furnham et al., 2014*), with enrichment of $1.87 \pm 0.07$ overall and $1.42 \pm 0.07$ when normalized by residue type (*Appendix 1—table 3*), with pi-pi contacts often playing a role in defining the geometry of the active site (*Figure 2B*) or forming networks of pi-pi contacts. We find that hydrogen bond frequency increases at $sp^2$ sidechains involved in pi-contacts (*Appendix 1—figure 4*), and when $sp^2$ groups hydrogen bond each other we observe increased frequencies of a third $sp^2$ group being found in simultaneous pi-stacking to both the donor and acceptor groups of the hydrogen bond (*Appendix 1—figure 1F* and *Figure 2C*), suggesting potential cooperativity via the electrostatic and geometric stabilization of the bond. We also observe up to 20-fold enrichment at the ends of secondary structure elements, relative to the median backbone contact rate of 1.7%, with enriched positions often involving the last hydrogen bond made within a helix or at the end of a strand (*Figure 2D* and *Appendix 1—figure 5*), commonly placing them in the context of local capping motifs thought to stabilize secondary structure (*Richardson and Richardson, 1988*). Finally, we find that protein-RNA interactions typically involve pi-pi contacts, especially with arginine. A detailed description of these observations is included in Appendix 1.

## Correlation of pi-pi contacts with solvation and lack of regular structure

Interactions at the surface of a protein are typically in competition with solvent and their enthalpic contribution often decreases with solvent exposure, as for protein-protein hydrogen bonds (*Efimov and Brazhnikov, 2003*). Planar pi-pi interactions, in contrast, cannot be formed with water, but often involve groups with hydrogen bond acceptors and donors; thus, we predicted that the frequency of pi-pi interactions in proteins could be increased in more solvated environments. To test this, we identified high-resolution structures with an abundance of solved water and then counted the observed solvent interactions by the number of water oxygen atoms within a broad VDW contact radius (4.9 Å) to each residue. We saw an unambiguous positive correlation between the number of water contacts and the probability that a residue is involved in a planar pi-pi-contact, with a significant increase in average probability observed for each additional water contact (*Figure 3A,B*),

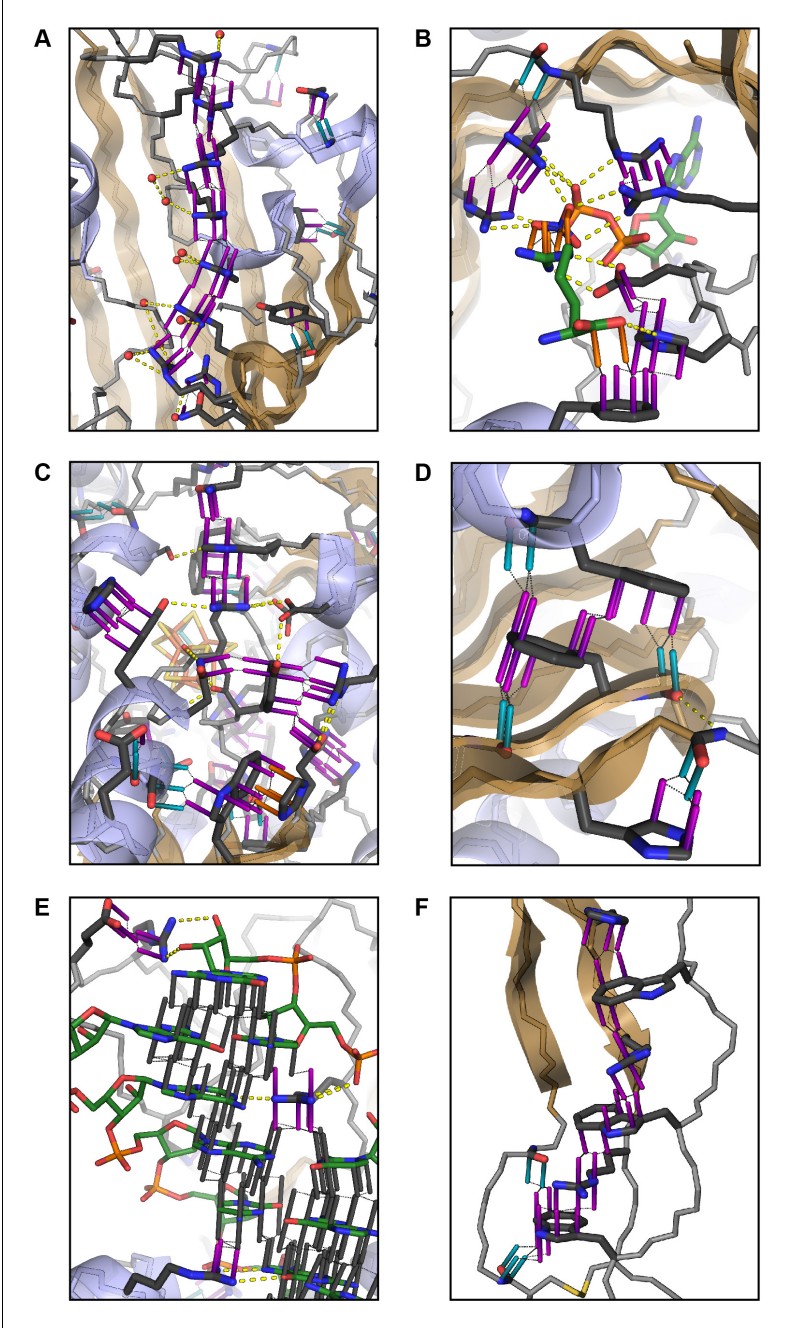

**Figure 2.** Examples of planar pi-pi contacts in folded protein structures. Pi-pi interactions shown using rods to describe the normal vector of the plane. Rods extend to a carbon VDW radius of 1.7 Å, colored by category with sidechain groups in purple, backbone in blue, small molecule ligands in orange, and RNA in gray. Ligand molecules are green, with relevant water molecules shown as red spheres and hydrogen bonds as yellow lines. (A) Arginine ladder motif in Porin P (PDB:2o4v). (B) Catalytic site from arginine kinase (PDB:1m15). (C) Network of interactions in nitrogenase (PDB: 3u7q). (D) Backbone/sidechain contacts at the ends of secondary structure elements (PDB:4b93). (E) RNA-binding interactions (PDB: 4lgt). (F) Interaction network stacked between disulfide bonds (PDB: 4v2a).

DOI: https://doi.org/10.7554/eLife.31486.007

climbing even as the average number of protein:protein VDW contacts declines. This relationship is true for the general case (unspecified residue identity) and is also individually true for each of the

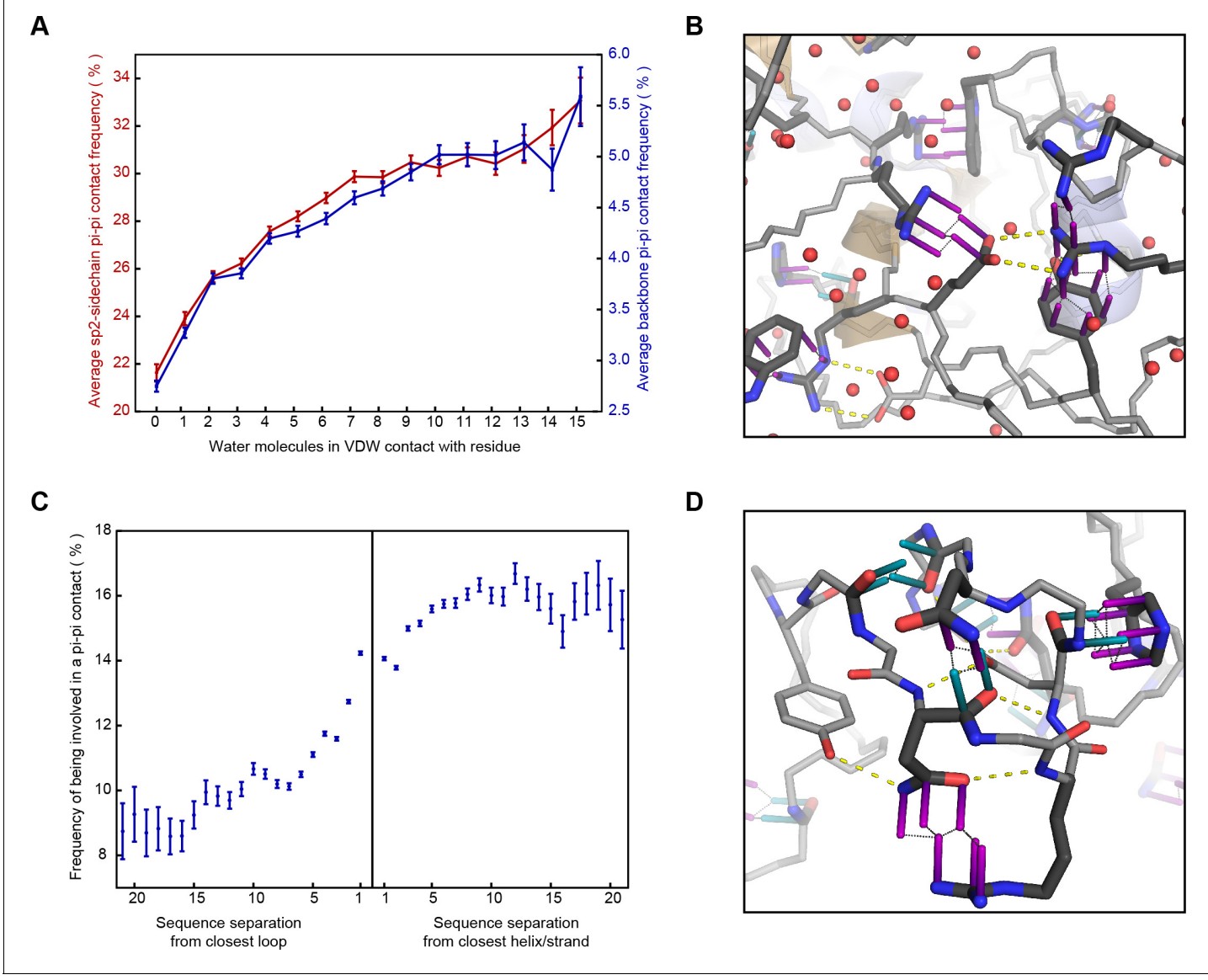

**Figure 3.** Correlation of planar pi-pi interactions with solvent and lack of secondary structure. (**A**) Contact frequency for sidechain groups (red) and backbone (blue) increases with the total number of solved water molecules within 4.9 Å of the residue, based on structures with >1 water oxygen per residue, including all molecules within 8 Å of the chain of interest, including symmetry partners. (**B**) Representative example of a pi-stacked sidechain in contact with 11 water molecules (PDB:4u98), showing how the interaction does not appear to compete with solvent. (**C**) Mean contact frequency vs. sequence distance from regular secondary structure and loop/turn regions. (**D**) Example of the range of interactions found >10 residues from helix/strand secondary structure (PDB:4b4h).

DOI: https://doi.org/10.7554/eLife.31486.008

The following figure supplements are available for figure 3:

**Figure supplement 1.** Effect of solvation on pi-pi category frequencies.
DOI: https://doi.org/10.7554/eLife.31486.009
**Figure supplement 2.** Enrichment of pi-pi contacts, relative to overall VDW contacts, as a function of the number of interactions with water.
DOI: https://doi.org/10.7554/eLife.31486.010

nine residues with pi orbital-containing sidechains. However, both the contact frequencies and the amino acid frequencies themselves increase with a greater dependence on solvation for non-aromatic residues, especially for the charged amino acids Arg, Glu and Asp, such that non-aromatic contacts become the dominant form of interaction at high solvation levels (*Figure 3—figure supplement 1*). The relative increase is highest for contacts involving sidechains of like-charge, especially

arginine (*Figure 3—figure supplement 2*), suggesting that solvation plays a role in the strength of the interactions.

Of relevance to intrinsically disordered protein regions that mediate interactions, we find that planar pi-pi interactions occur more often at positions with properties associated with disorder; they are more prevalent in proteins having overall less rigid secondary structure (with contacts for coil/loop/turn > strand > helix), especially disulfide bond containing proteins (*Figure 2F*), and in sequences that are locally enriched in residue types associated with backbone flexibility or breaking secondary structure (Gly, Ser, Thr, Pro) (*Appendix 1—figure 6*). Considering planar pi-pi contact frequencies as a function of the sequence position relative to secondary structure elements, we find that the frequency is highest in long loops, showing a sigmoidal relationship when transitioning from order to disorder that goes from 9.5% probability for residues >7 positions away from the closest loop/turn to 16% for residues >7 positions away from the closest helix/strand (*Figure 3C,D*).

## Pi-pi contacts in protein interactions

To test whether these interactions are compatible with the multivalent interactions involved in phase separation, we examined contact statistics for protein interactions, comparing sidechain pi-pi interaction frequencies within a chain to those between chains. We classified interfaces as sequence- or complex-specific (between different chains of a crystal structure) and opportunistic (at crystal packing interfaces). In both cases, we defined interface residues as those with sidechains having at least one VDW contact to any atom in a different chain. We found that both the overall contact frequencies at interface positions and local (<5 residue) contact frequencies remain similar to the frequencies observed at non-interface positions, but that there is a significant exchange of long-range ($\geq 5$ residue sequence separation) inter-chain to intra-chain contacts (*Figure 4*). This exchange is also observed for the residues in interfaces involved in crystal packing interaction, demonstrating that long-range planar pi interactions are not specific to particular protein folds, but are common features of protein-protein interactions. These results suggest that non-local pi-pi contact propensity could play a general role in mediating protein interactions, including those driving phase separation.

## Importance of pi-pi contacts for phase separation

In our bioinformatics analyses, we identified a type of interaction, planar pi-pi, which is more prevalent for solvated residues, RNA-binding interactions and regions lacking regular secondary structure. These properties are also associated with the emerging functional class of intrinsically disordered phase-separating proteins that coalesce through fluid, multivalent interactions to form protein-dense cellular bodies or membraneless organelles involved in RNA processing (*Mitrea and Kriwacki, 2016*), the nuclear pore (*Frey et al., 2006*) and extracellular biological materials (*Yeo et al., 2011*). The currently known phase-separating proteins are diverse, both in sequence and function (*Mitrea and Kriwacki, 2016*; *Chong and Forman-Kay, 2016*), but many are enriched in motifs we can now associate with high planar pi-pi contact frequencies (i.e. Pro-Gly, Phe-Gly, Ser-Arg, Tyr-Gly and Arg-Gly repeats) (*Mitrea and Kriwacki, 2016*; *Nott et al., 2015*; *Schmidt and Görlich, 2015*).

While phase separation of some proteins has been suggested to be driven by the potential for multivalent aromatic stacking and cation-pi interactions (*Nott et al., 2015*; *Brangwynne et al., 2015*), our observations show (i) that planar pi-pi interactions are a much more broadly distributed phenomenon in proteins than previously considered, especially in solvated protein regions, (ii) that aromatic residues are not required, (iii) that backbone pi groups make significant contributions, and (iv) that protein sequence can have distinct effects on both long-range contact propensity and local contact propensity. These led us to hypothesize that the number of pi orbitals available to make long-range multivalent contacts is an important feature in determining whether a disordered protein region can phase separate and, thus, that the sp$^2$-hybridization of the arginine sidechain is more important to phase separation than its charge. We tested this hypothesis using the N-terminal 236 residues of Ddx4, an intrinsically disordered region that contains both Arg-Gly and Phe-Gly dipeptide sequences and that can phase separate (*Nott et al., 2015*). We removed pi-character while leaving charge intact by replacing all 24 Arg residues with Lys. Matching our expectation, this protein region fails to phase separate under the conditions characterized for the wild-type Ddx4 sequence, even at concentrations of 400 mg/ml, 200 times higher than the lowest concentration for which phase separation is observed for the wild type, and four times higher than observed for

constructs with an equivalent mass change from mutating nine phenylalanine residues to alanine (*Appendix 1—table 4*). We note that arginine is likely key for the phase-separation, association and toxicity of C9orf72, which can encode Gly-Arg and Pro-Arg dipeptide repeat sequences (*Lee et al., 2016*).

## Prediction of phase separation using pi-pi contacts

Given this supportive experimental evidence for the role of pi interactions in phase separation and our observation that opportunistic non-local pi interactions are commonly found at protein crystal contacts, we chose to test the importance of these interactions for phase separation by determining the degree to which it is possible to predict general phase separation behavior using solely the pi-pi contact propensity of a protein sequence. We recognize that multiple physical interactions can contribute to driving phase separation (*Brangwynne et al., 2015*), but our goal was not to predict subtle differences in phase separation propensity or quantitative phase diagrams. Instead, we aimed to merely classify proteins as having the potential to self-associate under particular biological conditions or not, as a test of our hypothesis of the involvement of planar pi interactions. In this exercise, we define phase-separating proteins as those that for presumed functional reasons self-associate in a way that is at least transiently reversible and dynamic, allowing for the protein to self-concentrate as a function of available protein concentration, temperature or other condition. This basic definition does not cover the complexity of the phase diagram, merely the ability to reversibly self-concentrate, and does not consider competing transitions, such as irreversible aggregation and

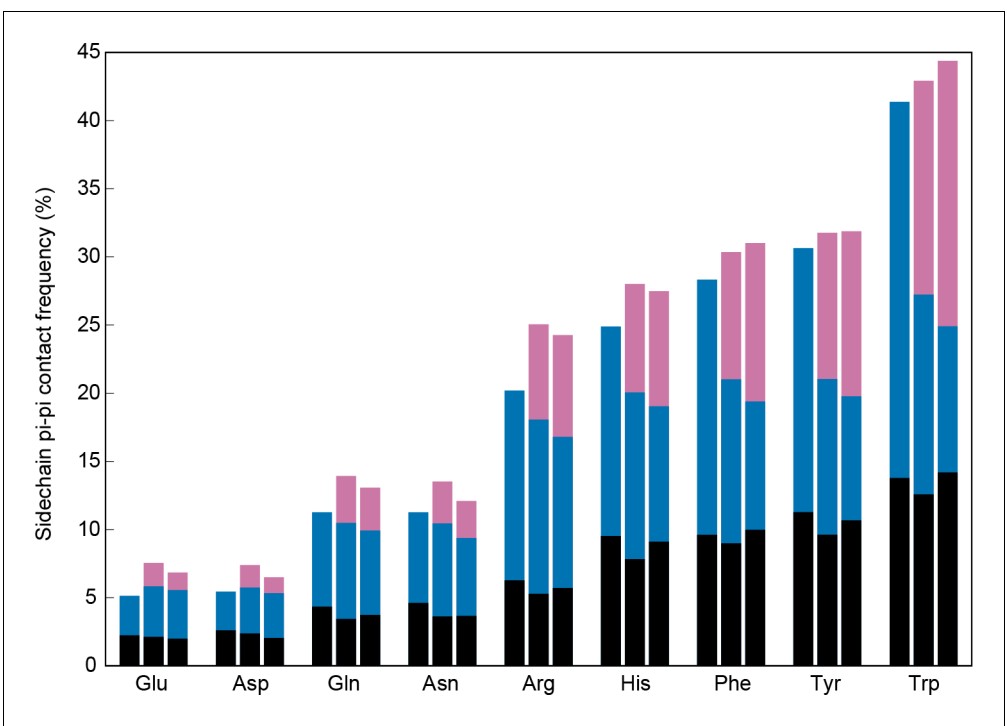

**Figure 4.** Sidechain contacts at interface positions. Contact frequencies are shown for the nine $sp^2$-containing sidechain types, split into three bars based on interface proximity. From left to right, these bars are i) no other chain within 4.9 Å of any sidechain atom, ii) within 4.9 Å VDW contact distance of any atoms in a different chain within the unit cell of the crystal, iii) within 4.9 Å of any atoms in a chain from a neighboring unit cell, as determined by crystal symmetry data. Bars are colored by the proportion of total contacts contributed by three categories, bottom/black corresponding to local (sequence separation ≤4 residues) intrachain contacts, middle/ blue to non-local intrachain contacts, and top/pink to interchain contacts, showing that overall contact frequencies and local contact frequencies remain similar and that the non-local contacts do not discriminate between intra and interchain.

DOI: https://doi.org/10.7554/eLife.31486.011

precipitation, which have typically been selected against in the natural sequences on which the predictor is designed to be used.

Using this definition, we applied a constrained training approach divided into two stages. In the first stage, we required accurate prediction of contact propensities for folded proteins, using sequence propensities for both local and non-local contacts. For this aim, we developed a statistical method for predicting the expected number of contacts given a protein sequence, using frequencies taken from the PDB, splitting observations by distinct residue pairs with varying sequence separation and applying a statistical comparison of the full list of pairs associated with a given $sp^2$ group to calculate expectations (see Materials and methods). The reliability of these predictions against folded proteins is given in *Figure 5A*. We then predicted the number of pi-pi contacts for a list of 11 proteins containing IDRs that have been shown to be sufficient for phase separation behavior in vitro (*Figure 5—source data 1A*), finding that 8 out of the 11 have a predicted number of planar pi-pi contacts per residue in the 99th percentile relative to folded proteins found in the RCSB PDB (*Figure 5B*).

For the second stage, we developed a phase separation predictor that ranks sequences only by the weighted combinations of pi-contact frequency predictions, without any other interaction or observational data. We used a stochastic optimization approach to find optimal weights and sequence window normalizations for converting pi-contact frequency predictions into a score function able to discriminate known phase-separating proteins from sequences found in the PDB. The individual components weighted and normalized include: (i) short- and long-range contacts as defined by residue pair sequence separation $\leq 4$ or $>4$, respectively, (ii) sidechain groups vs. the backbone peptide bond, (iii) absolute predicted frequency vs. normalized frequency compared to the specific group, and (iv) number of carbon atoms in the specific group. In constraining this stage of the test, we defined the fixed goal for optimization as the PDB normalized z-score difference between the highest scoring 1% of the PDB and the lowest scoring member of the phase separation training set. We then trained until reaching a plateau, and at that point we finalized the score, running a single validation test against a testing set of 62 proteins directly associated with phase separation in the literature. This testing set can be divided into three subsets by the nature of the evidence available: (i) sufficient for in vitro phase separation as a purified single component (which matches the training set), (ii) evidence of in vitro phase separation involving a multi-component system (e.g., phase separates on the addition of RNA), without evidence of independent phase separation, and (iii) direct evidence of in cell phase separation (where the protein itself has been labeled and dynamic exchange demonstrated by FRAP or similar methods) without evidence of in vitro phase separation or sufficiency.

We used receiver operating characteristic (ROC) plots comparing predictions of phase-separating proteins within the test set against predictions of phase-separating proteins in the human proteome to assay the ability of the predictor to rank known positives against the members of a set that we assume is primarily negative; the area under the curve (AUC) measurement describes the ability to discriminate between sets. For the human proteome as the negative set, we show an AUC of $0.88 \pm 0.02$ measured using the entire testing set as a positive, $0.93 \pm 0.01$ if we exclude sequences which only phase separate in complex with other polymers, and $0.96 \pm 0.01$ if we restrict to the 32 test set sequences that match the sufficiency criteria used for selecting the training set (*Figure 5C* and *Appendix 1—figure 7A*). These measurements are complicated by the potential for homology between test set and training set proteins. To control for this, we also measured discrimination using another positive set of the 59 artificial sequences designed and shown to phase separate by the Chilkoti lab (*Quiroz and Chilkoti, 2015*; *MacEwan et al., 2017*; *Simon et al., 2017*) (details in *Figure 5—source data 1C*), showing an AUC of $0.86 \pm 0.03$ against the human proteome as a negative set (*Appendix 1—figure 7B*).

Interpreting these AUC values is complicated by the fact that the true positive rate of the human proteome is unknown, and our analysis will treat unknown phase-separating proteins as false positives, inaccurately decreasing the AUC. Similar analysis against protein sets with less expected phase separation results in higher AUCs, going from $0.88 \pm 0.02$ for human to $0.92 \pm 0.01$ for *Caenorhabditis elegans*, $0.93 \pm 0.02$ for *Saccharomyces cerevisiae*, $0.98 \pm 0.01$ for *Escherichia coli*, and $0.97 \pm 0.01$ for our PDB testing set. Within the comparisons to *E. coli* and *S. cerevisiae* proteomes, we show examples of the proteome-dependent score distributions underlying the analysis (*Appendix 1—figure 7C,D*). Using a defined standard confidence threshold of $\geq 4.0$ standard deviations

from the PDB average for the propensity score (PScore) captures 0.3%, 2.2%, and 5.1% of the *E. coli*, *S. cerevisiae*, and human proteome sets, respectively, as compared to 0.1% of our full PDB set and 81% (26/32) of the self-sufficient for in vitro phase separation test set (dropping to 36/62 for the entire proteomic test set and to 35/59 for the synthetic test set).

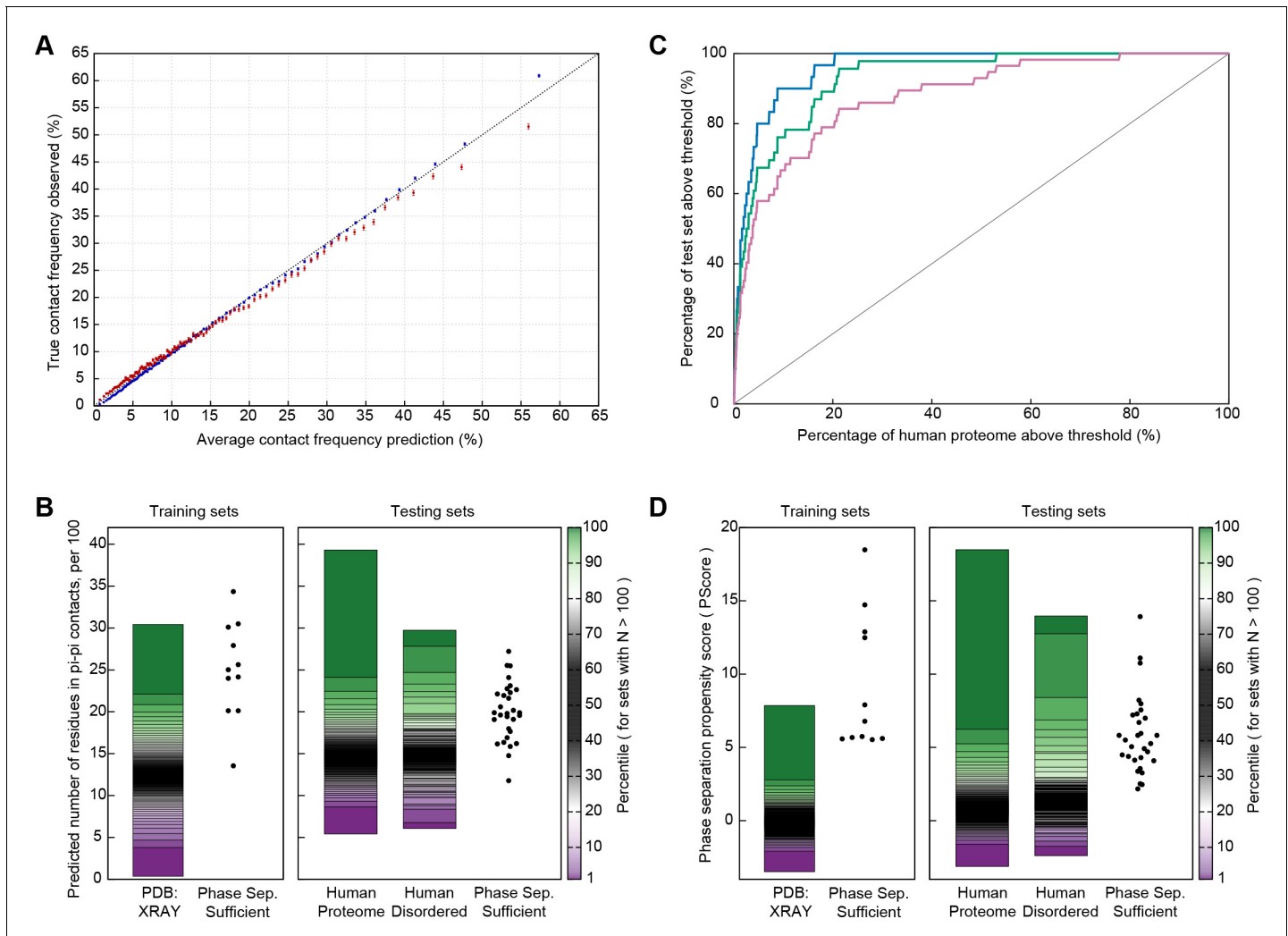

**Figure 5.** Prediction of phase separation based on planar pi-pi interactions. (A) Reliability plot showing average predicted and observed contact frequencies for percentile bins by pi-pi contact prediction for proteins in the PDB, with PDB sequences used for training in blue and the leave out set in red. Bars show SEM. (B) Highest number of contacts predicted, by window, for two phase separation predictor training sets and three test sets, for the unoptimized predictor. (C) Modified ROC curve showing the final predictor's performance on three test sets vs. the human proteome, with the full set in pink (N = 62), the full set minus the insufficient for phase separation set shown in green (N = 44), and the sufficient for phase separation set in blue (N = 32). (D) Results for the final predictor (as for panel b) plotted with the predictor's phase separation propensity scores (PScore). Data underlying B-D included in *Figure 5—source data 1* and *Figure 5—source data 2*.

DOI: https://doi.org/10.7554/eLife.31486.012

The following source data and figure supplements are available for figure 5:

**Source data 1.** Phase separation training, testing and designed protein test sets.
DOI: https://doi.org/10.7554/eLife.31486.015

**Source data 2.** Additional phase separation propensity scores used in final ROC analysis.
DOI: https://doi.org/10.7554/eLife.31486.016

**Figure supplement 1.** Contrasting behavior of disorder prediction algorithms and the phase separation prediction.
DOI: https://doi.org/10.7554/eLife.31486.013

**Figure supplement 2.** Comparison of scores used in generating phase separation predictions.
DOI: https://doi.org/10.7554/eLife.31486.014

When compared to the unweighted pi-contact predictions, the trained PScore confirms the training results, with the number of test set proteins that fall within or above the top 1% range of the PDB increasing from 11/30 to 29/30 (*Figure 5D*). This increase is matched by an increase in the percentage of human proteins in the same range, from 2.3% to 13.1%. Even though the score is trained for discrimination against folded proteins, we do not see a systematic increase in the scores of all disordered human proteins. Comparison against a top performing sequence homology-based disorder predictor (Disopred3, [*Jones and Cozzetto, 2015*]) and a physics-based disorder predictor (IUPRED-Long [*Dosztányi et al., 2005a*]) shows that disorder predictors are better at discriminating disordered proteins from the PDB and the human proteome, while the PScore is consistently better at identifying phase-separating proteins (*Appendix 1—table 5*). The majority of the proteins in our phase separation test set show disordered character, and the analysis shows that, while PScore does correlate with disorder, it only highlights a subset of disordered proteins and does not reflect a general disorder prediction (*Figure 5—figure supplement 1*). As a direct test of this discrimination, we find that using the subset of human proteins with known intrinsic disorder (*Piovesan et al., 2017*) as the phase separation negative set shows similar results as using the human proteome as the negative, at AUC:0.84 ± 0.03 for the full test set and AUC:0.93 ± 0.02 for the in vitro sufficient set.

We note that the optimization methodology used for developing our predictor, specifically training for discrimination against the PDB, was intended to exclude phase separation involving multivalent binding properties of folded proteins with multiple binding surfaces (*Pierce et al., 2016*; *Marzahn et al., 2016*) or multiple folded modular binding domains that interact with multiple linear sequence motifs (*Li et al., 2012*; *Banjade and Rosen, 2014*). Thus, we expect and find a lower success rate for prediction of phase separation of proteins using these mechanisms. We also note that the goal of the prediction experiment is to see whether observed phase separation can be predicted exclusively from contact probabilities as a test of the hypothesis that pi interactions are important for phase separation, but that our method uses probabilities found in the PDB, was trained on natural sequences, and was tested using sequences that are either found in nature or were designed based on sequences that are. The ability to predict contacts is expected to decrease for sequences not observed in nature and for sequences relying to a greater degree on other energetic contributions.

## Mechanistic implications of the optimized phase separation predictor

In order to identify the contact features that play the largest role in the optimized predictor, we did a retrospective analysis testing the predictive power of different scoring algorithms produced during the training process, and explored potential mechanistic implications by testing the power of individual score components, grouping contact predictions into long-range vs. short-range and backbone vs. sidechain (*Appendix 1—table 6*). Our analysis shows that, while training did improve the predictor, a comparable result can be obtained by using only the long-range contact rate predictions for the peptide backbone (*Figure 5—figure supplement 2*, as further described in Appendix 1). This property significantly upweights the role of residues, especially Gly and Pro, that are associated with high overall backbone pi-pi contact frequencies and with lower short-range contact frequencies for local sidechain groups, and is especially important for predicting elastin-like proteins, which often have very few $sp^2$-containing sidechains. Thus, these results highlight the increased availability of $sp^2$ groups for non-local pi-interactions as a key driving force behind the phase separation predictions and is consistent with highly multivalent weak interactions leading to phase separation, both in nonpolar structural proteins like elastin and highly charged RNA-binding proteins like FUS or Ddx4.

Many high contact frequency residue types are also associated with disordered proteins in general, so to control for that potential role we took a selection of 3501 human proteins predicted to have long disordered regions (as described in the methods), split them by PScore into high (PScore ≥4) and low (PScore <1) subsets, and compared the sequence characteristics distinguishing high PScore and low PScore sequences (*Appendix 1—figure 8A*). We find that non-phase-separating intrinsically disordered proteins are actually depleted in Gly and Pro, especially relative to the enrichment seen in phase-separating sequences and sequences predicted to phase separate. Conversely, they are most enriched in Lys, which on average is depleted in phase-separating sequences.

While the division of the predictor into two distinct protocols was used to avoid scores that simply describe sequence similarity to the training set, it is still possible that the training process picked up on specific sequence features in the training set. To explore the contribution of sequence

similarity to the score, we made a measurement of sequence profile similarity based on dipeptide composition (neighboring residue pair frequencies). We compared the high scoring regions selected by the predictor to each of the sequences used in the training set (*Appendix 1—table 7*, see Materials and methods). This analysis, shown in *Appendix 1—figure 8B*, finds that high scoring (PScore ≥4.0) human proteins are, on average, more similar to the training set than are human proteins in general, but that the majority fall within the normal range. Comparison to a set of 1000 BLAST-level sequence homologs of the training set suggests that the majority of the similarity is compositional preference, not homology.

Both sequence similarity and compositional behavior can also be related to the bias toward disorder regions observed in phase-separating proteins. To characterize this, we again took the high and low PScore subsets of our set of 3501 human proteins predicted to have long disordered regions and then compared their sequence profiles. It has previously been observed that disordered proteins have a Shannon entropy (a measurement of sequence complexity) that is lower, but significantly overlapping with ordered proteins (*Romero et al., 2001*). We find here that the high PScore set has a Shannon entropy that is far lower than the range seen for low PScore disordered proteins, which have Shannon entropies that fall in the range observed for folded proteins (*Appendix 1—figure 8C*). Comparing our phase separation test set with the human disprot set we can confirm that this bias toward lower complexity sequences is observed in known phase-separating sequences.

## Analysis and validation of predictions of phase separation

Given the favorable characteristics of our predictor, we investigated correlations of phase separation scores with protein interactions, various biological mechanisms that may regulate phase separation and gene ontology (GO) terms. The principle of sequences with high propensity for non-local pi-pi contact being more likely to self-associate implies that different proteins with high phase separation propensity scores would be more likely to interact with one another. By comparing score pairs from protein interactions taken from the I2D metadatabase (*Niu et al., 2010*), we confirm that high-scoring proteins and low-scoring proteins are both over two-fold more likely to interact with proteins of similar score, relative to expectations (*Figure 6A*). This holds true even when comparing interactions between largely hydrophobic or cytoskeletal proteins (such as elastin and collagen) and highly polar RNA-binding proteins (like Ddx4 and FUS).

This like-score interaction propensity is predicted by a model of phase separation in which multi-valent but individually low-affinity interactions between proteins of similar character coordinate the formation of large, dynamic complexes. To test this aspect of the score, we looked at large complex formation and interaction propensity by examining the background 'contamination' rates observed in affinity purification coupled with mass spectrometry (AP-MS). Large complex formation is measured by the number of negative control experiments in which each human protein appears, over a set of 411 experiments involving non-specific affinity purification steps performed without the specific affinity tag (*Mellacheruvu et al., 2013*). Within this dataset, we observe that 26/28 of our known human phase-separating proteins show up as a contaminant in at least one experiment (O/E = 3.51), and 17/28 show up more than 10% of the time (O/E = 14.9), confirming that phase-separating proteins show the expected behavior. By binning proteins by prediction scores, we show that this is also a trend for high PScore proteins in general (*Figure 6B*), suggesting that the pi contacts driving this score may play a general role in localizing proteins to large complexes.

Phase separation behavior could potentially be modulated by the addition, modification, or removal of even small segments with high phase separation propensity, leading to regulation of phase separation by alternative splicing and post-translational modification (*Romero et al., 2006*; *Hegyi et al., 2011*). To test the possible regulation by splicing, we ran our predictor against human sequences in the UniProtKB/Swiss-Prot (*Magrane et al., 2011*) variable splicing database. We found that 40 ± 2% of included proteins strongly predicted to phase separate (PScore ≥4) have alternative splice variants which either remove the prediction or significantly change the score (ΔPScore >1), often having multiple splice variants spanning a wide range of scores (*Figure 6C*). By comparison, an overall rate of significant changes in score (ΔPScore >1) of 23.0 ± 0.4% is observed for all proteins in the set.

To examine post-translational modifications (PTMs), we analyzed our scores against the database of known PTMs curated by PhosphoSitePlus (*Hornbeck et al., 2012*). We tested the relationship between predicted propensities and number of PTM sites, controlling for protein length by taking

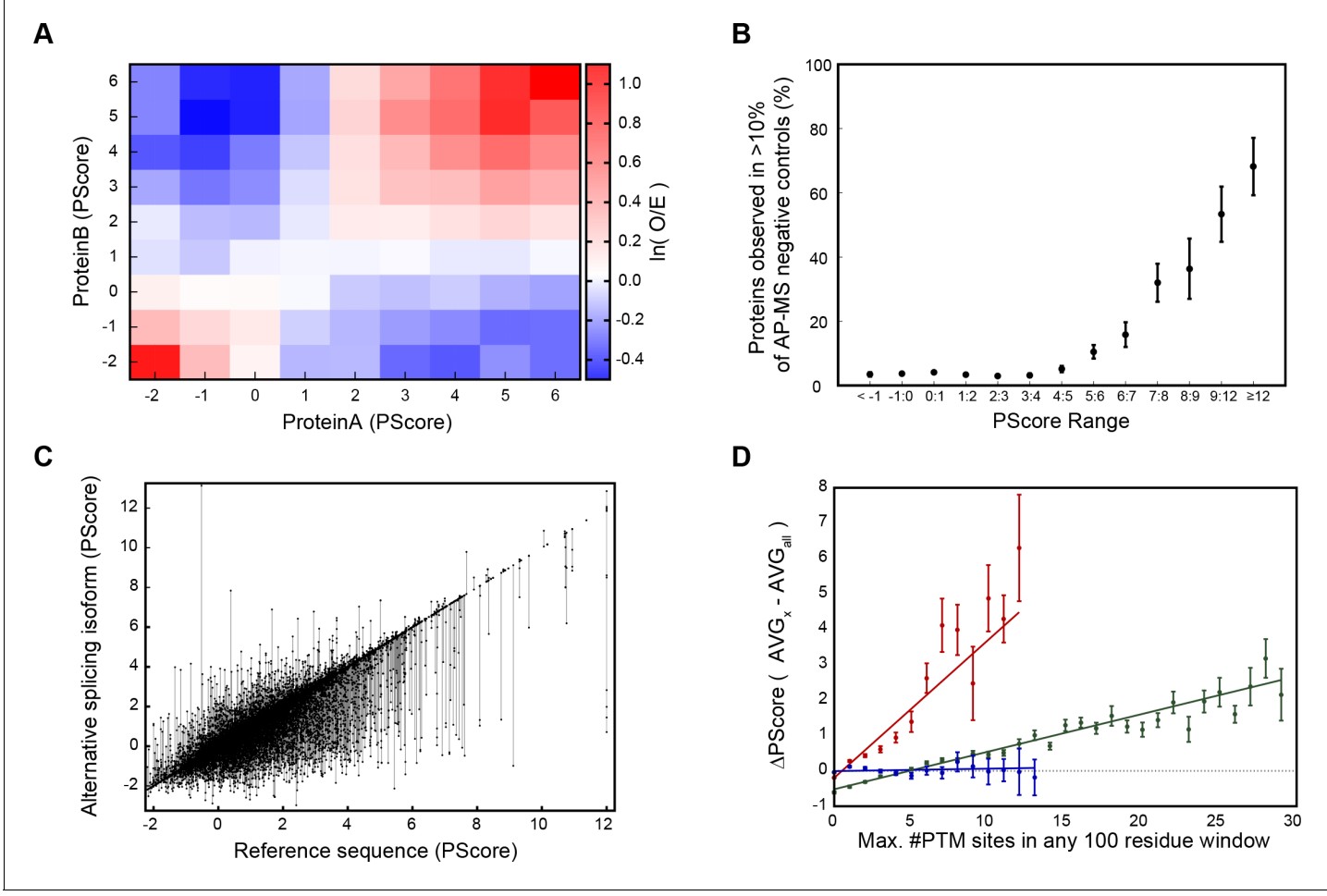

**Figure 6.** Association of phase separation propensity scores with protein interactions, splice isoforms, PTMs, and GO localization, process, and function terms. (A) Protein-protein interaction enrichment by the PScore of partner 1 vs. the PScore of partner 2. The color gradient shows the natural logarithm of the observed over expected ratio. (B) Percentage of human proteins at each PScore range that are detected in more than 10% of AP-MS negative control experiments. (C), Score ranges for alternative splicing variants shown as vertical lines sorted by reference sequence values. (D), Number of PTMs vs. average relative PScore, with methylation shown in red, phosphorylation in green, and ubiquitination in blue.

DOI: https://doi.org/10.7554/eLife.31486.017

PTM counts from the maximum number of annotations observed for any 100 residue window in a sequence. By comparing populations with an above average number of sites (greater than the average plus one standard deviation) against the baseline frequency, we see enrichment in high PScores ($\geq 4$) for a variety of PTM site annotations, including literature annotations of disease relevance and known regulatory function (*Appendix 1—table 8*). We also observe that for phosphorylation and methylation the absolute number of PTMs correlates with the average PScores observed, with methylation having a stronger effect than phosphorylation, and ubiquitination shown as a negative control (*Figure 6D*).

Next, we compared our phase separation predictions to known localization or function, as annotated in the GO database (*Figure 7A,B,C*). Ranking GO terms by enrichment of proteins with prediction values above our threshold (PScore $\geq 4$) enabled us to generate a list of terms associated with significant enrichment of pi-pi contacts (p<0.000001 and 5–50 fold observed over expected); this list includes 4.1% of the 27342 GO terms tested. This subset of the GO database demonstrates enrichment for phase separation propensity in known phase-separated compartments (stress granules, Cajal bodies, post-synaptic density [*Zeng et al., 2016*]), in RNA processing (transcription, splicing, modification, transport, and stability), in the assembly and plasticity of structural components (cytoskeletal organization, extracellular matrix assembly), and in signaling, regulation, and development

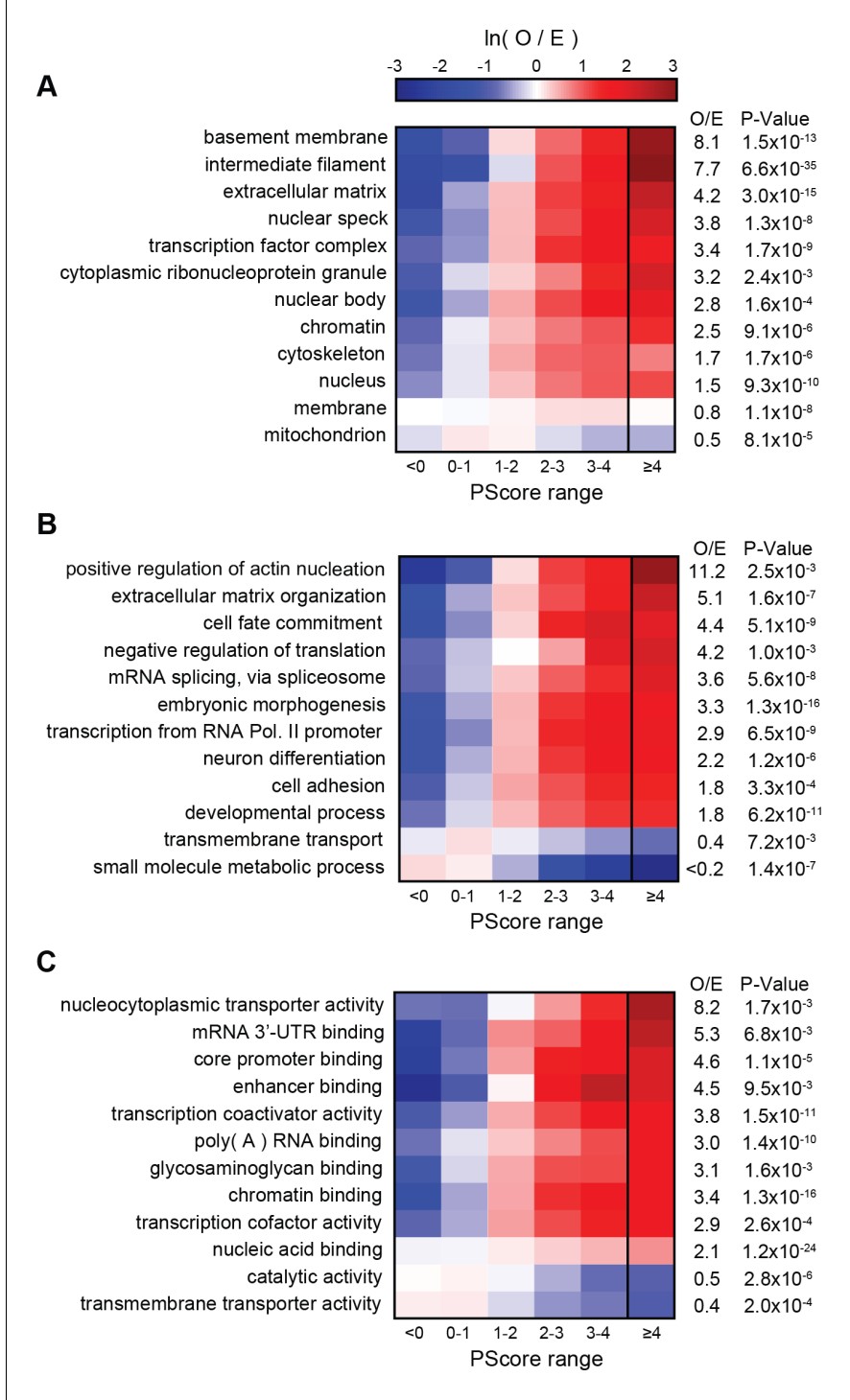

**Figure 7.** PScore enrichment by gene ontology annotation for subcellular localization (**A**), biological process (**B**), and molecular function (**C**). The color gradient shows the natural logarithm of the observed over expected ratio. Heatmaps show enrichment in vertebrate sequences across six defined score ranges, with the highest score range (PScore ≥4) labeled with human enrichment values calculated using PANTHER (see Materials and methods).
DOI: https://doi.org/10.7554/eLife.31486.018

(Notch signaling, NF-κB, Wnt). We note that the sequence property predicted here is a physical behavior that occurs on a cellular scale, so the observation of a similar score distribution for a

specific biological process, as observed for annotations involving localization to known phase-separating bodies, is an implicit prediction that phase separation is one of the physical mechanisms involved in the process. Consistent with that, we see similar score distributions for many processes involving organization of structural components, signaling, and cell-fate commitment. The property of phase separation is also strongly associated with the regulation and development of multicellular cooperation and neurogenesis. In contrast, the vast majority of GO terms (77.2%) show no enrichment in phase separation propensity, with significantly lower enrichment in categories involving metabolic processes and enzymatic catalysis. A selection of high-scoring human proteins associated with enriched functions is shown with per-residue scores and PTM annotations in *Appendix 1—figure 9*, with examples chosen from the highest scoring protein in any given gene ontology function/localization annotation related to neuronal plasticity or behavior in A, cytoskeletal biomaterials in B, signaling in C, and extracellular biomaterials in D.

Within the testing set, there exist some proteins which have not been shown to be capable of independent phase separation (*Han et al., 2012*), and which may associate with phase-separated bodies without sharing the same behavior. One of these, synaptic functional regulator FMR1 (*El Fatimy et al., 2016*), also known as fragile X syndrome protein FMRP, has a PScore of 4.7, and is involved in RNA binding, neurological development and regulation of translation, all GO terms enriched in high PScores. FMR1 is a multifunctional polyribosome-associated protein, which is highly expressed in the brain and in the testes, and is known to localize to granular bodies with two other proteins (FXR1 and FXR2) (*El Fatimy et al., 2016*) that are also predicted with high PScores (at 2.9 and 5.3). In order to validate that high PScore predicts sufficiency for phase separation and not associated properties like miscibility in the separated phases of other proteins, or other interactions with phase-separating proteins, we purified the highest scoring region (residues 445–632) and confirmed the ability to spontaneously undergo liquid phase separation at low temperature and high concentrations in physiological buffer conditions (*Figure 8A*). The concentration required for visual confirmation of liquid phase separation behavior is quite high, at 1 mM FMRP-LCR, but can be reduced through the use of crowding reagents (*Figure 8—figure supplement 1A*). To confirm the relevance of pi-character, we then replaced all 28 Arg residues with Lys, which resulted in a loss of phase separation behavior (*Appendix 1—table 4*).

To test whether or not the predictor is applicable to sequences that do not share motifs or functions with any of our training set proteins, we did a manual search for predictions with sequence properties and functions dissimilar from the training set proteins and selected two proteins, human engrailed-2 (UID: P19622, PScore 5.0), a DNA-binding homeobox protein, and the pAP isoform of the Human cytomegalovirus capsid scaffolding protein (UID: P16753-2, PScore 3.8), a protein that plays an essential structural role in assembling the viral capsid, a novel function relative to those known to involve phase separation. Both sequences have little overlap with any of the sequence motifs found in our training set (*Appendix 1—table 7*), aside from general enrichment in glycine and proline residues. Experimentally, we observe reversible liquid phase separation of pAP protein with increasing temperature, with viscoelastic properties similar to the coacervation of elastins (*Figure 8C*). We did not observe phase separation of engrailed-2 under the same buffer conditions, even at 1 mM protein concentration, but did observe temperature-dependent liquid droplet formation in the presence of a crowding reagent (20 mg/ml ficol) (*Figure 8—figure supplement 1B*). While these observations do not represent a robust or comprehensive test of prediction quality, they do suggest that the predictions provide a useful tool for selecting natural proteins capable of self-sufficient liquid demixing.

## Discussion

We tested the potential role played by pi-contacts in mediating phase separation by using the single property of pi-contact frequency to train a simplistic predictor of phase separation behavior found in natural sequences, finding that the single property of long-range pi-contact propensity is sufficient for marking the majority of known phase-separating proteins as outliers relative to the proteome, supporting the hypothesis that this sequence property is commonly associated with phase-separating proteins. While this association is demonstrably useful for identifying phase-separating proteins in proteomic datasets, these contacts may not be the predominant interaction driving the physical process of phase separation for each case, and could instead reflect a modulatory role since it is not

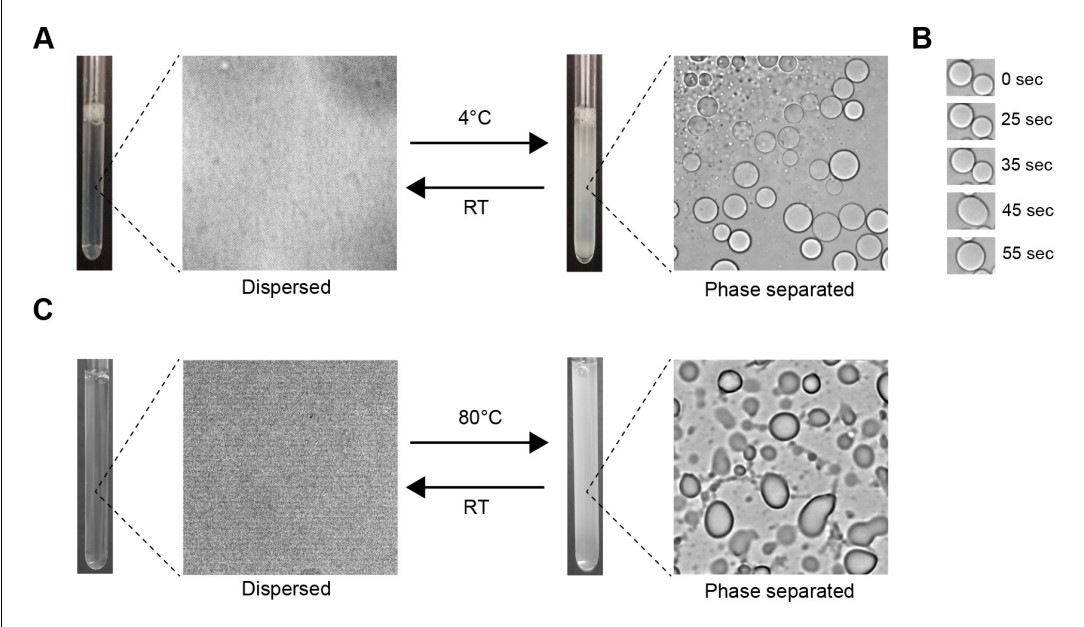

**Figure 8.** Visual confirmation of phase separation. (**A**) Test tubes containing transparent or turbid solutions of 1 mM FMR1 C-terminus (residues 445–632) along with their corresponding DIC microscopy images taken at room temperature or 4°C, respectively. (**B**) 1 mM FMR1 C-terminus forms droplets exhibiting liquid fusion properties at 4°C. (**C**) 40 µM solutions of Human Cytalomegalovirus pAP along with corresponding microscopy images taken at room temperature or 80°C, respectively.

DOI: https://doi.org/10.7554/eLife.31486.019

The following figure supplement is available for figure 8:

**Figure supplement 1.** Visual confirmation of phase separation, using 20 mg/ml ficol as a crowding agent.

DOI: https://doi.org/10.7554/eLife.31486.020

exclusive of other interactions like hydrogen bonds and charge interactions. However, tests showing that arginine to lysine mutations abrogate phase separation behavior do provide evidence of the importance of planar sp$^2$ groups for phase-separating systems.

The finding that a single contact potential can generate a reasonably accurate classifier of phase separation behavior suggests that a sequence-based prediction of phase separation behavior is a tractable problem, and that future development of an algorithm that can predict the complexities of the phase transition, environmental effects and concentration requirements is a reasonable goal. This goal could potentially be addressed by introducing the range of phase separation associated sequence properties that were intentionally excluded by our empirical test of the pi-contact association, including the electrostatic effects of charge patterning (*Nott et al., 2015*; *Lin et al., 2016*; *Das and Pappu, 2013*), multivalency of PTM sites and PTM-binding motifs, and transient structural interactions, including strand formation (*Murray et al., 2017*) and helical interactions (*Conicella et al., 2016*). There is also a role for incorporating predictions of competing states, the irreversible aggregation propensity of a sequence or its amyloidogenic potential. Incorporating annotation data associated with phase-separating proteins could be another avenue for generating a physiological classifier in a more comprehensive predictor.

The physical nature of pi-pi contacts and their underlying mechanistic relationship to phase separation are not revealed by the simple contact frequency measurements used in our predictions. These contacts are observed in folded proteins, both internally and near solvated interfaces and, while that suggests they play a general role in the energetics of protein-protein interactions, the nature of that role is not clear. There is potential for electrostatic or induced dipole and quadrupolar interactions, especially in the context of other dipole interactions and hydrogen bonds, but the flat surfaces of sp$^2$ groups could also enable solvation to drive contacts and lead to entropic contributions due to the relative freedom of movement inherent in packing flat plates, compared to the more rigid shape complementation involved in packing aliphatic groups. It is interesting to note that

these proposed mechanisms could be affected by temperature in opposite ways, and that our predictor using pi-contact frequencies is useful in identifying phase-separating proteins regardless of whether they associate more readily as temperatures decrease (such as Ddx4) (*Nott et al., 2015*) or increase (such as elastin) (*Yeo et al., 2011*).

As part of characterizing the proteomic associations highlighted by our empirical prediction test, we point out that manual inspection of our prediction results across the human proteome suggests that planar pi-contact associated phase separation likely facilitates a wide range of cellular functions. To highlight this, we selected a range of examples by taking the highest scoring member of gene ontology categories we found to be generally enriched in high PScore proteins. We see enrichment of phase separation propensity in proteins associated with cytoskeletal organization. These include proteins with known structural roles such as the cytoskeletal intermediate filament proteins desmin and vimentin (PScores 4.3 and 4.4), as well as keratins 8 and 18 (PScores 5.9 and 5.4), with scores deriving primarily from the disordered head and tail domains (*Appendix 1—figure 9*). Intermediate filaments form through dynamic processes (*Yeo et al., 2011*) consistent with a model in which phase separation-induced condensation concentrates proteins prior to the formation of (often fibrillar) structure (*Yeo et al., 2011*). Interestingly, helical domain mutations impeding structure formation cause these four proteins to instead accumulate in protein-rich membraneless inclusions such as Mallory-Denk bodies (*Strnad et al., 2008*; *Goebel and Bornemann, 1993*; *Ogrodnik et al., 2014*). We also predict high PScores for non-structural proteins involved in regulating cytoskeletal organization and in binding some of the previously mentioned cytoskeletal proteins, including focal adhesion kinase 1 (PScore 4.2) and DNAJB homolog 6 (PScore 8.8), the latter of which is also a chaperone that can prevent huntingtin aggregation (*Chuang et al., 2002*; *Hageman et al., 2010*; *Gillis et al., 2013*).

Many of the high PScore predictions involve proteins that are both involved in signaling pathways and known to either localize to membraneless organelles or interact with phase-separating proteins. For example, adenomatous polyposis coli protein (APC) (PScore 3.2) and axin1 (PScore 2.2), involved in the Wnt signaling pathway, interact in a dynamic fashion in the large and multimeric β-catenin destruction complex (*Pronobis et al., 2015*), and we find high PScores for other critical members of the complex, including β-catenin (PScore 5.5) and GSK3α (PScore 6.4). The β-catenin destruction complex formation is regulated by GSK3 phosphorylation, and we note that the predictor shows a difference between the two human GSK3 orthologs, with GSK3β having a PScore of 2.2. These orthologs are often functionally interchangeable (*Doble et al., 2007*), but there is evidence of isoform-specific roles for GSK3α (*Ma, 2014*) and the predicted differences could reflect a difference in modulating phase-separation behavior.

In conclusion, we have shown that planar pi-pi interactions are more prevalent in protein structures than previously described, with potential roles in structural motifs, catalysis and RNA binding. Planar pi-pi contact frequencies are increased in protein segments that lack regular secondary structure or have increased solvent exposure, pointing to their relevance for disordered protein regions. This, together with the enrichment of pi-containing groups in protein regions known to phase separate, provided an impetus for development of a phase-separation predictor based on the likelihood of forming non-local planar pi-pi contacts. The performance of the predictor supports the hypothesis that these pi-pi interactions can drive phase separation. While experimental data and computational work suggest other contributions (*Yeo et al., 2011*; *Pak et al., 2016*; *Nott et al., 2015*; *Lin et al., 2016*; *Kim et al., 2016*; *Brangwynne et al., 2015*), including the hydrophobic effect, electrostatics and multivalent binding of folded protein domains, our prediction test shows that an algorithm focused solely on pi-interactions performs well for the majority of proteins that we identified as phase-separating from the literature (*Figure 5C*, *Figure 5—source data 1B*). These results strongly suggest that most phase-separating proteins can make significant non-local planar pi-interactions, even in cases where there are other dominant or required forces driving phase separation. Thus, this represents a valuable tool for the currently expanding field of protein phase separation and its link to biological function and disease (*Mitrea and Kriwacki, 2016*; *Conicella et al., 2016*; *Patel et al., 2015*). In particular, the association of neurological diseases with proteins comprising RNA processing bodies (*Vanderweyde et al., 2013*), including those known to phase separate, highlights the importance of predictive methods for facilitating mechanistic studies of the underlying biology and pathology.

# Materials and methods

## Key resources table

| Reagent type (species) or resource | Designation | Source or reference | Identifiers | Additional information |
|---|---|---|---|---|
| Recombinant DNA reagent | His-SUMO-Ddx4 [1-236] | PMID 25747659 | | Expression vector (His-Sumo tagged) for Ddx4 residues 1–236, sequence from UID: Q9NQI0-1 (uniprot identification) |
| Recombinant DNA reagent | His-SUMO-Ddx4 [1-236(9FtoA)] | PMID 25747659 | | Expression vector (His-Sumo tagged) for Ddx4 residues 1–236, sequence from UID: Q9NQI0-1, 9 out of 14 phenylalanines mutated to alanine |
| Recombinant DNA reagent | His-SUMO-Ddx4 [1-236(14FtoA)] | PMID 28894006 | | Expression vector (His-Sumo tagged) for Ddx4 residues 1–236, sequence from UID: Q9NQI0-1, all phenylalanines mutated to alanine |
| Recombinant DNA reagent | His-SUMO-Ddx4 [1-236(RtoK)] | PMID 28894006 | | Expression vector (His-Sumo tagged) for Ddx4 residues 1–236, sequence from UID: Q9NQI0-1, all arginines mutated to lysine |
| Recombinant DNA reagent | His-SUMO-FMR1 [445-632] | This paper | | Expression vector (His-Sumo tagged) for FMR1 residues 445–632, sequence from UID: Q06787-1 |
| Recombinant DNA reagent | His-SUMO-FMR1 [445-632(RtoK)] | This paper | | Expression vector (His-Sumo tagged) for FMR1 residues 445–632, sequence from UID: Q06787-1, all arginines mutated to lysine |
| Recombinant DNA reagent | His-SUMO-pAP [A341Q] | This paper | | Expression vector (His-Sumo tagged) for SCAF isoform pAP, sequence from UID: P16753-2, alanine 341 mutated to glutamine |
| Recombinant DNA reagent | His-SUMO-EN2 | This paper | | Expression vector (His-Sumo tagged) for Engrailed-2, sequence from UID: P19622-1 |

## Analysis of pi-pi interactions

### Structures used for primary analysis

Protein structures determined by X-ray crystallography were downloaded from the PDB based on lists compiled using the Pisces web server (*Wang and Dunbrack, 2003*), May 7 2015, which identified 23074 non-redundant chains based on cutoffs of <60%,<5.0, and <0.5 for sequence identity, resolution, and R-factor, respectively. For calculating statistics, high-resolution structures were defined as a subset of 5718 structures with resolution ≤1.8 and R-factor ≤0.18. For structures determined using distance restraints from nuclear magnetic resonance spectroscopy, we took the full list of 2949 PDBs with distance constraints available from the BMRB (*Ulrich et al., 2008*) database of converted restraints (DOCR) (*Doreleijers et al., 2005*) as of July 3rd, 2015.

### Contact definition

To probe contact geometries we read each set of coordinates into custom python scripts, filtering input data by ignoring $sp^2$ systems that lack any of the expected heavy atoms (<0.1%) and only taking the first set of coordinates when represented by multiple conformations. The $sp^2$ systems were defined by atom names for each of nine sidechain groups (from W,F,Y,H,R,Q,N,E,D), the backbone peptide bond, and the C-terminal carboxyl group. Planar axes were defined as normal vectors by using the cross product method against defined lists of three sequential atoms. VDW contacts between $sp^2$ groups were determined by the full set of heavy atom (C, N, O) distance measurements, using a threshold of ≤4.9 Å to define contacts. This represents the upper range of VDW contacts between $sp^2$ groups (*Appendix 1—figure 1E*), because we intended to compare contact frequencies by data resolution and did not want to introduce arbitrary energetic cutoffs for atoms with potentially unreliable positions.

In analyzing the planar orientations of $sp^2$ groups found in VDW contact, we found enrichment of in-plane contacts, predominantly face-to-face, so we devised a simple system for identifying them that can be generalized across groups with variable numbers of atoms. These planar surface contacts were defined first by requiring at least two different pairs of atoms to be in VDW contact. Contact distances were further restricted by requiring that surfaces 1.7 Å above the $sp^2$ plane be ≤1.5 Å

apart (as shown in *Appendix 1—figure 1B*). This planar-surface distance requirement is used to ensure contacts that put the pi-orbitals in proximity to one another, and we note that while this threshold will accept atom-atom contacts as far as 4.9 Å apart the majority end up below 4.0 Å (*Appendix 1—figure 1E*, in purple). To restrict contacts to planar contacts, the dot product of the planar normal vectors were required to have an absolute value $\geq$0.8 (equivalent to an orientation difference from 0° to ~37°). This threshold retains >80% of the contacts identified by distance, and was chosen because interactions between planar groups show a noticeable enrichment relative to random orientation in this range (*Appendix 1—figure 1D*).

Annotation data for the full non-redundant set of PDBs analyzed are included in files *Figure 1—source data 1* and *2*, and scripts for creating contact annotations from a PDB are included in supplemental file *Source code 1*.

## Planar pi-pi contact frequency
Comprehensive lists of planar pi-pi contacts were computed for each chain and were stored in a database. Contact frequencies were calculated as the total number of observed contacts divided by the total number of residues considered. Residues were counted only for each non-redundant chain, and contacts include both the ones made within that chain and the ones that chain makes to any other chains present in the PDB (except when noted otherwise). Contacts to crystal symmetry partners were also measured but were kept separate and, except where specifically investigated to probe inter-chain contacts, were excluded from analysis based on the observation that VDW contacts made to symmetry partners can contain a small (<1%) population of extreme clashes (atoms <1 Å from one another).

## Pi-contact frequency vs. resolution
PDBs were sorted into 77 non-overlapping bins first by exact resolution and then by rolling any bins with less than 100 PDBs into the next acceptable bin within 0.25 Å. This method rounds up the small populations of resolution values while retaining as much resolution information as possible. Correlation values and lines of best fit were estimated using linear regression (inherited from the scipy python package, version 0.12.1) against bin averages, with bins weighted by sample size.

## Involvement of amino acid types in planar pi-pi contacts
In order to compare frequency of contacts involving all 20 common amino acids, we defined involvement based on the participation of any atom from that residue in a planar contact. For most sequence positions, this means at least one contact made to either one of the flanking peptide bonds or, for the nine amino acids that have them, the sidechain group. By this definition, backbone planar contacts involve both flanking residues.

## Sp$^3$ controls
To provide a prior expectation control for enrichment of planar pi-pi contacts, we took the terminal heavy-atom planar surfaces from fully sp$^3$ hybridized sidechains, using the following PDB atom names to define each planar group: Leucine: CD1, CG, CD2; Valine: CG1, CB, CG2; Methionine: CE, SD, CG; Isoleucine: CD1, CG1, CB; Cysteine: SG, CB, CA; Serine: OG, CB, CA; Threonine: OG1, CB, CG2; and Lysine: NZ, CE, CD.

## Small molecule datasets
The PDB was screened for crystallographic structures containing either amino acids or other small molecules as free ligands, with the other small molecules being restricted to those that (1) are present in more than 100 structures, (2) have a single sp$^2$ group, and (3) have all heavy atoms (C,N,O) falling within the sp$^2$ plane. These structures were then filtered for resolution ($\leq$3.0 Å) and redundancy ($\leq$90% identity) by using the Pisces web server. Contact frequencies were determined across the full list of ligands in these non-redundant sets, with contacts to amino acid ligands being divided into backbone carboxyl and sidechain sp$^2$ groups. As an internal control for amino acid contact frequencies, contact frequencies were determined for each amino acid based on the same set of structures used to define the ligand frequency. For sidechain groups, the controls are their direct

equivalents found within the protein, and for the amino carboxyl groups, we used the protein C-terminal carboxyl groups as the control. Population statistics are summarized in *Appendix 1—table 2*.

## Catalytic sites

We defined catalytic sites based on direct literature annotation as described in the Catalytic Site Atlas (*Furnham et al., 2014*), with 2914 residue positions identified over 928 protein structures. The full population of residues across the annotated chains was split into 40 bins according to identity and annotation status. Relative contact involvement frequencies, catalytic vs. non-catalytic, were obtained for each amino acid type. For each catalytic residue, we then identified the total number of VDW contacts made to any other residue, identified which VDW contacts fall into the subset defined by our pi-contact rules, and then, for the 2377 catalytic residues with at least one VDW contact to another catalytic residue, we computed the frequency of VDW contacts that are also pi-contacts.

## External measurements and secondary structure

Hydrogen bond data were calculated using PHENIX (*Adams et al., 2010*), with amino sidechains allowed to flip 180 deg to maximize the number of donor/acceptor pairs. DSSP (*Kabsch and Sander, 1983*) was used to define backbone secondary structures. Water contacts were defined by direct distance measurements, with the full set of water molecules, including symmetry partners, extracted using the SYMPEXP function from pymol (*Schrodinger LLC, 2015*). NMR restraints were obtained for 2949 structures from the Database Of Converted Restraints (*Doreleijers et al., 2005*). For defining short secondary structure motifs, we used the simplified one letter definitions provided by DSSP ('H', 'B', 'E', 'G', 'I', 'T', 'S', and ''), in order to maintain adequate sample size when comparing enrichment across motifs. For comparing ordered and disordered residues, clear helices and strands ('H' for $\alpha$-helix, 'G' for $3_{10}$ helix, and 'E' for $\beta$-strand) were defined as the ordered assignments, representing regular secondary structure.

## Predictor training and bioinformatics

### Pi-contact prediction for structures in the PDB

We trained a statistical potential for predicting pi-contact frequency from protein sequence for individual $sp^2$ groups, with contacts split by sequence separation into short-range ($\leq$4 residues apart) and long-range ($\geq$5 residues apart, or different chains). We trained against an 80% random cut of the 17388 proteins in our non-redundant crystal structure subset of the PDB, leaving the remaining 20% as a testing set for a single final test of the predictor. The final predictor, covered in detail in Appendix 2, operates by first averaging the frequencies observed for $sp^2$ groups found in specific sequence contexts (with context defined as all residues within 40 amino acids of a given residue) and then comparing the average values to the distributions observed for $sp^2$ groups with the same sequence identity (with nine sequence identities for sidechain $sp^2$ systems, and 400 distinct identities for the peptide backbone), where the final prediction is the contact frequency observed at matching positions in the PDB.

### Phase separation prediction benchmark

To develop a predictor for the phase separation propensity of a given protein sequence, we started by defining a set of 11 proteins which have been shown, in the literature, to phase separate in vitro as single purified components due to interactions involving intrinsically disordered regions of the protein. We also defined a leave out set of 62 proteins associated with phase separation in the literature by combining proteins matching three distinct criteria: 'in vitro sufficient' (N = 32), proteins satisfying the criteria used to select the test set, 'in vitro insufficient' (N = 18), proteins for which the literature contains evidence of in vitro phase separation in complex with other proteins or with RNA but phase separation as a single component not observed and '*in cellulo* associated' (N = 12), proteins without in vitro characterization, but with evidence of phase separation in live cells, as determined both by localization to a known dynamic protein body and by a direct measurement of dynamic character, typically involving FRAP recovery of a fluorescent tag. These benchmark sets are included in supplemental file *Figure 5—source data 1A,B and C*.

Over these datasets, we only found eight proteins less than 300 residues in length, with the smallest protein sequence observed (RBM3_HUMAN) being 157 residues long. To avoid extrapolating

our predictions onto an unobserved class of proteins, we decided to restrict testing to sequences ≥140 residues in length. To define additional control and training sets, we applied this sequence cutoff to a series of datasets, including the PDB sets used for developing the pi-contact predictor, with 13388/17388 training set and 3406/4347 test set sequences retained after restricting by length ≥140, the UniProt human reference proteome (September 2016, 18582/21047 sequences used), and the subset of the human proteome with known disorder, as defined by the DISPROT database (*Piovesan et al., 2017*) (205/249 sequences).

## Phase separation predictor training

The phase separation propensity predictor starts by inheriting a table of 8 pi-contact prediction values per $sp^2$ group in the sequence, splitting contacts by i) short-range (≤4 sequence separation) vs. long-range (>4), sidechain vs. backbone, and absolute predicted frequency vs. relative difference from $sp^2$ groups with the same identity (with nine sidechain $sp^2$ groups and 400 backbone groups, split by their associated sequence). Sequences are then scored by a series of weighted sequence window averages. Weights, window length, and normalization parameters were refined using a stochastic optimization process to maximize the score difference between the lowest scoring member of our 11 member training set, and the average score of the highest scoring 1% of the PDB training set. A full training history and details of the final predictor are described in Appendix 2. AUC values at different stages of training the predictor are tabulated in *Appendix 1—table 6*. Standard error of the mean values for AUC calculations were estimated by bootstrap using sampling with replacement (10,000 iterations) against both the test and human sets.

The final predictor consists of a single python script and associated database files, with the state at time of submission included in supplemental file *Source code 1*.

## Proteome analysis

Phase separation scores for analyzed proteins were considered with respect to known interactions, and functional annotations, using the gene ontology database (release Oct-04–2016), UniProt's Swiss-Prot and TREMBL sequence databases, including the reference proteome annotations, vertebrate protein sequence list, and variant splicing data (release 11-May-2016) (*Magrane et al., 2011*; *Pundir et al., 2017*; *Huntley et al., 2015*; *Suzek et al., 2007*), PTM data from PhosphoSitePlus (release Dec-16–2011) (*Hornbeck et al., 2015*), human protein-protein interactions collated under the Interologous Interactions Database (I2D) (version 2.9) (*Brown and Jurisica, 2007*), and background AP-MS detection rates from the Contaminant Repository for Affinity Purification Mass Spectrometry Database (CRAPome version 1.1) (*Mellacheruvu et al., 2013*).

GO term enrichment data over the full range of propensity scores were analyzed against all proteins with UniProt codes contained within both the vertebrate reference sequences and the gene ontology database. Enrichment scores and p-values for individual GO terms were obtained for a defined four sigma cut against the human proteome by using PANTHER (*Thomas et al., 2003*; *Mi et al., 2017*; *Mi et al., 2016*) analysis.

## Disorder prediction

Per residue disorder predictions were obtained using Disopred3.16 (*Jones and Cozzetto, 2015*) (standard command line and Refseq database) and IUPRED-Long (*Dosztányi et al., 2005a*; *Dosztányi et al., 2005b*) against the phase separation test and training sets, the PDB test set, the human Disprot set, and a random selection of 7397 sequences from the human proteome. To convert these into per-sequence scores for comparison to the PScore, we then used the optimized window averaging method developed during training of the predictor, where the window is defined as all residues within five sequence positions of the highest scoring sixty. These scores can be used to classify whether or not a given sequence has a number of disordered residues comparable to the length of a folded domain, either concentrated in a single large region or distributed throughout the sequence.

## Sequence analysis

Sequence similarity to the proteins within the training set was measured by computing dipeptide sequence profiles (the frequencies of all 400 possible i,i + 1 amino acid combinations in a sequence),

calculating block L1 distances between a query dipeptide profile and each of the training set pro-files, and then returning the lowest observed distance. Comparison to the sequence similarity of direct homologs was observed against a set of 1100 sequences obtained via BLAST by using the phase separation training set sequences used as queries against the seq database (E = 0.0000001).

Shannon entropy values were calculated for amino acid profiles of sequences by the standard equation (*Shannon, 2001*), and comparisons of high and low scoring disordered proteins were obtained from the subset of human sequences with Disopred3 predictions > 0.80, using the window averaging method described previously (N = 3501 out of 7397 sequences). High and low PScore sets were defined by PScore $\geq$4.0 (N = 310) and PScore <1.0 (N = 1044), corresponding to our stan-dard phase separation confidence threshold and scores less than one standard deviation above the PDB average, respectively.

## Experimental methods

### Protein expression and purification

Ddx4: Constructs for Ddx4 $^{1-236}$ wild type sequence (UID: Q9NQI0-1) and mutants were synthesized and subcloned into a pET Sumo vector (Genscript) to produce His-SUMO-Ddx4$^{1-236}$ (*Nott et al., 2015*), His-SUMO-Ddx4$^{1-236(9FtoA)}$ (*Nott et al., 2015*), His-SUMO-Ddx4$^{1-236(14FtoA)}$ (*Brady et al., 2017*), and His-SUMO-Ddx4$^{1-236(RtoK)}$ (*Brady et al., 2017*). Protein was overexpressed in *E. coli* and purified as described previously (*Nott et al., 2015*). Phase separation was induced at 24°C by dialy-sis of a high concentration of Ddx4 in 20 mM Na$_2$PO$_4$, 1 M NaCl, 5 mM TCEP, pH 6.5 into a buffer containing 20 mM Na$_2$PO$_4$, 100 mM NaCl, 5 mM TCEP, pH 6.5. Concentrations were measured by spectrophotometry, using an extinction coefficient of 23950 M$^{-1}$cm$^{-1}$ at 280 nm.

FMR1: His-SUMO-FMR1$^{445-632}$ (FMR1 from UID: Q06787-1) and His-SUMO-FMR1$^{445-632(RtoK)}$ were transformed into *E. coli* BL21-CodonPlus(DE3) RIL cells. Bacteria were grown in Luria Both at 37°C and protein expression was induced with 0.5 mM IPTG at OD$_{600\ nm}$ of ~0.6–0.8, followed by over-night growth at 24°C. Cells were harvested by centrifugation and pellets were stored at −20°C. Pro-tein pellets were re-suspended in lysis buffer containing 50 mM NaPO$_4$ pH 8.0, 6 M guanidinium chloride (GdmCl), 500 mM NaCl, 20 mM imidazole and 2 mM DTT. Cells were then lysed via sonica-tion and lysates were cleared by centrifugation at 39,000 g for 30 mins at 4°C. The supernatant was loaded onto a HisTrap column equilibrated with the lysis buffer followed by extensive washing with the same buffer (10 CV). The GdmCl was removed by washing the column with buffer containing 50 mM Na$_2$PO$_4$ pH 8.0, 500 mM NaCl, 20 mM imidazole and 2 mM DTT (10 CV). The protein was then eluted in the same buffer supplemented with 300 mM imidazole. The His-SUMO tag was cleaved with the SUMO protease, Ulp, while dialyzing against 50 mM NaPO$_4$ pH 8.0, 500 mM NaCl, 20 mM imidazole, and 10 mM DTT at 4°C over night. The dialysate was loaded again onto a HisTrap column equilibrated with dialysis buffer to separate the His-SUMO tag and the His-tagged Ulp from the FMR1$^{445-632}$ protein. All fractions were analyzed by SDS-PAGE, and fractions containing the protein of interest were combined and concentrated with ultrafiltration. Concentrated samples were passed over a Superdex 75 gel filtration column into a final buffer of 50 mM NaPO$_4$ pH 8.0, 2 M GdmCl, 200 mM NaCl, and 2 mM DTT. Protein identity was confirmed by mass spectrometry and frozen at −80°C until use. Concentrations were determined from the absorbance at 280 nm using a molar extinction coefficient of 9970 M$^{-1}$cm$^{-1}$.

pAP: His-SUMO-pAP$^{A341Q}$ (SCAF Isoform pAP from UID: P16753-2, with a single mutation A341Q added to confer protease resistance (*Brignole and Gibson, 2007*)), was transformed, grown, induced, and purified following the protocol for FMR1, but with growth post-induction done for 4 hr at 37°C, and with an additional Superdex 75 gel filtration step added between the first HisTrap step and the Ulp cleavage step. Concentrations were determined from the absorbance at 280 nm using a molar extinction coefficient of 35870 M$^{-1}$cm$^{-1}$.

Engrailed-2: The expression and purification steps of engrailed-2 (UID: P19622) from His-SUMO-EN2 were similar to the protocols used for FMR1, but with growth post-induction done for 4 hr at 25°C, and addition of a HiTrap SP XL (GE Healthcare) ion exchange chromatography step between the Ulp cleavage step and a Superdex 75 gel filtration step for removing His-SUMO by increasing NaCl concentration from 50 to 1000 mM in 50 mM NaPO$_4$, 2 mM DTT at pH 7.4. Concentrations were determined from the absorbance at 280 nm using a molar extinction coefficient of 22460 M-1cm-1.

## Phase separation test and differential interference contrast imagining

For *Figure 8A*, concentrated FMR1 protein samples were dialyzed against 20 mM $NaPO_4$ pH 7.4 and 2 mM DTT overnight at 4°C and then diluted with the same buffer to the desired protein concentrations for imaging. Samples were incubated on ice for 5 min before placing them onto a glass cover slip. For *Figure 8—figure supplement 1A*, FMR1 protein samples were instead dialyzed against 100 mM NaCl, 20 mM $NaPO_4$ pH 7.0 and 5 mM DTT, with 20 mg/ml ficol added prior to imaging.

Concentrated pAP and engrailed-2 protein samples were dialyzed against 100 mM NaCl, 20 mM $NaPO_4$ pH 7.0 and 5 mM DTT overnight at 4°C and then either diluted with the same buffer to the desired protein concentrations for imaging (pAP, *Figure 8C*) or with the addition of 20 mg/ml ficol (engrailed-2, *Figure 8—figure supplement 1B*). Droplet images were acquired using differential interference contrast with 40X, 63X or 100X objectives on either a Zeiss Axiovert 200M Epifluorescence microscope or a Zeiss Axio Observer. Temperatures were controlled using a PE100-ZAL inverted Peltier system from Linkam Scientific.

## Statistical analysis

Standard error estimated for measured population parameters was obtained by bootstrap analysis using random sampling with replacement, with 10,000 iterations. For measurements involving populations of features found within the PDB, we split observation data into dependent blocks by sampling against the list of PDBs used in calculating the parameter rather than by against the list of observed features. Statistics for many of these calculations are tabulated in *Appendix 1—table 1*. ROC curves were calculated as non-parametric step functions by empirical cumulative distributions, and AUC was estimated by direct measurement of the AUC, without smoothing. AUC values are tabulated in *Appendix 1—table 6*.

## Acknowledgements

We thank Hue Sun Chan, Régis Pomès, Simon Sharpe, Alan Moses, Fred Keeley, Lewis Kay, and Jacob Brady for useful discussions. We acknowledge funding to JDF-K from the Canadian Institutes of Health Research, Natural Sciences and Engineering Council of Canada and Canadian Cancer Society Research Institute. We also acknowledge the use of The Hospital for Sick Children Structural & Biophysical Core Facility.

## Additional information

### Funding

| Funder | Grant reference number | Author |
| --- | --- | --- |
| Canadian Institutes of Health Research | 114985 | Julie Deborah Forman-Kay |
| Natural Sciences and Engineering Research Council of Canada | 06718 | Julie Deborah Forman-Kay |
| Canadian Cancer Society Research Institute | 703477 | Julie Deborah Forman-Kay |

The funders had no role in study design, data collection and interpretation, or the decision to submit the work for publication.

### Author contributions

Robert McCoy Vernon, Conceptualization, Data curation, Software, Formal analysis, Validation, Investigation, Visualization, Methodology, Writing—original draft, Writing—review and editing; Paul Andrew Chong, Conceptualization, Supervision, Project administration, Writing—review and editing; Brian Tsang, Investigation, Writing—review and editing; Tae Hun Kim, Conceptualization, Investigation, Writing—review and editing; Alaji Bah, Conceptualization, Data curation, Investigation, Methodology, Writing—review and editing; Patrick Farber, Conceptualization, Resources, Data curation,

Investigation, Methodology, Writing—review and editing; Hong Lin, Resources, Investigation, Methodology; Julie Deborah Forman-Kay, Conceptualization, Supervision, Funding acquisition, Project administration, Writing—review and editing

### Author ORCIDs
Julie Deborah Forman-Kay (ID) http://orcid.org/0000-0001-8265-972X

### Decision letter and Author response
Decision letter https://doi.org/10.7554/eLife.31486.044
Author response https://doi.org/10.7554/eLife.31486.045

## Additional files

### Supplementary files
• Source code 1. Python scripts for identifying PDB contacts. Pi-pi contact identification scripts suitable for reproducing the annotation data contained in *Figure 1—source data 1* and *2*.
DOI: https://doi.org/10.7554/eLife.31486.021

• Source code 2. Final predictor code package. Python script and associated database files for the final phase separation propensity predictor.
DOI: https://doi.org/10.7554/eLife.31486.022

• Transparent reporting form
DOI: https://doi.org/10.7554/eLife.31486.023

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

## Appendix 1

DOI: https://doi.org/10.7554/eLife.31486.024

# Supplementary results

## Catalytic sites

To investigate the functional relevance of planar pi-pi interactions, we examined the contact frequencies at positions with known enzymatic function. We observe that residues in our non-redundant protein set which are annotated as being catalytic in the Catalytic Site Atlas (*Furnham et al., 2014*) (N = 912 PDBs, 2914 catalytic residues) are more likely to be involved in planar pi-pi contacts than expected by the contact frequency of the given residue type, at $1.87 \pm 0.07$ times expectation overall and $1.42 \pm 0.07$ when normalized by catalytic residue frequency (*Appendix 1—table 3*), with Glu, Gln, Asn, and Asp having the largest increases, going from 12–19% average contact frequency to 21–39% in catalytic positions. In addition, by normalizing contact frequency by the number of van der Waals contacts each residue is involved in we observe that catalytic residues are $2.1 \pm 0.2$ times more likely to have pi-pi stacking interactions with their VDW contacts that are also annotated as being catalytic than they are to non-catalytic neighbors, suggesting that the contacts themselves play a role in defining the constrained geometries of the active site. This analysis only covers the protein-protein interactions made within the binding pocket, and does not cover the nature of the contacts made by catalytic residues to ligands, which, judging from our small molecule binding data (*Appendix 1—table 2*), could significantly increase the overall rate. (An example of a catalytic side with both protein-protein and protein-ligand pi-pi contacts is shown in *Figure 2B*.). We note that the effect of these interactions may involve some energetic cooperativity with other interactions, as they are often found in large networks, and commonly involve sidechains stacked simultaneously to both the donor and acceptor groups of an $sp^2$-$sp^2$ hydrogen bond (*Figure 2C*).

## Backbone interactions

Roughly half of the observed pi-pi contacts involve the peptide backbone and seem to be particularly important in defining structure at the termini of strands and helices (*Figure 2D*). We found that contact frequency increases at backbone positions located at the transition points between different types of secondary structure, with up to a 3–4 fold increase in contact frequency around the terminal positions in strands and helices (*Appendix 1—figure 5*) primarily involving local sidechain contacts to the last peptide bond involved in hydrogen bonding. We further observed that many of these contacts are directly involved in motifs known to be important for stabilizing secondary structure during folding, including β-hairpins and helix-caps (*Serrano and Fersht, 1989*; *Trevino et al., 2007*). These interactions have previously been recognized for aromatic sidechains (*Tóth et al., 2001*), but we find that non-aromatic $sp^2$ sidechains are, in aggregate, more likely to form these backbone contacts (*Figure 1C*).

To examine the relevance of these backbone contacts in defining structural motifs we searched for examples of specific motifs with exceptionally high contact frequencies. We present our top examples in *Appendix 1—figure 5*. *Appendix 1—figure 5A* shows a β-hairpin with DSSP assignment (*Kabsch and Sander, 1983*) 'ETTTTE', a motif found in $14.4 \pm 0.5\%$ of the structures in our high resolution PDB set (N = 822/5718). Across these observations we find a planar pi-pi contact frequency at the peptide position involved in the hydrogen bond defining the turn $38 \pm 2\%$ of the time (N = 401/1051),~22 times the median contact frequency (of 1.7%) over all backbone groups, suggesting that these pi-pi contacts play a role in defining the motif.

As a way of providing a sidechain specific test that excludes aromatic interactions and cation-pi, we looked at arginine residues and found that when arginine is in a helix ($\geq 4$ residues from the N-terminal position, by DSSP) the sidechain group forms a pi-pi contact to the peptide bond

between residues i-3 and i-2 on average 9.2% of the time (N = 18507), where the arginine is effectively stacking to the surface of the helix. When looking at sidechains found closer to the helix N-terminus we find that this frequency increases to 13.6% for the position where the i-3/i-2 peptide group forms the terminal hydrogen bond (N = 2495), and then drops to 0.14% when it is no longer part of the helix (N = 1438).

## RNA binding

We also examined the planar pi-pi contact frequencies in RNA/protein complexes, using 94 X-ray structures with at least 10 RNA bases from our non-redundant proteins set. Overall, $9.2 \pm 1.4\%$ (N = 3847) of RNA bases in these RNA/protein complexes form planar pi-pi interactions with protein. This is heavily biased towards bases that are not already stacking to other bases. Thus, bases that have no base stacking interactions ($16.2 \pm 1.8\%$ of all bases) have planar pi-pi contacts to protein $31.2 \pm 4.4\%$ of the time, and those that only stack to RNA on one side ($17.3 \pm 1.1\%$ of all bases) have planar pi-pi contacts to protein $17.7 \pm 2.6\%$ of the time. This often situates the protein/RNA interface at positions for which the RNA secondary structure is disrupted.

Arginine sidechains are the largest contributor to RNA/protein planar pi-pi interactions, found in $35.0 \pm 3.3\%$ of cases, compared to $34.7 \pm 4.0\%$ involving any of the four aromatic sidechains (Phe, Tyr, His, Trp) and $21.9 \pm 2.8\%$ the peptide backbone. Arginine's role in binding RNA is well known and can also be explained by its abilities to form charge interactions with the phosphate backbone and hydrogen bonds to base pairs. We observe that arginine residues in VDW contact with RNA form planar pi-pi interactions 28% of the time, and that planar pi-pi contacts are, relative to other arginine to RNA VDW contacts, more likely to be found in simultaneous contact to the phosphate group. Thus, arginine in VDW contact with RNA has a 36% and 9% chance of being in contact with $PO_4$ or hydrogen bonded to a base, respectively, while arginine with a pi-pi contact to RNA has a 63% and 15% chance of being in contact with $PO_4$ or hydrogen bonded to a base, demonstrating that these interactions do not compete and may be synergistic or cooperative (*Figure 2E*).

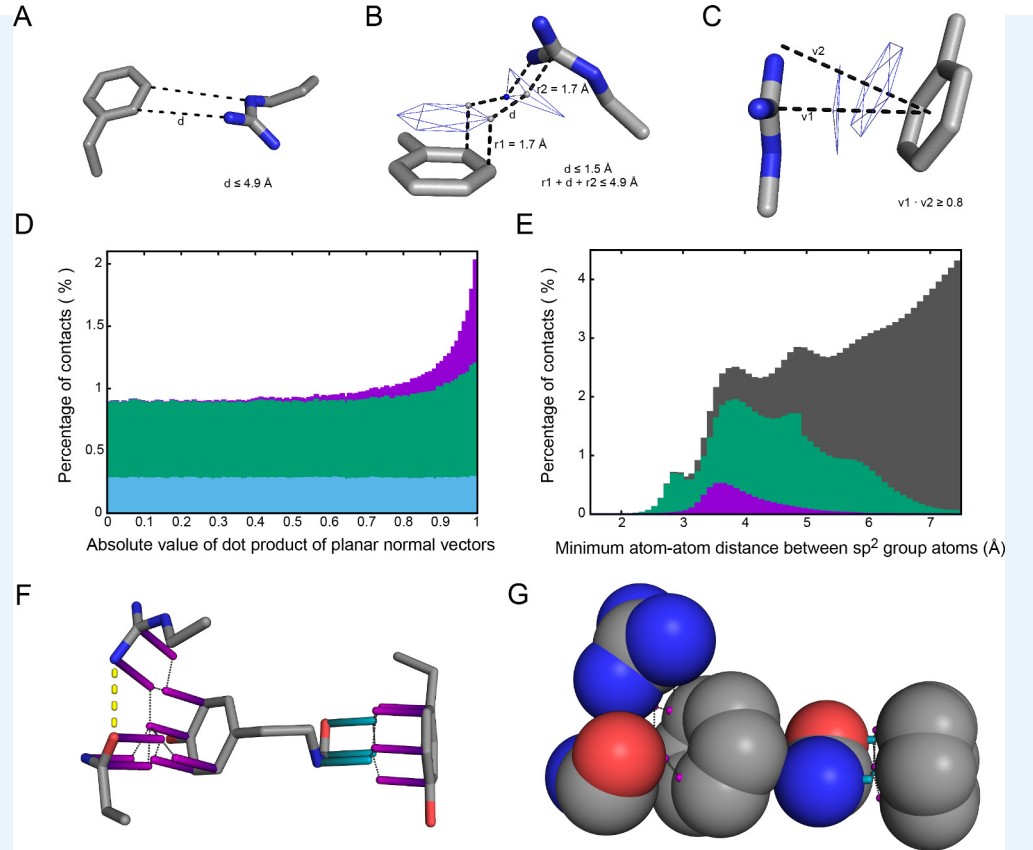

**Appendix 1—figure 1.** Contact definitions. (**A**) Contacts are identified first as $sp^2$ planes in which at least two pairs of atoms come within 4.9 Å of one another, and then by restricting to the subset with (**B**) planar surfaces (at the carbon VDW radius of 1.7 Å) with points along the planar normal vectors coming within 1.5 Å of one another and (**C**) a planar orientations for which the absolute value of the dot product of normal vectors is $\geq$0.8. (**D**) Shows the rationale for these restrictions, where binning sidechain-sidechain interactions by the relative orientation between planes shows that planar (same-orientation) interactions, primarily in the 0.8 to 1.0 range (angles between the planes from 0 deg to 36 deg), show enrichment relative to the uniform distribution expected for random orientations. Of these, interactions with only one atom-atom pair within VDW contact (shown in blue) have no bias. Enrichment comes entirely from contacts with either two pairs of planar surfaces within 1.5 Å of each other (shown in purple) or two distinct pairs of atoms within 4.9 Å but without the planar surface contact (shown in green). (**E**) Minimum distance measurements between pairs of atoms found in separate $sp^2$ groups, measured from the closest pairing for each atom. Gray shows all sidechain-sidechain measurements, and green/purple show distances corresponding to the groups in D. (**F**) Representative examples of sidechain-sidechain and sidechain-backbone pi-contacts are shown as sticks (PDB: 1gde), with carbon atoms in gray, oxygen in red, and nitrogen in blue. Planar normal vectors extended to the carbon VDW radius, representing pi-orbital locations, are shown as purple rods for sidechain groups and blue rods for backbone groups, and the yellow line denotes a hydrogen bond where both donor and acceptor atoms are in pi-contact distance to a third sidechain. (**G**) A space-filling representation of the $sp^2$ atoms in F, with gray lines between normal vector rods used to show the planar surface measurements taken for defining pi-contacts.

DOI: https://doi.org/10.7554/eLife.31486.025

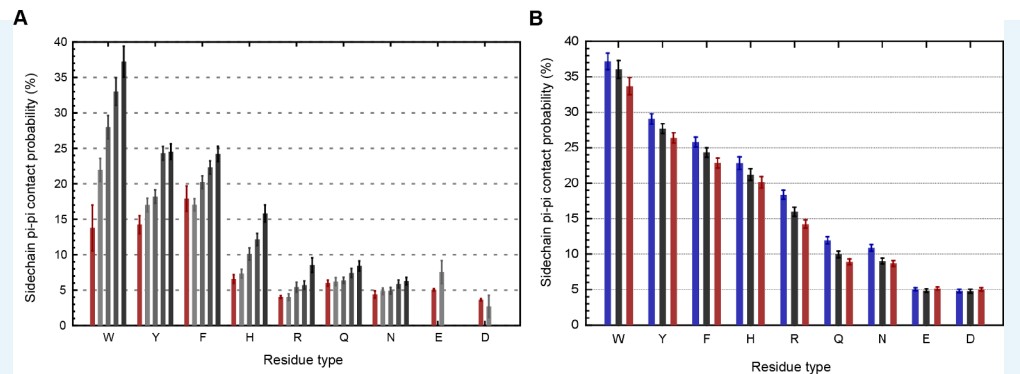

**Appendix 1—figure 2.** Cross validation against NMR restraints and X-ray structure resolution. (**A**), The relationship between contact frequency and experimental data quality is not unique to crystallography, as shown by the effect of increasing the number of restraints on sidechain specific contact frequencies over 2589 structures solved by NMR. For each sidechain/protein combination we calculated the average number of distance restraints involving sidechain atoms (from the first $sp^2$ atom onward), and then binned residues into five categories, with red for structures without any sidechain distance restraints for that residue type, and ranking quartiles from light gray to black by order of increasing restraints, where the consistent increase in contact frequency from left to right confirms that more restraints result in higher planar pi-contact frequencies. For Glu and Asp, less than 1% of the structures were derived using distance restraints to the carboxyl's lone carbon atom so we did not split them into quartiles. (**B**), To control for potential sample bias we also tested the relationship between resolution and contact frequency for crystallographic structures that have been solved at least three different times at different resolutions, with bars showing contact frequencies over identical populations of residues for the highest (blue), median (black), and lowest resolution (red) structures. Error bars show standard error of the mean (SEM).

DOI: https://doi.org/10.7554/eLife.31486.026

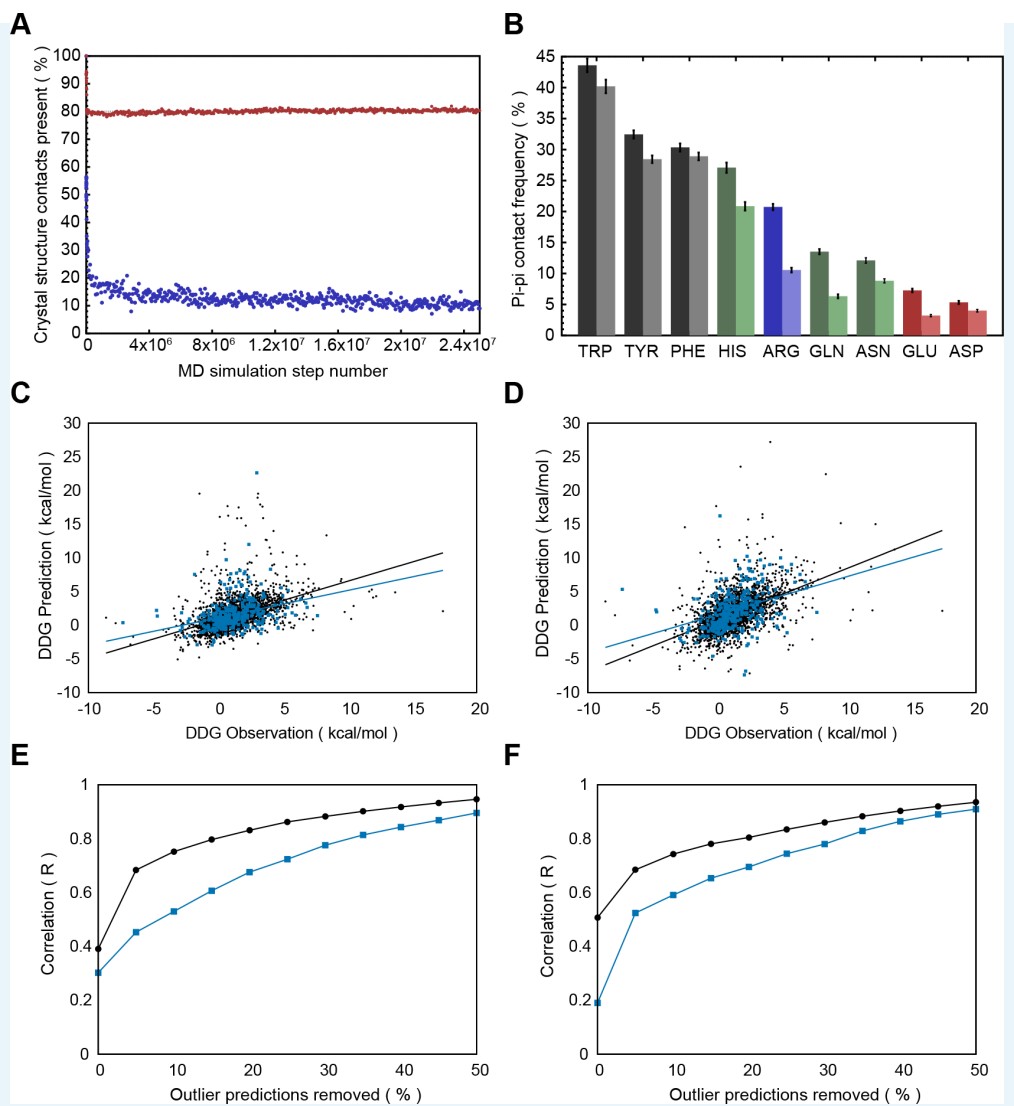

**Appendix 1—figure 3.** Pi-pi interactions underestimated by some energy functions. (**A**), Contact frequency during molecular dynamics simulations of 100 proteins, made available through Dynameomics (***Kehl et al., 2008***), shows a rapid initial loss of >80% of sidechain pi-contacts which continues to decline throughout the simulation (blue points). By comparison, sidechain hydrogen bonding shows a stable loss of only 20% of interactions (red points). (**B**), Minimization of 762 crystal structures against the Talaris2014 energy function by Rosetta3.4 (***Leaver-Fay et al., 2011***; ***O'Meara et al., 2015***), with starting contact frequencies (left bars) decreasing after minimization (right bars). (**C–F**), Analysis of the relationship between the energetic effects of point mutations (ΔΔG) and pi-contacts for experimental ΔΔGs (blue bars) and ΔΔGs predicted by simulation against the FOLDX force field (***Schymkowitz et al., 2005***) (**C,E**) and Rosetta (**D,F**). Panels C,D show predicted ΔΔG values vs. observation for residues that are not involved in pi-contacts in black, and residues that are involved in pi-contacts in blue, with lines of best fit colored the same. Panels E,F show how correlation values change as outliers are removed, with correlation consistently worse for mutations involving pi-contacts (blue lines) relative to those that don't (black lines).

DOI: https://doi.org/10.7554/eLife.31486.027

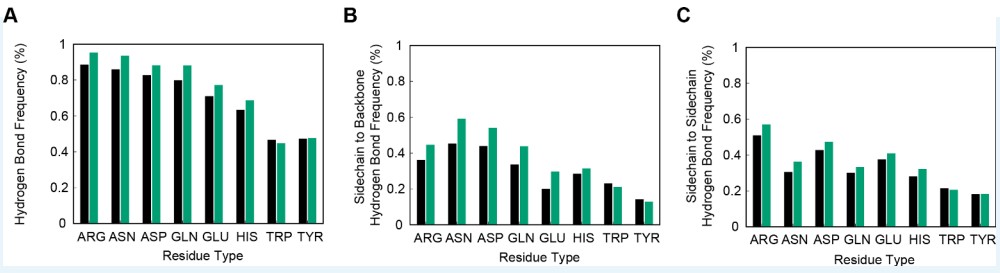

**Appendix 1—figure 4.** Hydrogen bonding correlates with planar-pi contacts. Percentage of sidechains involved in at least one hydrogen bond is shown for sidechains that are not in a planar-pi contact in black, and for sidechains that are in a planar-pi contact in green, with panel (**A**) showing the hydrogen bond frequency across all groups, including ligands and water, (**B**) showing the hydrogen bond frequency to backbone atoms, and (**C**) showing the frequency of hydrogen bonding to a sidechain. Hydrogen bond frequency consistently increases with planar pi-pi contacts for all sidechains but Trp and Tyr.

DOI: https://doi.org/10.7554/eLife.31486.028

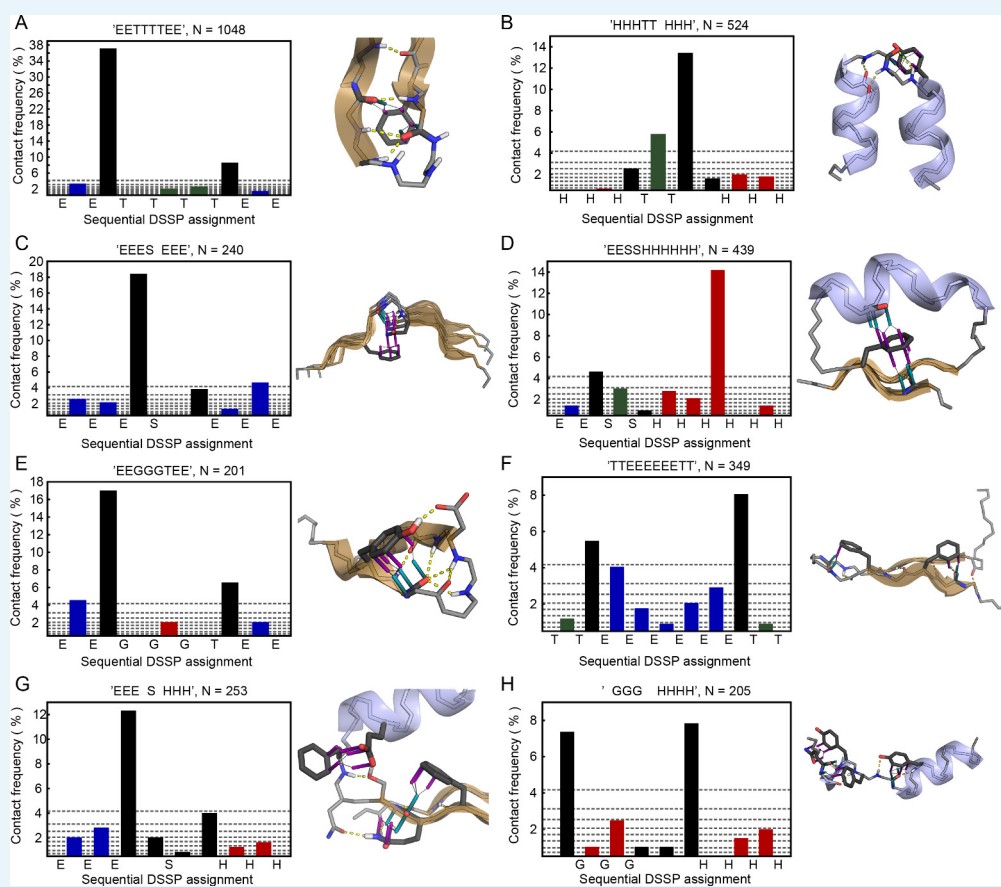

**Appendix 1—figure 5.** Backbone pi-pi contacts in secondary structure motifs. Examples of secondary structure motifs showing enrichment for local backbone pi-contacts (contacts made to sidechains within 5 residues of the peptide bond) are displayed. Bar graphs show contact frequency at each position in a motif, as defined by DSSP (*Kabsch and Sander, 1983*) abbreviated residue class ('E', 'S', 'T', 'H', 'G', and ' '), with bars colored by the associated residues, with green for peptide bonds between two residues classified as turns, blue for bonds in strands, red in helices, and black for bonds that are either unclassified or present at the transition point between classifications. Gray horizontal lines represent the decile values across all backbone contact frequencies, showing that the bonds most likely to end up in the top decile come primarily from transition points between secondary structures (ranging from

2x to 20x enrichment, relative to the median of 1.7%). Protein structures show representative examples of each motif with contacts found at the most enriched position, taken from (**A**), PDB:1aap, (**B**), PDB:1gte, (**C**), PDB:1k5c, (**D**), PDB:1nhc, (**E**), PDB:1k3i, (**F**), PDB:1i8k, (**G**), PDB:2c4w, and (**H**), PDB:1kwf.

DOI: https://doi.org/10.7554/eLife.31486.029

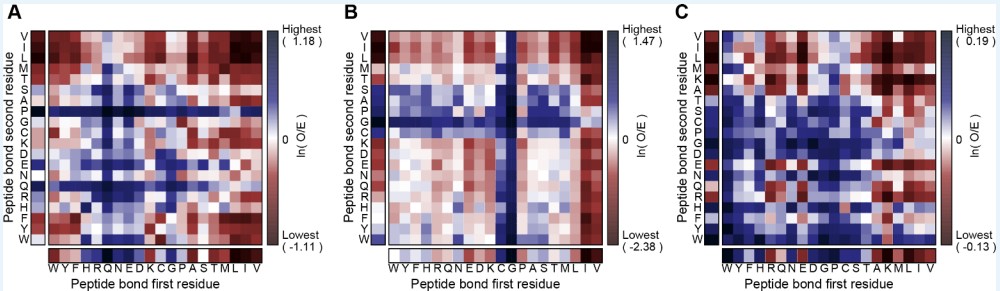

**Appendix 1—figure 6.** Peptide sequence effects on contact frequency. Heatmaps show enrichment in the total proportion of planar pi-pi contact involvements observed for peptide bonds between two residues (the first, N-terminal residue on the x-axis and the second, C-terminal residue on the y-axis) relative to the proportion of peptide bonds. Enrichment for (**A**) short-range contacts (sequence separation <5) and (**B**) long-range contacts (separation ≥5 or a different chain), respectively, to the peptide bond itself. (**C**), Enrichment for finding residues within 5 residues of a sidechain that makes a pi-contact to any group in the structure, demonstrating general sequence effects on the contact propensity of neighboring residues. The color gradient shows the natural logarithm of the observed over expected ratio.

DOI: https://doi.org/10.7554/eLife.31486.030

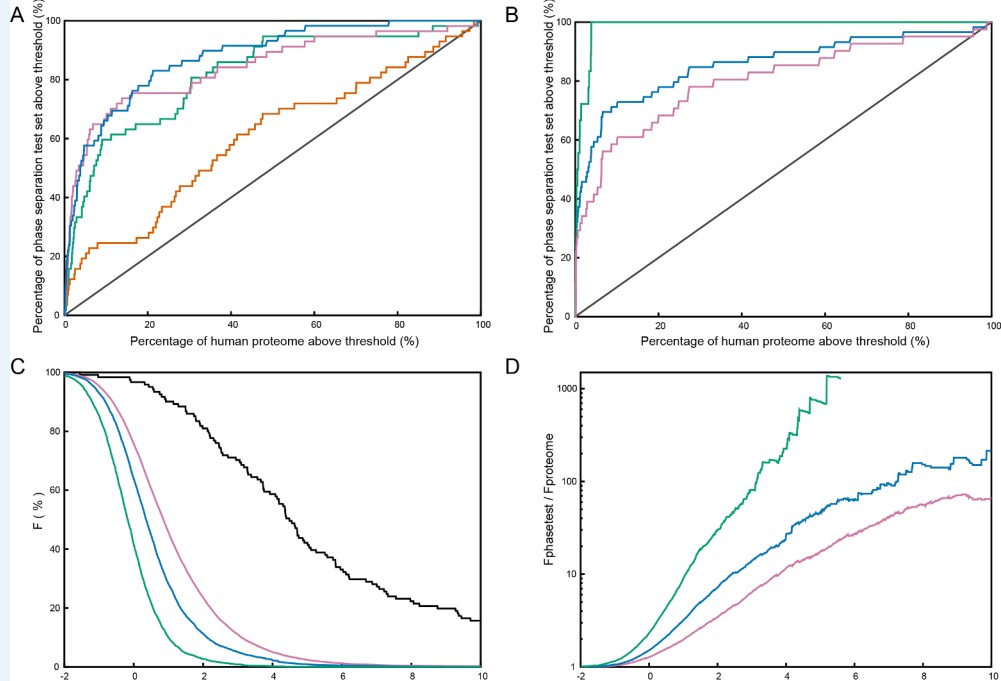

**Appendix 1—figure 7.** Phase separation propensity predictor testing. (**A**), ROC curve comparisons of predictor quality for scores made at different points during the training process, measuring ranking against the full test set (N = 62) vs. the human proteome (only sequences with length ≥140) with green showing the results for the highest number of pi-contacts predicted for any 100 residue window, without any weighting for type (AUC:0.82 ± 0.03), pink and orange showing the same measurement split between long-range (AUC:0.85 ± 0.03) and short-range contacts (AUC:0.62 ± 0.04), respectively, and blue showing

the final predictor, which uses weighted combinations of both short- and long-range contact predictions (AUC: 0.88 ± 0.02). (**B**), the final score tested against 59 phase-separating sequences designed by the Chilkoti lab (**Quiroz and Chilkoti, 2015**; **MacEwan et al., 2017**; **Simon et al., 2017**) (detailed in **Figure 5—source data 1C**), with comparisons against the full set shown (N = 59) in blue (AUC: 0.86 ± 0.03), and then split into green for 18 proteins shown to phase separate from soluble to insoluble as temperature decreases (AUC:0.99 ± 0.01) and pink for the remaining 41 proteins which phase separate from soluble to insoluble as temperature increases (AUC:0.80 ± 0.04). (**C**), Fraction of sequences at or above a given PScore, with the combined pool of phase separation test set proteins (N = 121), in black, being compared to three reference proteome sets, with human in pink, *S. cerevisiae* in blue, and *E. coli* in green. (**D**), Enrichment plot for data shown in (**C**), with ≥PScore frequency for the test set shown relative to proteome frequencies. Analysis based on **Figure 5**-source data 1 and 2.

DOI: https://doi.org/10.7554/eLife.31486.031

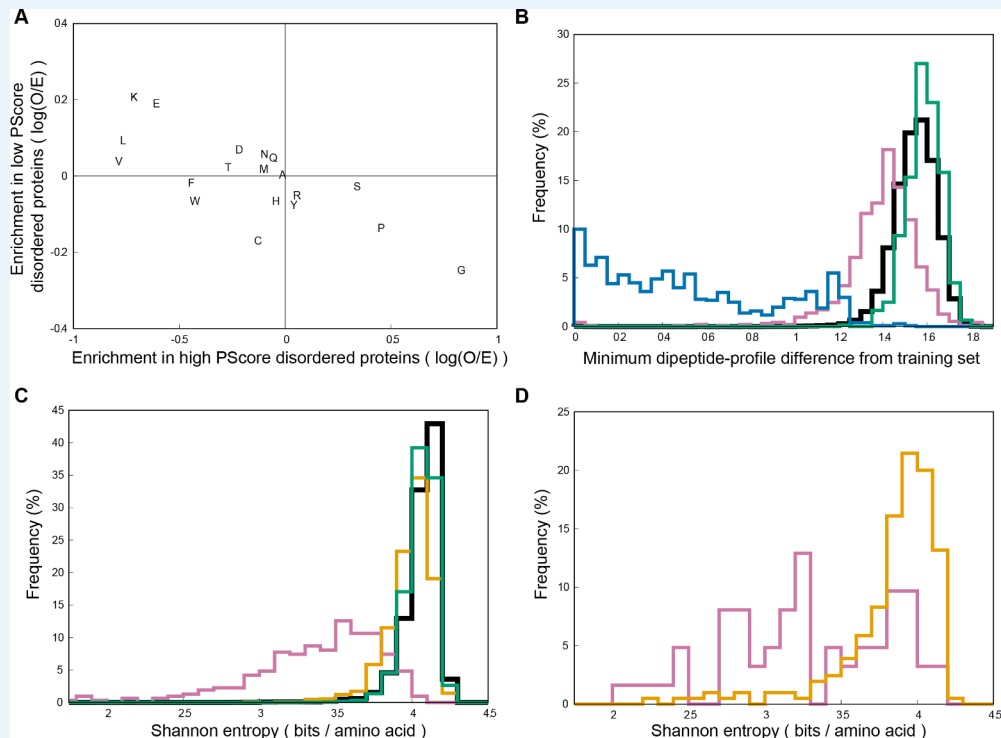

**Appendix 1—figure 8.** Sequence comparisons of high PScore proteins. Panel (**A**) shows compositional bias, relative to the human average, for the high PScore disordered proteins (x-axis) and low PScore disordered proteins (y-axis) used in panel B. High PScore disordered proteins are enriched primarily in Pro and Gly, while low PScore disordered proteins are not enriched in either, but enriched primarily in Lys and Glu, matching our observation that Arg to Lys mutations abrogate phase separation propensity. Panel (**B**) shows similarity to the training set measured by minimum dipeptide profile distance to any training set protein, as described in the methods. High PScore (≥4.0) human sequences (in pink) are on average closer to the training set than are all human proteins (in black) or PDB sequences (in green), but the range overlaps with both, and is distinct from the similarity seen in blast level homologs of the training set (in blue). Panel (**C**) shows Shannon entropy distributions of the human proteome (in black), the PDB (in green), and of a set of human proteins proteins predicted to have long stretches of disorder (Disprot3 ≥0.8) split into those with high PScore (≥4, N = 310) (in pink) and low PScore (<1.0, N = 1044) (in orange), showing that PScore but not disorder results in a bias towards lower sequence entropy, suggesting a compositional bias in phase-separating sequences. Panel (**D**) shows Shannon entropy values for our natural-protein phase separation test set (N = 62) in pink and the disorder-containing human proteins found in Disprot

(N = 205) in orange, confirming the observation in panel C that lower Shannon entropy sequences are associated with phase separation.

DOI: https://doi.org/10.7554/eLife.31486.032

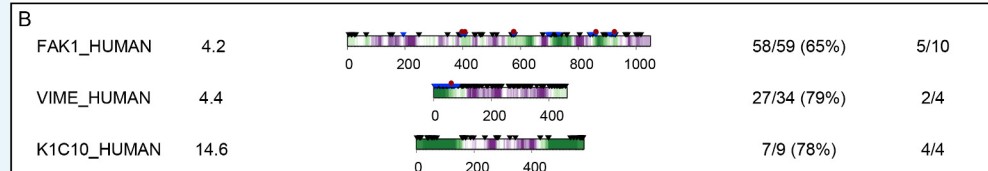
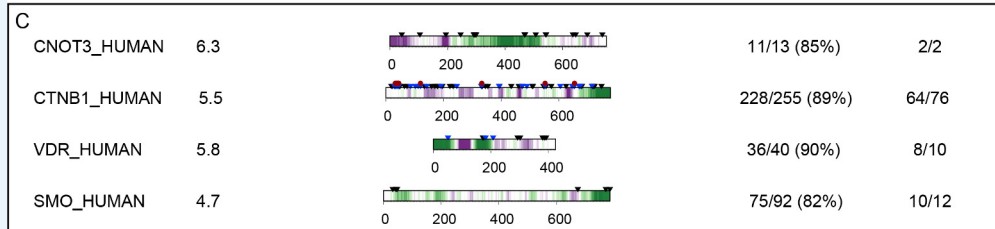
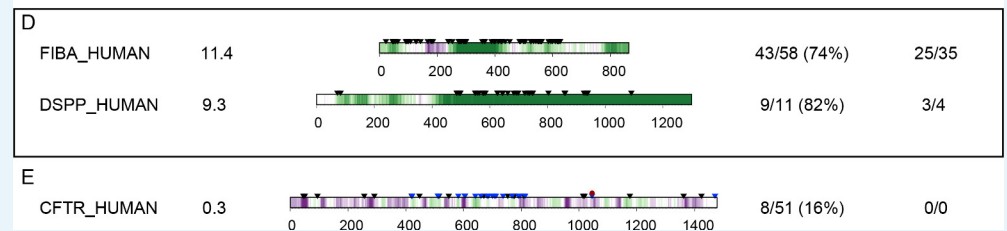

| Protein Name | PScore | | GO Terms (Enriched/Total) | Highest Scoring (Enriched/Total) |
|---|---|---|---|---|
| **A** | | | | |
| FMR1_HUMAN | 4.7 | | 76/88 (86%) | 9/12 |
| SYPH_HUMAN | 7.3 | | 18/28 (64%) | 10/15 |
| **B** | | | | |
| FAK1_HUMAN | 4.2 | | 58/59 (65%) | 5/10 |
| VIME_HUMAN | 4.4 | | 27/34 (79%) | 2/4 |
| K1C10_HUMAN | 14.6 | | 7/9 (78%) | 4/4 |
| **C** | | | | |
| CNOT3_HUMAN | 6.3 | | 11/13 (85%) | 2/2 |
| CTNB1_HUMAN | 5.5 | | 228/255 (89%) | 64/76 |
| VDR_HUMAN | 5.8 | | 36/40 (90%) | 8/10 |
| SMO_HUMAN | 4.7 | | 75/92 (82%) | 10/12 |
| **D** | | | | |
| FIBA_HUMAN | 11.4 | | 43/58 (74%) | 25/35 |
| DSPP_HUMAN | 9.3 | | 9/11 (82%) | 3/4 |
| **E** | | | | |
| CFTR_HUMAN | 0.3 | | 8/51 (16%) | 0/0 |

**Appendix 1—figure 9.** Prediction examples. Per-residue PScores used to calculate the final full sequence PScore are shown for a selection of human proteins, with residues colored from purple (PScore ≤ −2) to white (PScore = 0) to green (PScore ≥4.0). Black triangles denote residues annotated by PhosphoSitePlus as targets of PTMs, blue triangles denote modification sites with known regulatory significance, and red circles denote modification sites with known disease relevance. Proteins are annotated with the percentage of GO terms (with at least 10 human proteins) and high PScore-enriched GO terms (Panther analysis, PScore ≥4, with O/ E > 1) of which the protein is a member, as well as the total number of each for which the annotated protein has the highest PScore in the set. Examples are grouped by (**A**), involvement in synaptic plasticity and neuronal behavior, showing synaptic functional regulator FMR1, and synaptophysin; (**B**), intracellular biomaterials and related structural proteins, showing focal adhesion kinase 1, vimentin, and keratin type I cytoskeletal 10; (**C**), proteins involved in signaling pathways, showing CCR4-NOT transcription complex subunit 3, β-catenin, vitamin D3 receptor, and Smoothened homolog; and (**D**), proteins involved in extracellular biomaterials, showing fibrinogen alpha chain and dentin sialophosphoprotein. (**E**) The cystic fibrosis transmembrane conductance regulator is shown as an example of a negative prediction, even though containing a large region of intrinsic disorder (residues ~650–840).

**Appendix 1—table 1.** Contact statistics in high resolution, low R-factor protein structures.

| Measurement | Value | N |
|---|---|---|

*Appendix 1—table 1 continued*

| Measurement | Value | N |
|---|---|---|
| **Contacts per 100 residues** | | |
| Pi-Contacts per 100 residues, averaged over PDBs | 6.06 ± 2.5* | 5,718 PDBs |
| Pi-Contacts per 100 residues, averaged over all residues | 6.27 ± 0.03 | 1,384,228 residues |
| **Atom Contact Probabilities (%)** | | |
| Heavy Atoms in a Pi-Contact | 6.10 ± 0.03 | 10,836,487 atoms |
| sp$^2$ Heavy Atoms in a Pi-Contact | 10.52 ± 0.05 | 6,283,150 atoms |
| Heavy Atoms within 4.9 Å of any Pi-Contact | 32.1 ± 0.1 | 10,836,487 atoms |
| **Sidechain-Sidechain Contact Proportions (%)** | | 25,930 contacts |
| Aromatic to Aromatic | 24.73 ± 0.29 | " |
| Aromatic to Non-Aromatic | 53.24 ± 0.33 | " |
| Non-Aromatic to Non-Aromatic | 22.03 ± 0.28 | " |
| **All Contact Proportions (%)** | | 86,860 contacts |
| Sidechain to Sidechain | 29.85 ± 0.17 | " |
| Aromatic Sidechain to Backbone | 40.41 ± 0.20 | " |
| Non-Aromatic Sidechain to Backbone | 22.80 ± 0.16 | " |
| Backbone to Backbone | 6.94 ± 0.09 | " |
| Aromatic to Aromatic | 7.38 ± 0.10 | " |
| outnumbered by Aromatic to Non-Aromatic | 7.6 ± 0.1 to 1 | " |
| outnumbered by Non-Aromatic to Non-Aromatic | 3.9 ± 0.1 to 1 | " |
| **Arginine Sidechain Contacts (per 100 residues)** | | 61,877 residues |
| Contact to Aromatic | 9.74 ± 0.13 | " |
| Contact to Backbone | 10.6 ± 0.13 | " |
| Contact to Glutamine/Asparagine Sidechain | 1.96 ± 0.06 | " |
| Contact to Glutamate/Aspartate Sidechain | 1.49 ± 0.05 | " |
| Contact to Arginine Sidechain | 3.63 ± 0.11 | " |

*This error range shows the standard deviation between PDBs; other error ranges show standard error of the mean for averages computed over all PDBs.

DOI: https://doi.org/10.7554/eLife.31486.034

**Appendix 1—table 2.** Small molecule contact frequencies.

| Amino acid sp$^2$group* | # PDBs | # Ligand Groups | Ligand Pi-Pi Contact Frequency (%) | # Protein Groups | Protein contact frequency (%) | O/E (Ligand/ Protein) |
|---|---|---|---|---|---|---|
| GLU Sidechain | 84 | 209 | 16.3 ± 4.1 | 5353 | 5.0 ± 0.4 | 3.3 ± 0.8 |
| HIS Sidechain | 36 | 80 | 42.5 ± 8.0 | 530 | 17.4 ± 2.7 | 2.4 ± 0.7 |

*Appendix 1—table 2 continued on next page*

*Appendix 1—table 2 continued*

| Amino acid sp$^2$ group* | # PDBs | # Ligand Groups | Ligand Pi-Pi Contact Frequency (%) | # Protein Groups | Protein contact frequency (%) | O/E (Ligand/ Protein) |
|---|---|---|---|---|---|---|
| PHE Sidechain | 36 | 93 | 53.8 ± 9.0 | 1389 | 28.4 ± 2.3 | 1.9 ± 0.3 |
| ARG Sidechain | 61 | 145 | 43.9 ± 6.0 | 2878 | 15.9 ± 0.9 | 2.8 ± 0.5 |
| TYR Sidechain | 30 | 68 | 58.8 ± 12.2 | 806 | 19.9 ± 1.9 | 3.0 ± 0.7 |
| GLN Sidechain | 21 | 50 | 22.0 ± 8.9 | 800 | 13.0 ± 2.3 | 1.8 ± 0.8 |
| ASP Sidechain | 39 | 86 | 3.5 ± 2.0 | 2153 | 4.5 ± 0.8 | 0.8 ± 0.5 |
| TRP Sidechain | 43 | 109 | 50.5 ± 8.0 | 377 | 28.9 ± 2.3 | 1.7 ± 0.3 |
| ASN Sidechain | 11 | 32 | 6.3 ± 7.3 | 466 | 8.6 ± 1.9 | 0.7 ± 0.9 |
| Amino Carboxyl | 688 | 1704 | 8.9 ± 1.1 | 976 | 5.7 ± 1.2 | 1.5 ± 0.4 |

| Small molecule† | # PDBs | # Free Ligands | Ligand Pi-Pi Contact Frequency (%) | # sp$^2$ Atoms | RCSB Ligand ID | Isomeric SMILES |
|---|---|---|---|---|---|---|
| Ethanal | 44 | 76 | 3.9 ± 2.9 | 2 | ACE | CC = O |
| Formic Acid | 444 | 2093 | 11.0 ± 0.8 | 3 | FMT | OC = O |
| Acetate Ion | 1664 | 4794 | 12.9 ± 0.6 | 3 | ACT | CC([O-])=O |
| Acetic Acid | 403 | 1133 | 13.5 ± 1.5 | 3 | ACY | CC(O)=O |
| Nitrate Ion | 225 | 852 | 15.3 ± 1.7 | 4 | NO3 | [O-][N+]([O-])=O |
| Guanidine | 32 | 115 | 15.7 ± 4.7 | 4 | GAI | NC(N)=N |
| Urea | 23 | 91 | 16.5 ± 4.0 | 4 | URE | NC(N)=O |
| Imidazole | 279 | 684 | 26.6 ± 2.4 | 5 | IMD | C1C[NH+]C[NH]1 |

*Entries containing amino acids or small sp$^2$ containing planar molecules as free ligands were downloaded from the PDB (filtered to maximum sequence redundancy of 90% and 3 Å resolution) and pi-pi contact frequencies for ligands and their corresponding protein based equivalents were determined.

The majority of amino acids are more likely to form pi-pi contacts to protein when found as non-covalently bound ligands, rather than as residues within a protein, confirming that pi-pi contacts are a consistent property of amino acid interactions involving protein.

†In order to avoid bias due to the constrained geometries of functional binding sites we also analyzed the contact frequencies of a variety of common buffer components, with contact frequencies found to increase with number of sp$^2$-hybridized atoms.

Ranges show standard error of the mean.

DOI: https://doi.org/10.7554/eLife.31486.035

**Appendix 1—table 3.** Pi-pi contact enrichment for catalytic residues. Frequency of involvement in contacts, at either backbone or sidechain sp$^2$ groups, is shown for individual residue types, residue independent (ANY), and residue type normalized (AVG), where catalytic residue contact frequency shows values for residues annotated as catalytic in the catalytic site atlas (*Furnham et al., 2014*) and non-catalytic residue contact frequency shows values for all other residues in the same structures. To normalize for possible differences in the number of contacts made by catalytic residues we also show number of pi-pi contacts divided by total number of VDW contacts, labeled as percent of VDW, and the percent of VDW ratio shows enrichment by dividing the catalytic percent of VDW value by the non-catalytic value. Error values are obtained by our standard bootstrap analysis (see Materials and methods), and enrichment values of greater than two standard deviations are shown in bold.

| Residue type | Non-catalytic contact frequency (%) | Catalytic residue contact frequency (%) | N catalytic | Enrichment | Non-catalytic percent of VDW (%) | Catalytic residue percent of VDW (%) | Percent of VDW ratio (cat./non) |
|---|---|---|---|---|---|---|---|
| ANY | 13.1 ± 0.1 | 24.5 ± 0.9 | 2914 | 1.87 ± 0.07 | 1.91 ± 0.09 | 3.94 ± 0.40 | 2.06 ± 0.23 |
| HIS | 31.9 ± 0.9 | 35.9 ± 2.3 | 471 | 1.12 ± 0.06 | 3.01 ± 0.26 | 4.33 ± 0.84 | 1.45 ± 0.31 |
| ASP | 12.9 ± 0.5 | 21.2 ± 2.0 | 448 | 1.65 ± 0.14 | 1.71 ± 0.22 | 3.08 ± 0.81 | 1.83 ± 0.54 |
| GLU | 11.7 ± 0.4 | 32.2 ± 2.5 | 370 | 2.75 ± 0.18 | 2.44 ± 0.31 | 6.57 ± 1.38 | 2.74 ± 0.71 |
| ARG | 26.7 ± 0.8 | 30.3 ± 3.2 | 287 | 1.14 ± 0.12 | 1.77 ± 0.29 | 7.85 ± 1.48 | 4.57 ± 1.26 |
| LYS | 6.2 ± 0.4 | 9.7 ± 2.0 | 259 | 1.56 ± 0.35 | 0.82 ± 0.20 | 0.37 ± 0.37 | 0.48 ± 0.52 |
| TYR | 36.1 ± 1.2 | 30.4 ± 3.7 | 171 | 0.84 ± 0.10 | 2.69 ± 0.39 | 4.62 ± 1.38 | 1.75 ± 0.59 |
| SER | 8.8 ± 0.5 | 13.0 ± 2.6 | 169 | 1.48 ± 0.28 | 0.83 ± 0.23 | 1.39 ± 0.70 | 1.84 ± 1.20 |
| CYS | 8.5 ± 1.3 | 14.7 ± 2.9 | 150 | 1.73 ± 0.26 | 1.45 ± 0.33 | 0.92 ± 0.66 | 0.68 ± 0.54 |
| ASN | 18.9 ± 1.1 | 26.6 ± 4.3 | 109 | 1.41 ± 0.20 | 1.88 ± 0.42 | 6.11 ± 1.76 | 3.42 ± 1.29 |
| GLY | 12.0 ± 0.7 | 16.2 ± 4.5 | 99 | 1.35 ± 0.41 | 1.26 ± 0.48 | 5.87 ± 1.92 | 5.72 ± 4.33 |
| THR | 6.8 ± 0.6 | 4.7 ± 2.3 | 86 | 0.69 ± 0.38 | 0.34 ± 0.18 | 0.88 ± 0.87 | 3.42 ± 4.17 |
| GLN | 18.0 ± 1.6 | 40.3 ± 7.3 | 62 | 2.24 ± 0.34 | 3.12 ± 0.53 | 1.88 ± 1.30 | 0.61 ± 0.44 |
| ALA | 7.8 ± 0.9 | 7.0 ± 3.3 | 57 | 0.91 ± 0.48 | 0.85 ± 0.43 | 1.28 ± 1.28 | 2.04 ± 2.75 |
| PHE | 34.2 ± 2.1 | 35.9 ± 7.1 | 53 | 1.05 ± 0.23 | 2.52 ± 0.64 | 4.92 ± 2.39 | 2.10 ± 1.27 |
| TRP | 46.2 ± 3.2 | 45.1 ± 7.5 | 51 | 0.98 ± 0.14 | 3.57 ± 0.73 | 2.88 ± 2.00 | 0.87 ± 0.68 |
| AVG | 18.7 ± 0.4 | 24.2 ± 1.2 | N/A | 1.42 ± 0.07 | | | |

DOI: https://doi.org/10.7554/eLife.31486.036

**Appendix 1—table 4.** Effect of sp$^2$ sidechain mutations on phase separation. Phase separation critical concentration values for the N-terminus (1-236) of human Ddx4 and three mutants, 9FtoA and 14FtoA, as reported in (**Nott et al., 2015**), and RtoK, where all arginine residues have been mutated to lysine (**Brady et al., 2017**), as well as for the C-terminus (445-632) of human FMR1 and one mutant with all arginine residues mutated to lysine.

| Sample | Concentration at which phase separation is observed (conditions) | # F | # R | Total # | Mw (Da) |
|---|---|---|---|---|---|
| Ddx4 1–236 | (24°C, 20 mM Na$_2$PO$_4$ pH 6.5, 100 mM NaCl) | | | | |
| WT | ~2 mg/mL | 14 | 24 | 236 | 25430 |
| 9FtoA | ~100 mg/mL | 5 | 24 | 236 | 24745 |
| 14FtoA | ~350 mg/mL | 0 | 24 | 236 | 24364 |
| RtoK | Not observed up to 400 mg/mL | 14 | 0 | 236 | 24758 |
| FMR1 445–632 | (4°C, 20 mM Na$_2$PO$_4$ pH 7.4, 2 mM DTT) | | | | |
| WT | ~16 mg/mL | 2 | 28 | 188 | 20573 |
| RtoK | Not observed up to 216 mg/mL | 2 | 0 | 188 | 19789 |

DOI: https://doi.org/10.7554/eLife.31486.037

**Appendix 1—table 5.** Comparison of phase separation prediction and disorder prediction. Two disorder predictors were tested on matched positive and negative sets to the phase separation predictor, comparing the relative discrimination of known phase-separating and known disordered proteins from the PDB, the human proteome, and the same set of known disordered proteins. AUC values are highlighted in blue for AUC >0.8, and red for AUC <0.7. Error values were obtained by bootstrap analysis.

| Positive set | AUC (vs. PDB) | AUC (vs. Human) | AUC (vs. Disprot) |
|---|---|---|---|
| Disopred3 (Disorder Predictor) | | | |
| Phase Separation Test Set | 0.982 ± 0.005 | 0.72 ± 0.03 | 0.58 ± 0.02 |
| Disprot Set | 0.977 ± 0.007 | 0.66 ± 0.03 | N/A |
| IUPRED-Long (Disorder Predictor) | | | |
| Phase Separation Test Set | 0.893 ± 0.007 | 0.70 ± 0.03 | 0.60 ± 0.02 |
| Disprot Set | 0.89 ± 0.01 | 0.64 ± 0.03 | N/A |
| PScore (Phase Separation Predictor) | | | |
| Phase Separation Test Set | 0.961 ± 0.005 | 0.88 ± 0.01 | 0.84 ± 0.01 |
| Disprot Set | 0.79 ± 0.02 | 0.58 ± 0.03 | N/A |

DOI: https://doi.org/10.7554/eLife.31486.038

**Appendix 1—table 6.** Retrospective analysis of predictor quality at different stages during the training process. AUC values for distinguishing proteomic phase-separating sequences from the human proteome are shown for prediction scores made from pi-contact frequencies (average contacts predicted per residue) obtained at each training step of the protocol in order of their sequential development, with prediction scores calculated as the highest number of contacts predicted for any given 100 residue window in each sequence. Analysis of the relative effects of different contact types was added by excluding contacts from each score and retesting. Standard error of the mean (SEM), by bootstrap analysis, is consistently in the range from 0.021 to 0.039.

| Training step | AUC at training step | Sidechain contacts only | Backbone contacts only | Short-range sidechain only | Long-range sidechain only | Short-range backbone only | Long-range backbone only |
|---|---|---|---|---|---|---|---|
| (1) Baseline Frequencies | 0.57 | 0.51 | 0.84 | 0.52 | 0.50 | 0.73 | 0.80 |
| 2) Context-Averaged Frequencies | 0.57 | 0.51 | 0.86 | 0.53 | 0.51 | 0.77 | 0.83 |
| (3) Smoothed Frequency Predictions | 0.82 | 0.64 | 0.89 | 0.59 | 0.65 | 0.71 | 0.85 |
| (4) Weight Optimized Final Predictor | 0.88 | N/A | N/A | N/A | N/A | N/A | N/A |

DOI: https://doi.org/10.7554/eLife.31486.039

**Appendix 1—table 7.** Sequence similarity comparison. Frequencies of dipeptides (pairs of neighboring amino residues) were computed for phase-separating proteins and the human proteome, and enrichment was measured by the percentage of human proteins with lower frequency than found in a given sequence. The fifteen dipeptides enriched (≥99%) in the most sequences within the phase separation test sets are shown in the table vs. enrichment values obtained for the phase separation training set and three experimentally verified proteins. Values in the top fifth percentile are shown in bold.

| Protein Name | Dipeptide enrichment (Percentage of human proteome with lower frequency) | | | | | | | | | | | | | | |
|---|---|---|---|---|---|---|---|---|---|---|---|---|---|---|---|
| | GV | VG | VP | PG | FG | RG | GR | GG | YG | GS | SG | GA | GF | GD | DS |
| Training Set Proteins | | | | | | | | | | | | | | | |
| Elastin | **100** | **100** | **100** | **100** | 97 | 31 | 32 | **99** | **99** | 20 | 20 | **100** | 89 | 38 | 30 |
| Nsp1 | 30 | 34 | 31 | 26 | **100** | 31 | 30 | 75 | 52 | 90 | 38 | **99** | 60 | 66 | 68 |
| TIA1 | 73 | 75 | 46 | 26 | 86 | 31 | 86 | 77 | **99** | 29 | 53 | 26 | 84 | 54 | 30 |

*Appendix 1—table 7 continued on next page*

*Appendix 1—table 7 continued*

| Protein Name | Dipeptide enrichment (Percentage of human proteome with lower frequency) | | | | | | | | | | | | | | |
|---|---|---|---|---|---|---|---|---|---|---|---|---|---|---|---|
| | GV | VG | VP | PG | FG | RG | GR | GG | YG | GS | SG | GA | GF | GD | DS |
| LAF1 | 30 | 78 | 65 | 29 | 67 | 99 | 99 | 100 | 77 | 88 | 97 | 65 | 78 | 97 | 32 |
| EIF4H | 30 | 65 | 31 | 52 | 98 | 99 | 95 | 99 | 52 | 99 | 42 | 79 | 99 | 98 | 89 |
| Ddx3x | 51 | 70 | 34 | 43 | 89 | 98 | 97 | 96 | 93 | 93 | 95 | 68 | 96 | 59 | 78 |
| hnRNPA1 | 30 | 55 | 31 | 44 | 100 | 99 | 99 | 100 | 99 | 99 | 98 | 44 | 99 | 60 | 79 |
| DDX4 | 33 | 77 | 48 | 53 | 98 | 96 | 91 | 89 | 59 | 87 | 96 | 29 | 98 | 96 | 45 |
| FUS | 30 | 31 | 31 | 83 | 78 | 100 | 99 | 100 | 100 | 98 | 99 | 33 | 93 | 91 | 57 |
| EWS | 52 | 31 | 35 | 97 | 51 | 100 | 99 | 100 | 100 | 72 | 61 | 30 | 97 | 91 | 48 |
| TAF15 | 36 | 38 | 31 | 30 | 53 | 100 | 99 | 100 | 100 | 92 | 99 | 26 | 71 | 100 | 94 |
| Experimentally Verified Proteins | | | | | | | | | | | | | | | |
| FMR1 | 69 | 89 | 93 | 44 | 43 | 96 | 94 | 83 | 62 | 62 | 34 | 70 | 48 | 41 | 67 |
| SCAF pAP | 75 | 50 | 89 | 91 | 43 | 49 | 73 | 97 | 75 | 92 | 96 | 96 | 40 | 36 | 44 |
| Engrailed-2 | 30 | 31 | 31 | 97 | 70 | 78 | 90 | 100 | 52 | 99 | 91 | 99 | 40 | 95 | 97 |

DOI: https://doi.org/10.7554/eLife.31486.040

**Appendix 1—table 8.** High PScore enrichment for human proteins with a greater than average number of post-translational modification (PTM) site annotations in Phosphosite+. PTM counts are controlled for protein length by taking the maximum number observed in any 100 residue window, and the threshold for an above average PTM count is defined as greater than the average plus one standard deviation. Errors show SEM by bootstrap analysis.

| Phosphosite+ PTM annotation type | PTM count threshold | Above threshold (N) | PScore > 4 (%) | Enrichment |
|---|---|---|---|---|
| O-GlcNAc | 1 | 158 | 17 ± 3 | 3.4 |
| Methyl | 2 | 2051 | 13.3 ± 0.7 | 2.7 |
| Phosphate | 10 | 2485 | 10.8 ± 0.8 | 2.2 |
| O-GalNAc | 1 | 456 | 10.1 ± 0.1 | 2.0 |
| Sumo | 1 | 1999 | 9.0 ± 0.7 | 1.8 |
| Acetyl | 3 | 1543 | 8.0 ± 0.7 | 1.6 |
| Ubiquitin | 4 | 1875 | 6.3 ± 0.5 | 1.3 |
| | | | | |
| Disease Relevant | 1 | 298 | 11 ± 2 | 2.1 |
| Regulatory Function | 2 | 1087 | 7.6 ± 0.8 | 1.5 |
| | | | | |
| Database Baseline | 0 | 18582 | 5.0 ± 0.2 | 1.0 |

DOI: https://doi.org/10.7554/eLife.31486.041

# Appendix 2

DOI: https://doi.org/10.7554/eLife.31486.042

## Prediction experiment

### Experimental design

The protocol for developing the final phase separation predictor was conceived as an empirical test of the hypothesis that pi-contacts have relevance to phase separation, with the logic being that if pi-contacts play a functional or energetic role in mediating protein phase separation then an accurate prediction of pi-contact rates (contacts formed per residue) should often be sufficient for predicting phase-separation behavior. To this end, we defined strict training sets, with 11 phase-separating sequences as the positive standard and sequences in our PDB sets as our negative standard, we kept the testing sets internally blind until choosing an arbitrary point to finalize the predictor, and we intentionally excluded any analysis of sequence features, such as charge patterns, amyloid propensity, and direct homology, that could improve predictor quality without testing this hypothesis.

Our protocol was split into four design steps and one test step, as follows. (1) We defined sequence dependent contact rates for different $sp^2$ groups by taking statistics from the PDB, (2) we measured average frequencies observed for groups found in specific sequence context, (3) we used the context data to train a sequence-based pi-contact rate predictor against the PDB, (4) we then optimized a weighting function for combining pi-contact predictions against the ability to discriminate the phase separation training set proteins from the PDB, and (5) we finalized the weight optimized predictor and tested it against the phase separation test set and proteomic test sets a single time, shown in the main text, with supplemental analysis of predictor quality throughout the design process done as a retrospective analysis.

### Step 1: Pi-contact rate measurement

From the non-redundant PDB chains previously used for structure analysis, we selected nonredundant structures into the training set by randomly selecting 17388 structures. The remaining 4347 structures were taken as the leave out set, being removed and held for the final testing step. Independence of the two sets is restrained only by the 60% identity cutoff used in obtaining the full list, which prevents broadly identical proteins from showing up in both sets. The ability to further guarantee independence is limited by the nature of homology, and was considered outside of the scope of this manuscript.

Sequences were extracted from the PDB REFSEQ annotations and then mapped to residues found in the structure, with missing density annotated as such. Pi-contact observations were then mapped to their involved residues with contacts split by sequence separation into long-range ($\geq 5$ or different chain) and short-range ($\leq 4$ residue). Contact rates were determined for individual $sp^2$ groups using nine residue identities for the sidechain groups and 400 residue identities for backbone groups, with backbone groups defined by both flanking residues. To measure local sequence effects we also calculated rates for non-flanking residue pairs, up to 40 residues apart, which when combined with the backbone groups produced 16400 residue pair types (20 n-terminal residues x 40 sequence separation distances x 20 C-terminal residues), which are observed in the training set at a median sample size of N = 8622, ranging from N = 563 to N = 41796 from the least to most populated.

### Step 2: Pi-contact averaged frequencies

For the initial sequence-based pi-contact predictor contact rate, observations for pairs of residues found within a fixed window length were averaged to produce a context-dependent estimate. To account for sampling error in rare sequence pairs we estimated the standard error of these measurements by a limited bootstrap analysis, using 200 randomly sampled

(without replacement) 70% cut subsets of the training set. To average observation values, a given $sp^2$ system was first defined by its residue type (a single amino acid for sidechain groups or the sequential amino acids for backbone groups). Next, residues less than 40 residues away in the primary sequence (both sides) were compared to our precomputed database, to obtain a comprehensive list of rate values for all residue-distance-residue pairings found in this window. These values were then averaged using a sampling error correction, intended to weight observations by confidence, using the following equation, where $R$ is the error weighted contact frequency at position $x$ as obtained by averaging over the closest $l$ positions, $P$ is the database frequency observed for $x$ a given residue pair ($x$ and $x + y$), and $\sigma$ is the standard error of the mean obtained for that database frequency.

$$R_{x,l} = \sum_{\substack{-l \leq y \leq l \\ y \neq x}} \frac{P_{x,y}}{\sigma_{P,x,y}} \bigg/ \sum_{\substack{-l \leq y \leq l \\ y \neq x}} \frac{1.0}{\sigma_{P,x,y}} \tag{1}$$

This average value, which represents a very limited sequence-based prediction, does not contain information on whether or not local sequence increases or decreases the group's contact rate relative to what is expected by the nature of the group on its own. This was calculated by comparing it against values obtained by *Equation 1* for every instance of the matched system found in the training set, using the precomputed data to convert the average into a z-score.

$$Z_{x,l} = \left( R_{x,l} - \bar{R}_{x,l}^{DB} \right) / \sigma_{\bar{R}_{x,l}}^{DB} \tag{2}$$

We next tested the effect of sequence window size on the ability to predict total number of contacts for a given sequence by iteratively adding two more adjacent flanking residues in order to find an optimal window. While analyzing these scores for their ability to predict contact rates, we observed that the separate z-scores for long-range and short-range contacts, each derived from distinct non-overlapping sets of observations, are not independent; they each contain information on the probability expectation of the other, with sequence dependent correlations that could contain data on competition and cooperation.

## Step 3: Pi-contact predictor

To capitalize on the observation that short-range and long-range contact rates carry information on each other, we added an additional rate prediction step where the previous averaged rates are used in tandem short-range/long-range pairs to extract matching observation data from the original rate database. To do this we created a system of lookup tables in which our training set observations are tallied in two-dimensional arrays by half z-score bins. To obtain frequency values using these lookup tables we calculate the pair of z-scores for the query group and match them to a corresponding bin in the lookup tables.

Splitting observations into bins is problematic at extreme z-scores, as it significantly increases sampling error. To address this problem, we developed a smoothing method involving iterative sampling across a range of window-length dependent tables in order to use the natural random variance of the database to average the observations.

This method starts with the initial z-score calculation on a window of sequences covering the group itself and up to one flanking residue on either side, collecting values for observed contacts and total number of database entries from a look up table built using the same window size. We then add additional flanking residues, re-calculate the corresponding z-scores for the new window length, and then add the observed contacts and total entry numbers for that window size to the previous observation. From here, additional residues are added iteratively, one flanking residue and window specific score calculation at a time, up to a maximum window length of 40 residues on either side. The final frequency is determined by the total number of pi-contacts observed for similar groups in the database, summed over all windows, divided by the total number of similar groups found in the database. This is defined in the following equation, where $F$ is the contact frequency for position $x$, $O$ is the number of database pi-pi contacts observed within a bin defined

by a pair of long-range (LR) and short-range (SR) z-scores (Z) calculated for window length $i$, and $T$ is the total number of groups observed for the same bin.

$$F_x = \sum_{i=1}^{40} O\left(Z_{x,i}^{LR}, Z_{x,i}^{SR}\right) \Big/ \sum_{i=1}^{40} T\left(Z_{x,i}^{LR}, Z_{x,i}^{SR}\right) \tag{3}$$

Our selection criterion in developing this method, for determining scores, window lengths, grid spacing, and other details, was to test options against one another by filling statistical databases from defined 70% cuts of the training set and then testing how well the sum of frequency predictions made for each protein in the remaining 30% correlated with the total number of contacts found in each protein. Once our final pi-contact rate prediction method was selected we then filled our final databases using the full training set and did a single prediction quality test against the leave out database.

## Step 4: Phase separation prediction

As a starting point, we ran the pi-contact predictor developed for the PDB on each sequence in the PDB and in our 11 protein phase-separating protein training set and returned the highest number of contacts predicted for any 100 residue window. These contact frequency predictions showed a reasonably normal distribution for the PDB (skew and kurtosis of 0.22 and 0.62, as calculated using the scipy.stats python package) and above PDB-average predictions for our set of 11 phase-separating proteins (8/11 in the 99th percentile). Additional analysis showed that this enrichment was higher for long-range contact predictions than for short-range contacts. As an aside, the phase-separating protein with the lowest contact frequency, elastin, also has fewer sidechain groups and a lower average mass per residue. However, elastin still has a predicted contact frequency greater than the PDB average because of very high contact frequency predictions for its backbone.

Since different categories of contacts utilized in our predictor could have different effects on phase separation propensity, we trained a phase separation predictor by optimizing contact prediction weights and normalization methods against the ability to discriminate the lowest scoring member of the phase separation training set from the highest scoring members (the top percentile) of the non-redundant PDB set, by using a single score value per protein, as defined in the following equation where D is the relative discrimination score and S is the score function being tested.

$$D = \frac{\left(min\left\{S_{ps}\right\}_{ps=1}^{N_{ps}} - \bar{S}_{pdb}\right)}{\sigma_{\bar{S}_{pdb}}} - \frac{\left(\bar{S}_{pdbtop1\%} - \bar{S}_{pdb}\right)}{\sigma_{\bar{S}_{pdb}}} \tag{4}$$

We then ran the pi-contact predictor on every sequence with $\geq 140$ residues (based on the length of the smallest phase-separating protein in our training set) and stored prediction values for four contact frequencies (long-range/short-range vs. sidechain/backbone) and four corresponding relative frequencies (the average identity normalized z-scores produced as intermediate values during the contact prediction protocol).

To optimize weights against these eight values, we built a machine learning toolbox for stochastic optimization against our discrimination function. Both random and manual sampling of score functions and weights were tested, with weights determined by brute force sampling against a Metropolis Monte Carlo acceptance criterion, and score function changes tested by branching the weight optimization into parallel runs tested against the same criterion.

In terms of the general flow, we started with score functions that were a weight averaging of contact predictions over fixed window sizes throughout each sequence with the sequence score being the highest score observed. We used the optimization toolbox to test window sizes against one another, with each weight starting at the same value and allowed to deviate by sequential rounds of small random additions and subtractions. In order to avoid scoring the full set of PDB sequences for every small weight change we added fast diversification screening steps, where we introduced sub-optimization rounds against a small select subsets

of the PDB (100–2000 sequences), returning new weight sets to the primary optimization protocol for re-scoring against the full PDB and the primary acceptance/rejection step.

After the first few thousand rounds this results in a diverse population of weights associated with similar performance, which can be mined to capture the range of values that are acceptable for any given weight. At this point, we introduce weight changes using random selections from that population, instead of by random addition and subtraction, which allows for more aggressive sampling (how many weights can change per step, and how far they can change) at a lower observed rejection rate. This weight resampling method also allowed for better parallel computing, as the most successful weight combination were able to automatically propagate between processes.

During this process weight optimization was entirely stochastic, but we also tested a variety of score formulations and window lengths by parallel competition. This branched optimization strategy added a series of changes to the base protocol, including normalizing window averages by the number of carbon atoms in the window, adding multiple overlapping window lengths to allow for weights to differ by sequence distance, and finally, by changing how the final score is determined per sequence, which went from the top scoring position in the sequence, to the top decile position, and then to an average over all positions within 5 residues of the top scoring 60.

The final propensity score can be described as the following steps. Raw contact prediction numbers are generated for each residue in the sequence as an initial step. For every residue in the sequence, we then start iterating over the flanking residues while summing up values for eight contact prediction terms (long/short-range vs. backbone/sidechain vs. frequency/z-score) as well as one residue-based term (number of carbon atoms), with final sums being kept for three windows, defined as the closest 40, 80, and 120 flanking residues. Each set of window sums is then scored by the following equation, using the final set of 27 optimized weights, where $S$ is the window score associated with residue $x$, spanning the closest residues from $i = x$ to $i = N$, $w$ is a weight constant, $C$ is the number of carbon atoms in a residue, and $F$ and $Z$ are the pi-contact frequency and z-score values from *Equations 3 and 2.*

$$S_{N,x} = \left( \sum_{a=LR,SR} \sum_{b=BB,SC} \left( w_{f,a,b} \sum_{i=1}^{N} F_{i,a,b} \right) + \sum_{a=LR,SR} \sum_{b=BB,SC} \left( w_{z,a,b} \sum_{i=1}^{N} Z_{i,a,b} \right) \right) / \left( \sum_{i=1}^{N} C_i \right)^{w_c} \quad (5)$$

Finally, the full list of scores is then sorted, the top scoring 60 residues are identified, and an average score (T) for regions of the sequence associated with this high-scoring subset is obtained by summing over all positions found within 5 residues of the top 60. This value is then scaled by conversion to a z-score relative to our non-redundant PDB set.

$$PScore_{seq} = \frac{\left( T_{seq} - \bar{T}_{pdb} \right)}{\sigma_{\bar{T}_{pdb}}} \quad (6)$$

## Step 5: Retrospective analysis

One of the key limitations in designing the predictor is that phase separation in biological systems is not a fully defined phenomenon, and calculating true/false positive or negative rates are limited by the fact that there are very few gold standard positives to train against and there is no gold standard negative set. The vast majority of proteomic sequences are simply untested. Our experimental design attempts to address this by limiting training to a single test approach, validated against proteomic data a single time, but this setup meant that many design decisions were made blind, and the relative final impact of the different steps in the protocol remain untested.

To address this, we went through each step, creating matched predictors using the frequency and weight data available at the time, with new score functions returning the highest contact prediction sum observed over any 100 residue window. Results are shown in *Appendix 1—table 6*, demonstrating an increase in performance against the final test set for each training step.

We then compared the relative effects of different types of contacts, short-range vs. long-range and sidechain vs. backbone, by testing scores made from each side of the comparison, excluding the contact rates from the other. This analysis, also shown in *Appendix 1—table 6*, identifies baseline backbone pi-contact rates as being sufficient for the majority of the predictions, where the baseline observed rates for our 400 backbone peptide group definitions split the phase separation test set from the human proteome at an AUC of $0.841 \pm 0.027$, compared to $0.881 \pm 0.021$ for the final predictor. Score distributions demonstrating the sufficiency of long-range backbone contact predictions in recapitulating the phase separation predictions of the final predictor are shown in *Figure 5—figure supplement 2*, with panels A and B corresponding to training step 3, short-range backbone only and long-range backbone only, respectively, and panel C corresponding to the final predictor.

