## [Decision Letter]

Thank you for submitting your article "Pi-Pi Contacts: An Overlooked Protein Feature Relevant To Phase Separation" for consideration by *eLife*. Your article has been favorably evaluated by a Senior Editor and three reviewers, one of whom, Yibing Shan, is a member of our Board of Reviewing Editors.

The reviewers have discussed the reviews with one another and the Reviewing Editor has drafted this decision to help you prepare a revised submission.

Summary:

The manuscript by Vernon et al. describes an analysis of (planar) Pi-Pi interactions in proteins, and uses the resulting insights to further our understanding of liquid-liquid phase separating proteins. It concludes that Pi-Pi interactions are generally an important yet overlooked factor in folded proteins. Based on the analysis, the authors propose that these interactions are a key driver of phase separation. These are potentially important findings for our basic understanding of phase separation as well as for our understanding of protein structures.

Essential revisions:

1) The authors have identified trends of PDB structures without providing any information on the expected prior. Are there more planar Pi-Pi interactions involving non-aromatic residues because there are overwhelmingly more non-aromatic sidechains than aromatic sidechains in the PDB? Throughout the manuscript, the authors present analyses of enrichment/depletion or correlation of Pi-Pi interactions in various settings. These are generally presented clearly and honestly, but at the same time the authors do not adequately discuss the issues of confounding or indirect effects.

1a) The reviewers realize that it is not trivial to device a good prior, given that the distribution of amino acids are not random throughout protein structures. One of the reviewers suggested a prior as the following:

PDB analysis: The analysis in Figure 1 is very interesting, but a little hard to interpret when the numbers are not normalized by the expected background values. For example, there are ~5x more H/R/Q/N/E/D/G residues in the human proteome than there are Y/F/W residues. If we blindly assume pairwise interactions without any bias, we should naively expect ~20x more interactions amongst H/R/Q/N/E/D/G residues than amongst Y/F/W residues. This naive random network approximation is clearly not appropriate, but some accounting of the drastically different numbers of these residues should be included for the reader to make sense of what these analysis mean. Is 13x the expected value without any bias, given the combination of 3D distribution of residues across folded structures and the amino acid frequencies? For example, the counter hypothesis to this is that Y/F/W residues are typically buried in protein cores generally lack cavities, and so through a simply packing-effect there are necessarily many aromatic to non-aromatic contacts. If you consider other atom types (i.e., non sp2 carbon atoms) how do these compare?

When examining different types of inter-residue interactions (Figure 1) the sidechain and backbone accessible surface area (and numbers of sp2 atoms) differ wildly between different residues. The intention of this figure is to demonstrate that certain types of residues engage more frequently in planar Pi interactions, but without a prior that takes the number/size of those sidechains into account, can one infer any causation from the fact that bigger sidechains with more sp2 atoms are more frequently engaged in planar sp2-X interactions?

1b) Another example of the confounding effects is the enrichment of the Pi-Pi interactions in the active site (Appendix 1 Supplementary Results). Is it clear that this cannot be explained by the fact that active sites have more of the residues that can form the Pi-Pi interactions? Similarly, to what extent that the occurrence of Pi-Pi interactions near water molecules can be explained by the fact that a dominating majority of the residues are polar there? These issues are important because the authors need to establish not just a correlation, but also causation.

1c) It appears that Figure 1 are exclusively examining planar pi-pi interactions (given the absence of any sidechain interactions in residues that lack sp2 carbons in their sidechains). These analyses provide a relative picture with respect to one another, but there is no information on the global importance of these interactions in the greater context of all interactions in folded proteins. For example, how do these numbers compare to hydrogen bonds or to interactions between non- sp2 atoms and sp2 atoms. Such analysis is needed to understand if planar pi-pi interactions are more frequent than other planar interactions involving sp3 atoms in planar systems.

2) The authors develop a predictor and a PScore for phase separation based on the idea that Pi-Pi interactions are important for this behaviour. This is potentially of widespread interest, but the authors need to clarify the premise of a "phase separation predictor" which is important to this work conceptually. Assigning a score that will be referred to as a phase separation propensity assumes a gross simplification to the physics of phase separation. How should one think about this predictor? For a given set of solution conditions such as pH, salt concentration, type of salt, and temperature are we to think of the score as a predictor of the saturation concentration? Would it be of the low or high concentration arm of the binodal and what if the phase diagram is replete with complexities? It would help if the authors provide a coherent description of the key concepts and parameters that underlie the physics of phase separation for a generic protein under a given set of solution conditions, and with these parameters in hand explain if the Pscore serves as a proxy for one or more of the physical parameters. If so, the authors could assemble a data set from extant measurements to calibrate the Pscore. If the Pscore does not or should not be viewed as a proxy for any physical parameter, then this should be made clear and a detailed explanation of how the parameter should be used and what insight one might gather from its usage would be essential.

The authors fall short of explaining what signals the predictor captures and fully discussing its limitations. It is unclear what goes into the PScore. To this point, for polyNE the Pscore is -10, polyFG gives -6.5 and polyRG +3, while polyH gives +57, polyC gives +28 and polyD gives +12. The PScore is highest for a protein that is basically an Asn-Glu polyelectrolyte repeat, highly unlikely a sequence that will drive phase separation on its own. It will most certainly need a complexing partner and it is this type of caveats that can mislead a naive user. The PScore would be much more interesting if the authors can really demonstrate that the reason why it works (if it does work) is because of some specific mechanistic (e.g. Pi-Pi) insight. As it stands, the link is somewhat poorly explained because there is a lot of fitting and tweaking in the development of PScore. And so it is not clear why it works, and what physical mechanisms – if any – it captures. The authors should both discuss/demonstrate better the limitations of the PScore, and demonstrate that the reason it works is because of the specific mechanism proposed (rather than e.g. some correlated variable etc.).

Overall, the work that concerns the predictor is more of a preliminary nature and its weight should be reduced and the claims should be tone down in the revision.

2a) The authors effectively train a predictor on known sequences. While this has some basis in the information mined from the PDB, to what extent is the predictor simply finding sequences that are similar to known sequences? For example, what is the correlation between proteins identified in the authors' prior study published in Molecular Cell in 2015?

2b) The final optimization appears to involve separating known phase separating proteins from the top scoring proteins in the PDB. How can the authors distinguish between developing a predictor for phase separation vs. one for (long-range) intrinsic disorder? How well would a disorder predictor (e.g. one with a similarly long window size) do? This is important because the goal is not just to separate the two classes of proteins, but also to provide mechanistic insight. Are the Pi-Pi interactions causative, or is it a (spurious) correlation? As the authors are well aware, there are differences in amino acid composition between IDP/IDRs and folded proteins, that include some of the Pi-containing residues (in particular Asn, Glu, Phe, Tyr, Trp have rather different occurrences in IDPs vs. folded proteins). How can it be demonstrated that the optimization isn't merely picking out differences in disorder? One of the reviewers performed a rudimentary comparison between PScores and disorder for a subset of proteins and found a substantial correlation. The authors should use one of the better disorder predictors, perhaps creating a score like the one used (max score over a similar window) and see how well this performs compared to the one actually optimized from the liquid phase separation data.

2c) The authors show a number of examples of predictions in Appendix—Figure 9. It is, however, unclear how these proteins were chosen. It would be useful to have an argumentation for why these particular proteins were selected, since they are obviously not at the top of the list (e.g. FAK1 is #822). This is in part to ensure that the authors haven't selected proteins that they know, via other avenues, will form droplets. For example, one of the proteins presented (Androgen receptor) has been presented at a couple of conferences to form droplets. While I want to stress that I am not implying that the authors have somehow cheated, I think it is crucial that the authors haven't chosen the proteins in Appendix—Figure 9 based on specific knowledge that some of them have already been shown to form droplets. Many RGG domains have been shown to phase separate. Using the predictor to 'select' one (out of the entire proteome) does not help demonstrating the value of their predictor. It would be far more compelling if the authors found a sequence that does not share features with the set of sequences that are already known to phase separate.

2d) The PScore appears to provide the maximum value observed over any 100-residue window. If that's the case, then larger proteins will obviously have a greater likelihood of reaching a high PScore. While this may be fully realistic, it wasn't clear to me whether this has indeed been observed experimentally.

2e) The authors benchmark their predictions using a ROC analysis (Appendix—Figure 7). It is not fully clear what the "negative" set is, but it appears to be the entire proteome. Is that correct? In general, one should be cautious when using ROC analyses and reporting AUC when there is a large imbalance in the size of the two datasets. If the negative set is very large, one can get a rather large AUC even with a somewhat poor predictor. At the same time, if the goal is just to predict the top few percent hits, then a good predictor could have a relatively low AUC, but still find most true positives at a low false discovery rate (see e.g. Berrar and Flach, Briefings in Bioinformatics, 2011). What is the AUC when e.g. averaged over random subsets of "negatives" that are of the same size as the positive subset? Also, although it might become a bit arbitrary, it would be useful to have an idea on a "standard value" above which one would consider the predictions "reliable".

2f) The sequence tested phase separates at 1 mM after incubation on ice for 5 minutes. It would be surprising if, under those conditions, it had not phase separated. For context, many of the other proteins studied undergo phase separation at room temperature at concentrations that are 10^4 lower. If the selected region from FMR1 does undergo phase separation at something closer to physiological conditions including this would be critical, otherwise, this 'experimental validation' is less convincing.

3) The analyses of MD, Rosetta, FoldX are maybe interesting, but would require a much more detailed analysis which is out of scope for a revision. As is, the analyses fall far short of establishing that the Pi-Pi interaction is not represented in these computational models. It is difficult to believe that MD can fold so many proteins while entirely missing something supposed to be so fundamental. The importance of this work does not lie on the notion that the planar Pi-Pi interaction is not accounted for in existing computational models (molecular dynamics, FoldX, and Rosetta). The reviewers suggest the authors to strongly tone down the related statements, if not to delete the MD simulation analysis or move it to a supplemental section. Many of the strong assertions, including statements declaring that this analysis uncovers unappreciated facts should be toned down.

3a) Subsection “Prevalence of Pi contacts in the PDB” and in Figure 1: "These data suggest that the relative importance of pi-pi interactions are underestimated in the force-fields that are used in the structure calculations and thus appear more frequently in structures that are heavily constrained by experimental observations." It might be worth noting here that these are generally extremely primitive FFs, that are in general unrelated to those used e.g. in molecular dynamics simulations. The fact that these force fields do not include Pi-Pi interactions is not surprising, nor very interesting; these force fields are really minimal and capture almost no physical effects well.

3b) Although the analysis of the dynameomics simulations looks interesting, it should be noted that these simulations were performed with a relatively old force field, and that more recent and accurate force fields exist. Although all fixed charge force fields share the same basic functional form, the higher accuracy in more modern force fields would likely mean greater structural stability overall. Thus, whether or not the Pi-Pi interactions are lost in these simulations have little to say about the force fields that are more commonly used in the field nowadays. An interaction not accounted for explicitly in a MD force field could still be accounted for implicitly by the force field which is obtained by fitting to empirical data. For a reference point, past studies (Gallivan and Dougherty, PNAS 1999) showed that although pi-cation interactions are not explicitly accounted for in OPLS-AA force field for MD, but the energy calculation of Lys-Phe and Arg-Phe interaction using this force field correlates reasonably well with the quantum results. MD force fields have been much improved from then.

3d) Both in the MD calculations and Rosetta calculations the benchmark is the number of Pi-Pi contacts found in the crystalline state. In the case of salt bridges, it has been demonstrated by NMR in at least one case that a set of salt bridges found in the crystal structure are absent in solution (JH Tomlinson et al., JACS, 2009). In such cases, the expected (real) behaviour would be that the interactions found in the XR structure should dissolve in solution. From a cursory comparison between the contact frequencies in Xray structures (Appendix—Figure 2) and NMR structures (Appendix—Figure 2), it appears there are fewer contacts in the solution state – even when the NMR structures have many constraints. Could this be due to real differences between the two states (and also temperature differences), and would this imply that the loss of contacts e.g. upon minimization (Appendix—Figure 3) represents reality?

3e) The way the data is presented in Appendix—Figure 3 is confusing. Why not simply compare the experiment and calculated numbers? Also, the Protherm database is notoriously problematic, and curated versions exist (e.g. Conchuir et al., PLOS ONE, 2015 or Kellogg et al., Proteins, 2010).

---

## [Author Response]

Essential revisions:1) The authors have identified trends of PDB structures without providing any information on the expected prior. Are there more planar Pi-Pi interactions involving non-aromatic residues because there are overwhelmingly more non-aromatic sidechains than aromatic sidechains in the PDB? Throughout the manuscript, the authors present analyses of enrichment/depletion or correlation of Pi-Pi interactions in various settings. These are generally presented clearly and honestly, but at the same time the authors do not adequately discuss the issues of confounding or indirect effects.

We have augmented our statistical analyses and Discussion to address these concerns, as detailed below.

1a) The reviewers realize that it is not trivial to device a good prior, given that the distribution of amino acids are not random throughout protein structures. One of the reviewers suggested a prior as the following:PDB analysis: The analysis in Figure 1 is very interesting, but a little hard to interpret when the numbers are not normalized by the expected background values. For example, there are ~5x more H/R/Q/N/E/D/G residues in the human proteome than there are Y/F/W residues. If we blindly assume pairwise interactions without any bias, we should naively expect ~20x more interactions amongst H/R/Q/N/E/D/G residues than amongst Y/F/W residues. This naive random network approximation is clearly not appropriate, but some accounting of the drastically different numbers of these residues should be included for the reader to make sense of what these analysis mean. Is 13x the expected value without any bias, given the combination of 3D distribution of residues across folded structures and the amino acid frequencies? For example, the counter hypothesis to this is that Y/F/W residues are typically buried in protein cores generally lack cavities, and so through a simply packing-effect there are necessarily many aromatic to non-aromatic contacts. If you consider other atom types (i.e., non sp2 carbon atoms) how do these compare?

We have added normalizations to compare to expectations. For the contact frequency by residue type (Figure 1), we have added a line for the expectation given an average residue identity (the average across all residue types). This measurement is primarily to demonstrate the relative involvement of different residue types in the contacts, and does not address the root expectation issue, which is the question of whether or not these occur at the rate expected by random chance given the overall composition and packing of protein structures. To address this, we chose to normalize for packing by normalizing pi-contacts by the total number of VDW contacts, and to reduce the effect of amino acid composition by looking specifically at sidechain to backbone contacts. We have added a section describing this analysis, demonstrating that overall sp2 sidechains show enrichment in planar contacts relative to terminal planes taken from sp3 sidechains.

To examine whether sp^2^ containing sidechains engage in stacking behavior beyond what could be expected for average contact frequencies and overall packing considerations, we determined sidechain contacts to backbone peptide groups, focusing on the percentage of VDW contacts (with two or more pairs of atoms within 4.9Å) which satisfy our planar-pi criterion, and then compared the frequencies observed for sp^2^ sidechain groups to those observed for planar surfaces on the terminal end of sp^3^ sidechains, using atom groups as listed in the Materials and methods section. This metric addresses the issue of amino acid composition effects by taking advantage of the even distribution of backbone groups and allows for normalization of contact frequency for sidechains of different size. Enrichment of sp^2^ planar contacts relative to sp^3^ is clearly observed for all sp^2^ sidechains except Asn and Gln, which our previous analysis showed are more likely to form contacts with their backbone than with their sidechains (Figure 1—figure supplement 1). Further analysis of the relative frequency of planar pi VDW contacts to other VDW contacts as a function of resolution demonstrates that for some contact types the increased pi-contact frequencies with increasing resolution (lower values in Å) are at the expense of decrease in other VDW contacts, suggesting that these contacts represent a specific geometric constraint present in the experimental data, rather than an overall increase in VDW contact frequency at higher resolution (Figure 1—figure supplement 2). [subsection “Prevalence of Pi contacts in the PDB”, fourth paragraph]

When examining different types of inter-residue interactions (Figure 1) the sidechain and backbone accessible surface area (and numbers of sp2 atoms) differ wildly between different residues. The intention of this figure is to demonstrate that certain types of residues engage more frequently in planar Pi interactions, but without a prior that takes the number/size of those sidechains into account, can one infer any causation from the fact that bigger sidechains with more sp2 atoms are more frequently engaged in planar sp2-X interactions?

The data on pi-pi contacts as a percentage of total VDW contacts described in our response to point 1b addresses this issue by including the effect of sidechain size into the prior expectation, such that we can say that the largest sp2 sidechains show increased pi-contact frequency even when accounting for the increased number of VDW contacts in which they are involved.

1b) Another example of the confounding effects is the enrichment of the Pi-Pi interactions in the active site (Appendix 1 Supplementary Results). Is it clear that this cannot be explained by the fact that active sites have more of the residues that can form the Pi-Pi interactions?

We have expanded this section in both the main and supplementary text to clearly state the enrichment value for pi contacts in active sites after normalizing by residue type, and have added a table to the supplementary data (Appendix—Table 3) showing relative contact enrichment for each residue type independently.

We observe increased frequency of pi-pi contacts at positions with known catalytic function (Furnham et al, 2014), with enrichment of 1.87 ± 0.07 overall and 1.42 ± 0.07 when normalized by residue type (Appendix—Table 3). [subsection “Enrichment of pi-pi contacts in catalytic, capping and RNA-binding sites”]

We observe that residues in our non-redundant protein set which are annotated as being catalytic in the Catalytic Site Atlas (Furnham et al., 2014) (N=912 PDBs, 2914 catalytic residues) are more likely to be involved in planar pi-pi contacts than expected by the contact frequency of the given residue type, at 1.87 ± 0.07 times expectation overall and 1.42 ± 0.07 when normalized by catalytic residue frequency (Appendix—Table 3).

Similarly, to what extent that the occurrence of Pi-Pi interactions near water molecules can be explained by the fact that a dominating majority of the residues are polar there? These issues are important because the authors need to establish not just a correlation, but also causation.

Non-aromatic pi-containing sidechains are polar, and these are more common in general and particularly near solvated surfaces. To account for this, we now show data for the relationship between pi-pi contacts and water contacts after normalizing by the total number of VDW contacts, and have separated the statistics for all distinct sidechain-sidechain interactions independently, showing that pi-pi contact frequency increases with water contacts for the majority of distinct residue pair interactions, with some aromatic interactions being notable exceptions (Figure 3—figure supplement 2, referred to in the first paragraph of the subsection “Correlation of pi-pi contacts with solvation and lack of regular structure”). We have also been careful to avoid implying causation in this analysis, with the exception of the role of solvation in modulating contacts between sidechains of like charge.

1c) It appears that Figure 1 are exclusively examining planar pi-pi interactions (given the absence of any sidechain interactions in residues that lack sp2 carbons in their sidechains). These analyses provide a relative picture with respect to one another, but there is no information on the global importance of these interactions in the greater context of all interactions in folded proteins. For example, how do these numbers compare to hydrogen bonds or to interactions between non- sp2 atoms and sp2 atoms. Such analysis is needed to understand if planar pi-pi interactions are more frequent than other planar interactions involving sp3 atoms in planar systems.

We have added an enrichment comparison against a selection of sp3 planes, normalizing by overall VDW contact rates, showing that sp2 sidechains are more likely than sp3 sidechains to form planar contacts when normalized by total number of VDW interactions (Figure 1—figure supplement 1), as described in more detail in the response for point 1a.

However, the issue of global importance relative to other interactions is more difficult. Hydrogen bonds far outnumber pi-contacts, but the vast majority of protein hydrogen bond donors and acceptors are always in a hydrogen bond. They either form a hydrogen bond internally or they hydrogen bond with solvent. Thus, the global energetic importance is determined not by the presence of a hydrogen bond but by its relationship to solvent and other context specific non-pairwise energies. We now address an aspect of this issue by analyzing the co-prevalence of hydrogen bonding and pi-pi contacts (Appendix—Figure 4, referred to in the subsection “Enrichment of pi-pi contacts in catalytic, capping and RNA-binding sites”), demonstrating that non-aromatic pi sidechains are more likely to be observed in hydrogen bonds when they are in pi-pi contact. Addressing the question of the relative importance of pi-pi to other interactions more comprehensively involves energetic considerations that are outside of the scope of the current manuscript, particularly given our emphasis on the greater association of pi-pi interactions with lack of ordered structure and solvation, properties attributed to disordered protein regions.

2) The authors develop a predictor and a PScore for phase separation based on the idea that Pi-Pi interactions are important for this behaviour. This is potentially of widespread interest, but the authors need to clarify the premise of a "phase separation predictor" which is important to this work conceptually. Assigning a score that will be referred to as a phase separation propensity assumes a gross simplification to the physics of phase separation. How should one think about this predictor? For a given set of solution conditions such as pH, salt concentration, type of salt, and temperature are we to think of the score as a predictor of the saturation concentration? Would it be of the low or high concentration arm of the binodal and what if the phase diagram is replete with complexities? It would help if the authors provide a coherent description of the key concepts and parameters that underlie the physics of phase separation for a generic protein under a given set of solution conditions, and with these parameters in hand explain if the Pscore serves as a proxy for one or more of the physical parameters. If so, the authors could assemble a data set from extant measurements to calibrate the Pscore. If the Pscore does not or should not be viewed as a proxy for any physical parameter, then this should be made clear and a detailed explanation of how the parameter should be used and what insight one might gather from its usage would be essential.

We have clarified in the text that the phase separation property being predicted is defined as a binary classification of an observed physiological behavior.

We recognize that multiple physical interactions can contribute to driving phase separation (Brangwynne, Tompa and Pappu, 2015), but our goal was not to predict subtle differences in phase separation propensity or quantitative phase diagrams. Instead, we aimed to merely classify proteins as having the potential to self-associate under particular biological conditions or not, as a test of our hypothesis of the involvement of planar pi interactions. In this exercise, we define phase separating proteins as those that for presumed functional reasons self-associate in a way that is at least transiently reversible and dynamic, allowing for the protein to self-concentrate as a function of available protein concentration, temperature or other condition. This basic definition does not cover the complexity of the phase diagram, merely the ability to reversibly self-concentrate, and does not consider competing transitions, such as irreversible aggregation and precipitation, which have typically been selected against in the natural sequences on which the predictor is designed to be used. [subsection “Prediction of phase separation using pi-pi contacts”, first paragraph]

Detailed prediction of the physics of phase separation is hindered by the fact that the known phase-separating proteins have not been analyzed in a systematic fashion. Different categories of proteins are typically characterized in distinct ways according to the ways in which their phase separation behavior is specifically involved in physiological roles. However, others and we are interested in such a goal. This would include incorporation of other known physical interactions and we now explicitly define the potential aspects of such a predictor in the discussion of the limitations of the PScore, described below in the response to point 2c.

The authors fall short of explaining what signals the predictor captures and fully discussing its limitations. It is unclear what goes into the PScore. To this point, for polyNE the Pscore is -10, polyFG gives -6.5 and polyRG +3, while polyH gives +57, polyC gives +28 and polyD gives +12. The PScore is highest for a protein that is basically an Asn-Glu polyelectrolyte repeat, highly unlikely a sequence that will drive phase separation on its own. It will most certainly need a complexing partner and it is this type of caveats that can mislead a naive user. The PScore would be much more interesting if the authors can really demonstrate that the reason why it works (if it does work) is because of some specific mechanistic (e.g. Pi-Pi) insight. As it stands, the link is somewhat poorly explained because there is a lot of fitting and tweaking in the development of PScore. And so it is not clear why it works, and what physical mechanisms – if any – it captures. The authors should both discuss/demonstrate better the limitations of the PScore, and demonstrate that the reason it works is because of the specific mechanism proposed (rather than e.g. some correlated variable etc.).

The sequence features of the training and testing sets are all found in the PDB to a degree enabling reasonable computation of relative enrichment. This is not the case for simple repeat polymers such as polyC. We have added a section highlighting that the predictions are derived based on pi-contact frequencies observed in natural sequences, and that we specifically do not expect useful predictions on unnatural polymers.

We also note that the goal of the prediction experiment is to see whether observed phase separation can be predicted exclusively from contact probabilities as a test of the hypothesis that pi interactions are important for phase separation, but that our method uses probabilities found in the PDB, was trained on natural sequences, and was tested using sequences that are either found in nature or were designed based on sequences that are. The ability to predict contacts is expected to decrease for sequences not observed in nature and for sequences relying to a greater degree on other energetic contributions. [subsection “Prediction of phase separation using pi-pi contacts”, last paragraph]

In order to address physical mechanisms, we have added a section contrasting the features of phase separating sequences from non-phase separating disordered sequences, showing that disordered sequences which don’t phase separate have sp3 enrichment relative to phase separating sequences, and are especially enriched in lysine content.

Many high contact frequency residue types are also associated with disordered proteins in general, so to control for that potential role we took a selection of 3501 human proteins predicted to have long disordered regions (as described in the Materials and methods), split them by PScore into high (PScore ≥4) and low (PScore <1) subsets, and compared the sequence characteristics distinguishing high PScore and low PScore sequences (Appendix—Figure 8). We find that non-phase separating intrinsically disordered proteins are actually depleted in Gly and Pro, especially relative to the enrichment seen in phase separating sequences and sequences predicted to phase separate. Conversely, they are most enriched in Lys, which on average is depleted in phase separating sequences. [subsection “Mechanistic implications of the optimized phase separation predictor”, second paragraph]

Importantly, we have also reorganized how the prediction experiment is presented to stress that the predictor itself is the test of mechanistic relationship, highlighting a section previously relegated to the supplemental text, which shows that prediction by pi-contacts works even without the fitting and tweaking steps used to develop the final predictor.

In order to identify the contact features that play the largest role in the optimized predictor we did a retrospective analysis testing the predictive power of different scoring algorithms produced during the training process, and explored potential mechanistic implications by testing the power of individual score components, grouping contact predictions into long range vs. short range and backbone vs. sidechain (Appendix—Table 6). Our analysis shows that, while training did improve the predictor, a comparable result can be obtained by using only the long-range contact rate predictions for the peptide backbone (Figure 5—figure supplement 2, as further described in Appendix 1). This property significantly upweights the role of residues, especially Gly and Pro, that are associated with high overall backbone pi-pi contact frequencies and with lower short-range contact frequencies for local sidechain groups, and is especially important for predicting elastin-like proteins, which often have very few sp^2^-containing sidechains. Thus, these results highlight the increased availability of sp^2^ groups for non-local pi-interactions as a key driving force behind the phase separation predictions and is consistent with highly multivalent weak interactions leading to phase separation, both in non-polar structural proteins like elastin and highly charged RNA-binding proteins like FUS or Ddx4. [subsection “Mechanistic implications of the optimized phase separation predictor”, first paragraph]

We have also added a section to the Discussion describing potential mechanisms driving pi-contact formation, as well as our limitations in assessing these properties from statistical analysis of the PDB.

The physical nature of pi-pi contacts and their underlying mechanistic relationship to phase separation are not revealed by the simple contact frequency measurements used in our predictions. These contacts are observed in folded proteins, both internally and near solvated interfaces and, while that suggests they play a general role in the energetics of protein-protein interactions, the nature of that role is not clear. There is potential for electrostatic or induced dipole and quadrupolar interactions, especially in the context of other dipole interactions and hydrogen bonds, but the flat surfaces of sp^2^ groups could also enable solvation to drive contacts and lead to entropic contributions due to the relative freedom of movement inherent in packing flat plates, compared to the more rigid shape complementation involved in packing aliphatic groups. It is interesting to note that these proposed mechanisms could be affected by temperature in opposite ways, and that our predictor using pi contact frequencies is useful in identifying phase separating proteins regardless of whether they associate more readily as temperatures decrease (such as Ddx4) (Nott et al., 2015) or increase (such as elastin)(Yeo, Keeley and Weiss, 2011). [Discussion, third paragraph]

Overall, the work that concerns the predictor is more of a preliminary nature and its weight should be reduced and the claims should be tone down in the revision.

We have changed the Discussion section to begin by re-stating that the predictor is not just preliminary, it isn’t even a properly designed predictor, but is instead an experiment to test whether or not a single interaction frequency can provide general predictive value. In the Discussion section we now describe the many known properties of phase separating proteins that could be included in producing an actual single purpose predictive algorithm, which is a goal outside the scope of this work.

We tested the potential role played by pi-contacts in mediating phase separation by using the single property of pi-contact frequency to train a simplistic predictor of phase separation behavior found in natural sequences, finding that the single property of long-range pi-contact propensity is sufficient for marking the majority of known phase-separating proteins as outliers relative to the proteome, supporting the hypothesis that this sequence property is commonly associated with phase-separating proteins. While this association is demonstrably useful for identifying phase separating proteins in proteomic datasets, these contacts may not be the predominant interaction driving the physical process of phase separation for each case, and could instead reflect a modulatory role since it is not exclusive of other interactions like hydrogen bonds and charge interactions. However, tests showing that arginine to lysine mutations abrogate phase separation behavior do provide evidence of the importance of planar sp^2^ groups for phase separating systems.

The finding that a single contact potential can generate a reasonably accurate classifier of phase separation behavior suggests that a sequence-based prediction of phase separation behavior is a tractable problem, and that future development of an algorithm that can predict the complexities of the phase transition, environmental effects and concentration requirements is a reasonable goal. This goal could potentially be addressed by introducing the range of phase separation associated sequence properties that were intentionally excluded by our empirical test of the pi-contact association, including the electrostatic effects of charge patterning(Nott et al., 2015; Lin, Forman-Kay and Chan, 2016; Das and Pappu, 2013), multivalency of PTM sites and PTM binding motifs, and transient structural interactions, including strand formation (Murray et al., 2017) and coil-coil interactions (Conicella et al., 2016). There is also a role for incorporating predictions of competing states, the irreversible aggregation propensity of a sequence or its amyloidogenic potential. Incorporating annotation data associated with phase separating proteins could be another avenue for generating a physiological classifier in a more comprehensive predictor. [Discussion, first and second paragraphs]

2a) The authors effectively train a predictor on known sequences. While this has some basis in the information mined from the PDB, to what extent is the predictor simply finding sequences that are similar to known sequences? For example, what is the correlation between proteins identified in the authors' prior study published in Molecular Cell in 2015?

We have added a section describing similarity to the training set proteins, both to assess the average degree of similarity and to highlight how difficult it is to actually categorize low complexity sequences by similarity. For comparing similarity between the wide range of primarily non-homologous low complexity regions in our phase separation sets, we chose to compare dipeptide profiles, defined as the frequency of pairs of i & i+1 residue sets, enumerated over the full 400 pairs, with similarity measured by the sum of absolute distances between profiles. This metric captures data on both overall composition and specific motifs and repeats. While the division of the predictor into two distinct protocols was used to avoid scores that simply describe sequence similarity to the training set, it is still possible that the training process picked up on specific sequence features in the training set. To explore the contribution of sequence similarity to the score we made a measurement of sequence profile similarity based on dipeptide composition (neighboring residue pair frequencies). We compared the high scoring regions selected by the predictor to each of the sequences used in the training set (Appendix—Table 7, see Materials and methods). This analysis, shown in Appendix—Figure 8B, finds that high scoring (PScore ≥4.0) human proteins are, on average, more similar to the training set than are human proteins in general, but that the majority fall within the normal range. Comparison to a set of 1000 BLAST-level sequence homologs of the training set suggests that the majority of the similarity is compositional preference, not homology. [subsection “Mechanistic implications of the optimized phase separation predictor”, third paragraph]

Measuring the overall correlation of PScore values with previous classifiers is left out because in prior work the difficulty of comparing distant sequences was addressed by setting some minimum similarity threshold and then excluding analysis for everything outside of that. For example, the algorithm published in the Nott 2015 Molecular Cell work excludes sequences which don’t have specific residue types with defined spacings, and as such returns a flat score of 0.0 for ~75% of the human proteome and ~75% of our phase separation test set. It does perform well on the subset of proteins for which it can return a score, but that subset represents a small minority of phase-separating sequences.

2b) The final optimization appears to involve separating known phase separating proteins from the top scoring proteins in the PDB. How can the authors distinguish between developing a predictor for phase separation vs. one for (long-range) intrinsic disorder? How well would a disorder predictor (e.g. one with a similarly long window size) do? This is important because the goal is not just to separate the two classes of proteins, but also to provide mechanistic insight. Are the Pi-Pi interactions causative, or is it a (spurious) correlation? As the authors are well aware, there are differences in amino acid composition between IDP/IDRs and folded proteins, that include some of the Pi-containing residues (in particular Asn, Glu, Phe, Tyr, Trp have rather different occurrences in IDPs vs. folded proteins). How can it be demonstrated that the optimization isn't merely picking out differences in disorder? One of the reviewers performed a rudimentary comparison between PScores and disorder for a subset of proteins and found a substantial correlation. The authors should use one of the better disorder predictors, perhaps creating a score like the one used (max score over a similar window) and see how well this performs compared to the one actually optimized from the liquid phase separation data.

We have added a new section showing the relationship between disorder predictions and our phase separation prediction, highlighting that the correlation comes from the fact that the majority of known phase separating proteins are also disordered, but that they then represent a clear subset of all disordered proteins, distinctly different from non-phase separating disordered proteins in ways that are captured by the phase separation predictor but not by disorder predictors.

Even though the score is trained for discrimination against folded proteins we do not see a systematic increase in the scores of all disordered human proteins. Comparison against a top performing sequence homology-based disorder predictor (Disopred3, (Jones and Cozzetto, 2015)) and a physics-based disorder predictor (IUPREDLong (Dosztanyi et al., 2005)) shows that disorder predictors are better at discriminating disordered proteins from the PDB and the human proteome, while the PScore is consistently better at identifying phase-separating proteins (Appendix—Table 5). The majority of the proteins in our phase separation test set show disordered character, and the analysis shows that while PScore does correlate with disorder it only highlights a subset of disordered proteins, and does not reflect a general disorder prediction (Figure 5—figure supplement 1). As a direct test of this discrimination, we note that using the subset of human proteins with known intrinsic disorder (Piovesan et al., 2017) as the phase separation negative set shows similar results as using the human proteome as the negative, at AUC:0.84 ± 0.03 for the full test set and AUC:0.93 ± 0.02 for the in-vitro sufficient set. [subsection “Prediction of phase separation using pi-pi contacts”, sixth paragraph]

A section comparing the sequences of high PScore and low PScore disordered proteins has been added in order to describe these differences, showing that high PScore is associated with lower sequence complexity due to exclusion of a wide range of amino acid types relative to low PScore disordered proteins.

Both sequence similarity and compositional behavior can also be related to the bias towards disorder regions observed in phase-separating proteins. To characterize this, we again took the high and low PScore subsets of our set of 3501 human proteins predicted to have long disordered regions and then compared their sequence profiles. It has previously been observed that disordered proteins have a Shannon’s entropy (a measurement of sequence complexity) that is lower, but significantly overlapping with ordered proteins (Romero et al., 2001). We find here that the high PScore set has a Shannon’s entropy that is far lower than the range seen for low PScore disordered proteins, which have Shannon entropies that fall in the range observed for folded proteins (Appendix—Figure 8C). Comparing our phase separation test set with the human disprot set we can confirm that this bias towards lower complexity sequences is observed in known phase-separating sequences. [subsection “Mechanistic implications of the optimized phase separation predictor”, last paragraph]

2c) The authors show a number of examples of predictions in Appendix—Figure 9. It is, however, unclear how these proteins were chosen. It would be useful to have an argumentation for why these particular proteins were selected, since they are obviously not at the top of the list (e.g. FAK1 is #822). This is in part to ensure that the authors haven't selected proteins that they know, via other avenues, will form droplets. For example, one of the proteins presented (Androgen receptor) has been presented at a couple of conferences to form droplets. While I want to stress that I am not implying that the authors have somehow cheated, I think it is crucial that the authors haven't chosen the proteins in Appendix—Figure 9 based on specific knowledge that some of them have already been shown to form droplets. Many RGG domains have been shown to phase separate. Using the predictor to 'select' one (out of the entire proteome) does not help demonstrating the value of their predictor. It would be far more compelling if the authors found a sequence that does not share features with the set of sequences that are already known to phase separate.

We acknowledge the issue with the examples. Our goal was to show the diversity of function in the 943 human sequences above our confidence threshold so as to highlight the complexity and scale of the underlying predictions, but even selecting examples by known function ends up adding a lot of bias. We’ve clarified this issue by simplifying the list to the highest scoring individual examples in various function annotation categories, and by stating in the text that these examples do not represent a validation set. Since we also don’t want examples that are already known to phase separate, we’ve removed androgen receptor and replaced it with the vitamin D receptor.

A selection of high-scoring human proteins associated with enriched functions are shown with per-residue scores and PTM annotations in Appendix—Figure 9, with examples chosen from the highest scoring protein in any given gene ontology function/localization annotation related to neuronal plasticity or behavior in A, cytoskeletal biomaterials in B, signaling in C, and extracellular biomaterials in D. [subsection “Analysis and validation of predictions of phase separation”, fifth paragraph]

In response to the issue with selecting a single RGG domain to test, we have also selected two sequences which, as recommended, do not share features with the set already known to phase separate, and have confirmed their ability to phase-separate, as described in the response below to point 2i.

2d) The PScore appears to provide the maximum value observed over any 100-residue window. If that's the case, then larger proteins will obviously have a greater likelihood of reaching a high PScore. While this may be fully realistic, it wasn't clear to me whether this has indeed been observed experimentally.

The summation score used by the final predictor was selected by the training process, in which both length-biased (highest scoring region) and length-independent (percentile values) scores were trialed against the acceptance criterion. The best performing setup involves taking the highest scoring 60 residue positions, even if scattered throughout a larger sequence, and then averaging the score of all residues within five positions of any of those sixty. The length bias and opportunity for distributed patches of motifs that this provides are then actually part of the training, and in essence represents a general prediction. Length dependency has also been observed for polymer systems that phase separate and we do have experimental data on Ddx4 confirming these trends (unpublished data). Since it is not a systemic test, however, we feel it falls outside of the scope of the manuscript.

2e) The authors benchmark their predictions using a ROC analysis (Appendix—Figure 7). It is not fully clear what the "negative" set is, but it appears to be the entire proteome. Is that correct? In general, one should be cautious when using ROC analyses and reporting AUC when there is a large imbalance in the size of the two datasets. If the negative set is very large, one can get a rather large AUC even with a somewhat poor predictor. At the same time, if the goal is just to predict the top few percent hits, then a good predictor could have a relatively low AUC, but still find most true positives at a low false discovery rate (see e.g. Berrar and Flach, Briefings in Bioinformatics, 2011). What is the AUC when e.g. averaged over random subsets of "negatives" that are of the same size as the positive subset? Also, although it might become a bit arbitrary, it would be useful to have an idea on a "standard value" above which one would consider the predictions "reliable".

We use a range of negative sets to show the discriminatory power measured against different expectations. The PDB test set and *E. coli* proteome are closer to a gold standard negative set, but the lack of disordered proteins means that discrimination is too easy so we focus primarily on the human proteome, where the AUC represents a lower bound for discrimination based on a likely to be incorrect assumption that phase separation is a very rare property. To test the effect of set size we have run the suggested calculations, measuring AUC over random negative subsets with the same size as the positive, and the average values are consistently within bootstrap error, at +/- 0.01 AUC.

We have clarified in the text that a PScore of ≥ 4.0 is our standard confidence value.

Using a defined standard confidence threshold of ≥ 4.0 standard deviations from the PDB average for the propensity score (PScore) captures 0.3%, 2.2%, and 5.1% of the *E. coli, S. cerevisiae*, and human proteome sets, respectively, as compared to 0.1% of our full PDB set and 81% (26/32) of the self-sufficient for in vitro phase separation test set (dropping to 36/62 for the entire proteomic test set and to 35/59 for the synthetic test set). [subsection “Prediction of phase separation using pi-pi contacts”, fifth paragraph]

2f) The sequence tested phase separates at 1 mM after incubation on ice for 5 minutes. It would be surprising if, under those conditions, it had not phase separated. For context, many of the other proteins studied undergo phase separation at room temperature at concentrations that are 10^4 lower. If the selected region from FMR1 does undergo phase separation at something closer to physiological conditions including this would be critical, otherwise, this 'experimental validation' is less convincing.

The conditions used to assess phase separation of FMR1 were chosen primarily to allow the visual confirmation of liquid behavior and reversibility for a physiologically reasonable buffer solution that only contains FMR1, where the goal was to confirm the physical property. We do have data showing FMR1 phase separating at lower concentrations in more physiological conditions, and have included a figure (Figure 8—figure supplement 1) showing that the addition of even a small amount of crowding agent (20 mg/ml ficol) is enough to reduce the concentration required for visual confirmation of phase separation five-fold (from 1 mM to 200 µM). We have additional data involving the effects of known physiological binding partners, but that begins to move away from the predicted property of self-sufficient phase-separation, so we have left it out.

Instead of further characterizing FMR1 we have also addressed the experimental validation issue by testing two additional proteins that do not fall in any of the sequence or structure classes currently associated with phase separation and reporting their ability to phase separate.

To test whether or not the predictor is applicable to sequences that do not share motifs or functions with any of our training set proteins we did a manual search for predictions with sequence properties and functions dissimilar from the training set proteins and selected two proteins, human engrailed-2 (UID: P19622, PScore 5.0), a DNA binding homeobox protein, and the pAP isoform of the Human cytomegalovirus capsid scaffolding protein (UID: P16753-2, PScore 3.8), a protein that plays an essential structural role in assembling the viral capsid, a novel function relative to those known to involve phase separation. Both sequences have little overlap with any of the sequence motifs found in our training set (Appendix—Table 7), aside from general enrichment in glycine and proline residues. Experimentally we observe reversible liquid phase separation of pAP protein with increasing temperature, with viscoelastic properties similar to the complex coacervation of elastins (Figure 8). We did not observe phase separation of engrailed-2 under the same buffer conditions, even at 1mM protein concentration, but did observe temperature dependent liquid droplet formation in the presence of a crowding reagent (20mg/ml ficol) (Figure 8—figure supplement 1). While these observations do not represent a robust or comprehensive test of prediction quality, they do suggest that the predictions provide a useful tool for selecting natural proteins capable of self-sufficient liquid demixing. [subsection “Analysis and validation of predictions of phase separation”, last paragraph]

3) The analyses of MD, Rosetta, FoldX are maybe interesting, but would require a much more detailed analysis which is out of scope for a revision. As is, the analyses fall far short of establishing that the Pi-Pi interaction is not represented in these computational models. It is difficult to believe that MD can fold so many proteins while entirely missing something supposed to be so fundamental. The importance of this work does not lie on the notion that the planar Pi-Pi interaction is not accounted for in existing computational models (molecular dynamics, FoldX, and Rosetta). The reviewers suggest the authors to strongly tone down the related statements, if not to delete the MD simulation analysis or move it to a supplemental section. Many of the strong assertions, including statements declaring that this analysis uncovers unappreciated facts should be toned down.

We appreciate this point and have toned down our statements related to empirical force fields, as detailed below.

3a) Subsection “Prevalence of Pi contacts in the PDB” and in Figure 1: "These data suggest that the relative importance of pi-pi interactions are underestimated in the force-fields that are used in the structure calculations and thus appear more frequently in structures that are heavily constrained by experimental observations." It might be worth noting here that these are generally extremely primitive FFs, that are in general unrelated to those used e.g. in molecular dynamics simulations. The fact that these force fields do not include Pi-Pi interactions is not surprising, nor very interesting; these force fields are really minimal and capture almost no physical effects well.

The analysis has been toned down to make it clear that the goal is to confirm that the frequency of observed pi interactions is driven by experimental constraints and is not derived from the commonly used force fields employed in refining protein structures.

Modeling and analysis of protein structures typically involves the use of coarse-grained energy functions. To test the degree to which contact frequencies in solved structures derive from experimental constraints, rather than the force fields used, we explored how well planar pi interactions are captured by the simple energy functions used in certain protein modeling protocols. We examined a few different modeling protocols by either running available methods or downloading pre-computed datasets (see Materials and methods). In general, planar pi-pi contacts were lost during simulations (Appendix—Figure 3) and energy minimization (Appendix—Figure 3). In one older molecular dynamics simulation of folded proteins, made available for 100 proteins via Dynameomics (Kehl et al., 2008), 90% of the planar pi-pi contacts found in the starting structures were lost during simulation, with the majority being lost within the first few simulation steps. Similarly, modeling of the energetic effect of mutations, the ∆∆G of unfolding, using both FOLDX (Schymkowitz et al., 2005) and Rosetta (Kellogg, Leaver-Fay and Baker, 2011), shows decreased prediction accuracy at positions involved in pi contacts (Appendix—Figure 3), based on comparison to a reference set of ∆∆G measurements (Bava et al., 2004).

These observed issues in modeling pi-contacts may be overcome by more recent and sophisticated energy functions, but our results are consistent with the inherent energetic importance of planar pi interactions, rather than their observation being due to simple force fields used in refining protein structures. [subsection “Prevalence of Pi contacts in the PDB”, last paragraph]

3b) Although the analysis of the dynameomics simulations looks interesting, it should be noted that these simulations were performed with a relatively old force field, and that more recent and accurate force fields exist. Although all fixed charge force fields share the same basic functional form, the higher accuracy in more modern force fields would likely mean greater structural stability overall. Thus, whether or not the Pi-Pi interactions are lost in these simulations have little to say about the force fields that are more commonly used in the field nowadays. An interaction not accounted for explicitly in a MD force field could still be accounted for implicitly by the force field which is obtained by fitting to empirical data. For a reference point, past studies (Gallivan and Dougherty, PNAS 1999) showed that although pi-cation interactions are not explicitly accounted for in OPLS-AA force field for MD, but the energy calculation of Lys-Phe and Arg-Phe interaction using this force field correlates reasonably well with the quantum results. MD force fields have been much improved from then.

We have clarified in the text that this is not a systemic analysis of modern force fields, and specifically refer to the database as “older”, shown above in the response to point 3b.

3d) Both in the MD calculations and Rosetta calculations the benchmark is the number of Pi-Pi contacts found in the crystalline state. In the case of salt bridges, it has been demonstrated by NMR in at least one case that a set of salt bridges found in the crystal structure are absent in solution (JH Tomlinson et al., JACS, 2009). In such cases, the expected (real) behaviour would be that the interactions found in the XR structure should dissolve in solution. From a cursory comparison between the contact frequencies in Xray structures (Appendix—Figure 2) and NMR structures (Appendix—Figure 2), it appears there are fewer contacts in the solution state – even when the NMR structures have many constraints. Could this be due to real differences between the two states (and also temperature differences), and would this imply that the loss of contacts e.g. upon minimization (Appendix—Figure 3) represents reality?

The temperatures at which crystal structures are solved likely play a role in the frequency of contacts, but, other than crystal packing artifacts, those contacts do represent a real low energy state for the groups involved. In terms of assessing whether or not the NMR frequencies represent the real solution state behavior, we point out that i) the top values obtained from the most constrained quartile measurements in Appendix—Figure 2 do not imply that those constraints are comprehensive, ii) even in NMR structures with many constraints overall specific residues types can be still be left completely unassigned, and iii) in the handful of structures in which every sp2 group type can be observed in the constraints there are never any constraints to oxygen atoms, so planar geometry of oxygen containing groups is largely unconstrained. This problem with the geometry of the constraints is true even for otherwise well-assigned residues like aromatic sidechains, because there is also the issue that planar atoms are often ambiguous, and modeling protocols have a variety of ways of treating that. One of the most common ways involves treating ambiguous pairs of atoms (e.g., Phe Hdelta1, Hdelta2) as a single pseudoatom located at the center of the plane, and we can show that this modeling decision alone results in a significant drop in pi-contact frequencies. Thus, while the actual solution state sampling rate of these contacts is expected to be lower than those seen in crystal structures solved at low temperatures, even the highest frequencies observed in the NMR set are likely lower than the expected real solution state behavior. By extension, the loss of 90% of contacts upon minimization would not represent reality.

3e) The way the data is presented in Appendix—Figure 3 is confusing. Why not simply compare the experiment and calculated numbers? Also, the Protherm database is notoriously problematic, and curated versions exist (e.g. Conchuir et al., PLOS ONE, 2015 or Kellogg et al., Proteins, 2010).

We have redone this analysis with the Conchuir et al. curated version, and have changed the presentation to focus on correlation values (both with and without the removal of outliers). We have also used the ΔΔG prediction values from Rosetta, present in the curated dataset, to add a side-by-side comparison with FOLDX (Appendix—Figure 3).

Similarly, modeling of the energetic effect of mutations, the ∆∆G of unfolding, using both FOLDX (Schymkowitz et al., 2005) and Rosetta (Kellogg, Leaver-Fay and Baker, 2011), shows decreased prediction accuracy at positions involved in pi-contacts (Appendix—Figure 3), based on comparison to a reference set of ∆∆G measurements (Bava et al., 2004). [subsection “Prevalence of Pi contacts in the PDB”, last paragraph]